# Sporadicity in Decentralized Federated Learning: Theory and Algorithm

## Abstract

Decentralized federated learning methods are a family of techniques employed by devices in a distributed setup to (i) reach consensus over a common model which (ii) is optimal with respect to the global objective function. As this is carried out without the presence of any centralized server, prominent challenges of conventional federated learning become even more significant, namely heterogeneous data distributions among devices and their varying resource capabilities. In this work, we propose *Decentralized Sporadic Federated Learning* (`DSpodFL`), which introduces sporadicity to decentralized federated learning. `DSpodFL` includes sporadic stochastic gradient calculations and model exchanges for aggregations. Our motivation is to achieve joint computation and communication savings without losing statistical performance. We prove that by using a constant step size, our method achieves a geometric convergence rate to a finite optimality gap. Through numerical evaluations, we demonstrate the resource savings achieved by `DSpodFL` compared to the existing baselines.

## 1 Introduction

Federated Learning (FL) has exploded in popularity as a privacy-preserving method for distributed machine learning (ML) (McMahan et al., 2017). In conventional FL, clients (e.g., edge devices) are connected to a central server (e.g., a cellular base station) via a star topology configuration (Konečný et al., 2016). In this setup, the FL process iterates between (i) client-side local model updates, via stochastic gradient descent (SGD) on local datasets, and (ii) server-side global aggregations/synchronizations based on uploaded local models (Bonawitz et al., 2019).

Relying on a central server for model aggregations in FL poses several challenges, including the presence of a single point of failure, and burden on client-server communication resources. To address this, recent research has proposed Decentralized Federated Learning (DFL) (Koloskova et al., 2020) to eliminate the server's role in FL. In DFL, clients conduct gradient descent on their local models similar to FL, but exchange their models with neighboring clients instead of a central server to form consensus-driven aggregations over local networks (Huang et al., 2022).

However, the literature on DFL (Koloskova et al., 2020; Huang et al., 2022; Liu et al., 2022) and distributed optimization more generally (Nedić et al., 2018) has largely overlooked the impact of heterogeneity in the resources available for computation and communication among clients (Li et al., 2020). This heterogeneity causes (i) computing gradients at every iteration to be costlier (e.g., in terms of energy consumption and delay) at devices with weaker/slower processing units, and (ii) higher transmission delays for clients with low-quality communication links (e.g., lower available bandwidth and blocked wireless channels) (Zehtabi et al., 2022a), among other impacts. As a result, in existing DFL methods, resource-abundant devices often remain idle as they wait to receive local models from stragglers – devices that take more time to finish their local computations and model transmissions – before starting the next round of model training. In this work, we address these shortcomings of existing DFL approaches through the following contributions:

- We propose *Decentralized Sporadic Federated Learning* (`DSpodFL`), which formalizes the notion of *sporadicity* in DFL (Section 3). `DSpodFL` systematically reduces the frequency of both SGD computations and ML model aggregations by allowing devices to conduct these processes intermittently without delaying DFL. Our framework based on indicator variables can accommodate a wide range of device participation profiles encountered due to heterogeneity.

| Paper | Decentralized | Sporadic SGDs | Sporadic Aggregations | Generalized Data Heterogeneity [1] |
|---|:---:|:---:|:---:|:---:|
| Koloskova et al. (2020) | ✓ | | ✓ | ✓ |
| Yang et al. (2022) | | ✓ | ✓ | |
| Wang & Ji (2022) | | | ✓ | |
| Wang & Nedic (2022) | ✓ | | | |
| **Ours** | ✓ | ✓ | ✓ | ✓ |

*Table 1: Summary comparison of our work against recent key papers, across key dimensions of DFL.*

- We analytically characterize the convergence behavior of `DSpodFL` under mild assumptions on the communication graph, data heterogeneity, and gradient noises (Sec. 4). Our analysis reveals that `DSpodFL` obtains a geometric convergence rate to a neighborhood of the globally optimal ML model for a constant SGD step size (Theorem 4.1). We show that this optimality gap is directly proportional to the step size, but also depends on the connectivity of the communication graph, number of devices, frequency of computations, frequency of communications, and the dataset.

- Our numerical experiments (Section 5) demonstrate that `DSpodFL` obtains significant improvements over DFL baselines in terms of achievable accuracy for different communication and computation delays incurred. Further, we find that `DSpodFL` provides robustness to the degree of data diversity across devices, and most efficiently leverages increases in graph connectivity.

## 2 RELATED WORK

We compare our work against others in terms of key dimensions of DFL, summarized in Table 1.

**Sporadic SGDs.** In our paper, sporadicity in conducting SGD is modeled through devices using different step sizes (possibly 0) at each iteration. A similar idea of uncoordinated step sizes have emerged in the gradient-tracking literature (Nedić et al., 2017; Xin et al., 2019; Pu et al., 2020). The recent work Wang & Nedic (2022) is the closest to our paper in this regard, as it also focuses on distributed gradient descent (DGD). Nevertheless, our focus in our paper is fundamentally different, as it is mainly motivated by achieving resource efficiency as opposed to privacy preservation via uncoordinated step sizes. To achieve this, `DSpodFL` allows for a broader range of scenarios where each device's step size can jump between 0 and a positive value throughout the iterations. Only Pu et al. (2020) allows for some of the devices to choose a 0 (but non-varying) step size.

Other works in centralized ML (Stich, 2018; Lin et al., 2019; Woodworth et al., 2020) and conventional FL (McMahan et al., 2017; Karimireddy et al., 2020; Mishchenko et al., 2022) have proposed algorithms with multiple local SGDs between consecutive model aggregations. However, these works focus on a fixed number of SGDs across clients. In this respect, Yang et al. (2022) proposes Anarchic FL, in which at each round of training the number of SGD steps can differ for each client. By contrast, we study the DFL setting, which introduces the dimension of decentralized consensus-based aggregations. Although decentralized counterparts of fixed local SGD methods have recently been studied (Nguyen et al., 2022; Liu et al., 2023), the consequences of enabling each device to choose its own number of local SGDs are not yet well understood. Our theoretical analysis quantifies these effects, in the binary case of each client either conducting an SGD and/or communicating or not in each iteration. This is the most practical case in a fully decentralized scenario, where it is unclear how to coordinate all devices to compute fixed number of local SGDs before collaborating.

**Sporadic Aggregations.** On the other hand, the component of sporadicity in communications (aggregations) for our proposed method puts this paper in the same line of work as randomized gossip algorithms. Works such as Even et al. (2021); Boyd et al. (2006); Pu & Nedić (2021) study gossip algorithms with only two devices performing aggregations at each iteration, while Koloskova et al. (2019; 2020); Kong et al. (2021); Chen et al. (2021) allow more general mixing matrices, similar to our work. Moreover, works such as Srivastava & Nedic (2011); Zhao & Sayed (2014); Lian et al. (2018) focus on asynchronous DFL, a similar idea to devices conducting sporadic aggregations. Different from these works, we analyze the effect of each device communicating with a subset of its neighbors at each iteration simultaneously with sporadic SGDs. We also note that in conventional FL, some works have studied client selection methods (Wang & Ji, 2022; Nguyen et al., 2020), which correspond in effect to sporadic aggregations implemented by a central server.

---

[1] See Assumption 4.2-(c), where $0 \leq \zeta \leq 2\beta$ in general, and need not be $\zeta = 0$.

In this work, we focus on enhancing resource efficiency through sporadicity. Other approaches Tang et al. (2018); Koloskova et al. (2019); Zhao et al. (2022); Vogels et al. (2020); Taheri et al. (2020); Lu & De Sa (2020); Shen et al. (2018) have considered the effects of compression, quantization, and sparsification for communication-efficient FL. A family of network topologies with better mixing efficiencies is studied in Song et al. (2022). However, these techniques are focused on communication efficiency, and do not take processing efficiency into account. Our method allows for processing efficiency as well as introducing sporadicity into gradient calculations.

## 3 METHODOLOGY

We first formalize `DSpodFL`'s architecture and sporadicity. A summary of notation used throughout this paper can be found in Appendix A. Appendix B contains the full pseudocode of `DSpodFL`.

### 3.1 DECENTRALIZED FL WITH SPORADICITY

We consider a decentralized FL architecture consisting of $m$ devices indexed by $\mathcal{M} := \{1, \ldots, m\}$. Training proceeds in a series of iterations $k = 1, ..., K$. At each time $k$, the devices are connected through a set of communication links modeled as a time-varying graph $\mathcal{G}^{(k)} = (\mathcal{M}, \mathcal{E}^{(k)})$, where $\mathcal{E}^{(k)}$ denotes the edge set of the graph. The goal is for all the devices to learn the globally optimal model $\theta^\star = \arg\min_{\theta \in \mathbb{R}^n} F(\theta)$, while conducting stochastic gradient descent (SGD) on their local objectives $F_i(\theta)$ and exchanging model parameters with their one-hop neighbors, where

$$F(\theta) = \frac{1}{m} \sum_{i=1}^m F_i(\theta), \qquad F_i(\theta) = \sum_{(\mathbf{x},y) \in \mathcal{D}_i} \ell_{(\mathbf{x},y)}(\theta), \tag{1}$$

in which $\mathcal{D}_i$ is the local dataset of device $i \in \mathcal{M}$, $(\mathbf{x}, y)$ denotes a data point with features $\mathbf{x}$ and label $y$, and $\ell_{(\mathbf{x},y)}(\theta)$ is the loss incurred by ML model $\theta$ on a data point $(\mathbf{x}, y)$.

In the DFL setup, each device $i \in \mathcal{M}$ has its own ML model $\theta_i$. Therefore, achieving the goal of minimizing the global objective function in equation 1 should be achieved under $\theta_1 = \theta_2 = \cdots = \theta_m = \theta^\star$. This means that the devices need to also reach a consensus over their model parameters alongside implementing gradient descent (Nedic, 2020). To this end, it is necessary to consider an effective consensus mechanism in a fully decentralized setup of DFL.

To achieve this, we propose `DSpodFL`, where each device updates its local model as follows:

$$\theta_i^{(k+1)} = \theta_i^{(k)} + \underbrace{\sum_{j=1}^m r_{ij} \left( \theta_j^{(k)} - \theta_i^{(k)} \right) \hat{v}_{ij}^{(k)}}_{\text{Aggregation}} - \underbrace{\alpha^{(k)} \mathbf{g}_i^{(k)} v_i^{(k)}}_{\text{SGD}}, \tag{2}$$

where $\theta_i^{(k)}$ is the vector of model parameters of device $i$ at iteration $k$, $\mathbf{g}_i^{(k)} = \nabla F_i(\theta_i^{(k)}) + \epsilon_i^{(k)}$ is the local stochastic gradient of device $i$ at iteration $k$ with $\epsilon_i^{(k)}$ being the SGD noise. Also, $v_i^{(k)}$ is a binary indicator variable, capturing the sporadicity in SGD iterations, which is 1 if the device performs SGD in that iteration and 0 otherwise. Also, $\hat{v}_{ij}^{(k)}$ is an indicator variable, capturing the sporadicity in model aggregations, which indicates whether the link $(i, j)$ is being used for communications at iteration $k$ or not. The mixing weight assigned to the link $(i, j)$ is denoted as $r_{ij}$ and can be defined using the Metropolis-Hastings heuristic (Boyd et al., 2004) as $r_{ij} = \frac{1}{1 + \max\{|\mathcal{N}_i|, |\mathcal{N}_j|\}}$ when $i \neq j$, and 0 if $i = j$ or $j \neq \mathcal{N}_i$, in which $\mathcal{N}_i$ is the set of neighbors of device $i$. The Metropolis-Hastings heuristic is a convenient method for generating doubly stochastic matrices which are both average-preserving and consensus-generating. However, the mixing weight $r_{ij}$ can be chosen differently as long as the matrix $\mathbf{R} = [r_{ij}]_{1 \leq i,j \leq m}$ is doubly stochastic.

**Interpreting Sporadicity.** The novelty of our proposed algorithm lies in the introduction of sporadicity (i.e, indicator variables $v_i^{(k)}$ and $\hat{v}_{ij}^{(k)}$) for the purpose of enhancing resource-efficiency in DFL. Specifically, we can achieve computational efficiency by setting $v_i^{(k)} = 0$ in iterations where computing new SGDs at device $i \in \mathcal{M}$ does not significantly benefit the *statistical/inference* performance of the decentralized system. Similarly, we can achieve communication efficiency by setting $\hat{v}_{ij}^{(k)} = 0$ when using link $(i, j)$ at iteration $k$ for exchanging model parameters among devices $i$ and $j$ does not considerably improve the statistical/inference performance of the system.

## 3.2 Matrix Form of Model Updates

Henceforth, we presume two properties for the indicator random variables in `DSpodFL`:

$$\mathbb{E}_{v_i^{(k)}}\left[v_i^{(k)}\right] = d_i^{(k)}, \qquad \mathbb{E}_{\hat{v}_{ij}^{(k)}}\left[\hat{v}_{ij}^{(k)}\right] = \mathbb{E}_{\hat{v}_{ji}^{(k)}}\left[\hat{v}_{ji}^{(k)}\right] = b_{ij}^{(k)} = b_{ji}^{(k)},$$

in which $d_i^{(k)} \in (0, 1]$ captures device $i$'s probability of conducting SGD at iteration $k$, and $b_{ij}^{(k)} \in (0, 1]$ captures the probability of link $(i, j)$ being used for communication at iteration $k$.[2] In addition, we define $d_{\max}^{(k)} = \max_{i \in \mathcal{M}} d_i^{(k)}$ and $d_{\min}^{(k)} = \min_{i \in \mathcal{M}} d_i^{(k)}$. Note that the probability distributions of these indicator variables can be time-varying.

We can rewrite the update rule given in equation 2 compactly as

$$\mathbf{\Theta}^{(k+1)} = \mathbf{P}^{(k)}\mathbf{\Theta}^{(k)} - \alpha^{(k)}\mathbf{V}^{(k)}\mathbf{G}^{(k)}, \tag{3}$$

where $\mathbf{\Theta}^{(k)}$ and $\mathbf{G}^{(k)}$ are matrices with their rows comprised of $\theta_i^{(k)}$ and $\mathbf{g}_i^{(k)}$, respectively, and $\mathbf{V}^{(k)}$ is a diagonal matrix with $v_i^{(k)}$ as its diagonal entries for $1 \leq i \leq m$. Note that $\mathbf{G}^{(k)} = \nabla^{(k)} + \mathbf{E}^{(k)}$, where $\nabla^{(k)}$ (respectively, $\mathbf{E}^{(k)}$) is the matrix whose $i$ th row is $\nabla F_i(\theta_i^{(k)})$ (respectively, $\epsilon_i^{(k)}$) for $1 \leq i \leq m$. Furthermore, the elements of $\mathbf{P}^{(k)} = [p_{ij}^{(k)}]_{1 \leq i, j \leq m}$ are defined as

$$p_{ij}^{(k)} = \begin{cases} r_{ij}\hat{v}_{ij}^{(k)} & i \neq j \\ 1 - \sum_{j=1}^{m} r_{ij}\hat{v}_{ij}^{(k)} & i = j \end{cases}. \tag{4}$$

Note that the random matrix $\mathbf{P}^{(k)}$, by definition, is *doubly stochastic and symmetric* with non-negative entries, i.e., $\mathbf{P}^{(k)}\mathbf{1} = \mathbf{1}$ and $\left(\mathbf{P}^{(k)}\right)^T = \mathbf{P}^{(k)}$.

Finally, we define the row vector $\bar{\theta}^{(k)}$ as the average of the models parameters of all the $m$ devices, which based on equation 3, we can get

$$\bar{\theta}^{(k+1)} = \bar{\theta}^{(k)} - \alpha^{(k)}\overline{\mathbf{g}v}^{(k)}. \tag{5}$$

## 4 Convergence analysis

Before presenting our convergence analysis, we formalize a few standard assumptions, given in Sec. 4.1. Then, in Sec. 4.2, we provide our theoretical results. Proposition 4.1 and Theorem 4.1 establish our main results, in which Theorem 4.1 shows a linear convergence to a neighborhood of the globally optimal solution when employing a constant step size. We relegate the lemmas on which the main results are obtained to Appendix C.

### 4.1 Assumptions

Similar to Nedic & Ozdaglar (2009), to be able to construct doubly-stochastic mixing matrices, we make the following assumption.

**Assumption 4.1 (Simultaneous bidirectional communication)** *If device $i$ communicates with device $j$ at some iteration $k$, device $j$ also communicates with device $i$ at the same iteration.*

Next, we make the following assumptions on the local and global objective functions and quantify the data heterogeniety across the devices through the diversity of their gradients.

**Assumption 4.2 (Smoothness, strong convexity and gradient diversity)** *The local objective function at each device $i \in \mathcal{M}$, i.e., $F_i$, is (a) $\beta_i$-smooth and (b) $\mu_i$-strongly convex:*

*(a)* $\|\nabla F_i(\theta) - \nabla F_i(\theta')\| \leq \beta_i \|\theta - \theta'\| \leq \beta \|\theta - \theta'\|,$

*(b)* $\langle \nabla F_i(\theta) - \nabla F_i(\theta'), \theta - \theta' \rangle \geq \mu_i \|\theta - \theta'\|^2 \geq \mu \|\theta - \theta'\|^2,$

---

[2]$\mathbb{E}_X[\cdot]$ denotes the expectation operator conditioned on all the random variables except $X$, i.e., it averages out all the dependencies on $X$.

for all $(\theta', \theta) \in \mathbb{R}^n \times \mathbb{R}^n$, where $\mu = \min_{i \in \mathcal{M}} \mu_i$ and $\beta = \max_{i \in \mathcal{M}} \beta_i$. Also, the gradient diversity Lin et al. (2021) across the devices is measured via $\delta_i > 0$ and $\zeta_i \geq 0$ as follows:

(c) $\|\nabla F(\theta) - \nabla F_i(\theta)\| \leq \delta_i + \zeta_i \|\theta - \theta^\star\| \leq \delta + \zeta \|\theta - \theta^\star\|$,

for all $\theta \in \mathbb{R}^n$ and all $i \in \mathcal{M}$, in which $\delta = \max_{i \in \mathcal{M}} \delta_i$ and $\zeta = \max_{i \in \mathcal{M}} \zeta_i$.

Note that these measures are related to each other via the inequalities $\mu \leq \mu_i \leq \beta_i \leq \beta$, $0 \leq \zeta_i < \beta_i + \beta$ and $0 \leq \zeta \leq 2\beta$ (see Appendix D). In addition, note that we do not make the stricter assumption of $\zeta = 0$, and this makes the bounds we calculate more tighter, and causes our theoretical analysis to be more conclusive.

**Assumption 4.3 (Stochastic gradient noise)** *We make the following standard assumptions on the stochastic gradient noise $\epsilon_i^{(k)}$ for all $i \in \mathcal{M}$ and all $k \geq 0$:*

(a) *Stochastic gradient noise is zero mean and has a bounded variance:* $\mathbb{E}_{\epsilon_i^{(k)}}[\epsilon_i^{(k)}] = 0$, *and there exists a scalar $\sigma_i^2$ such that $\mathbb{E}[\|\epsilon_i^{(k)}\|_2^2] \leq \sigma_i^2 \leq \sigma^2$, where $\sigma^2 = \max_{i \in \mathcal{M}} \sigma_i^2$.*

(b) *Random vectors $\epsilon_{i_1}^{(k)}$ and $\epsilon_{i_2}^{(k)}$, indicator variables $v_{j_1}^{(k)}$ and $v_{j_2}^{(k)}$, and $\hat{v}_{l_1,q_1}^{(k)}$ and $\hat{v}_{l_2,q_2}^{(k)}$ are all mutually uncorrelated, for all $i_1 \neq i_2$, $j_1 \neq j_2$ and $(l_1, q_1) \neq (l_2, q_2)$.*

**Assumption 4.4** *The graph union of the underlying time-varying network graph, i.e., $\mathcal{G} = (\mathcal{M}, \cup_{k=0}^{\infty} \mathcal{E}^{(k)})$, is connected.*

Assumption 4.4 implies that if $\mathbf{P}^{(k)} = [p_{ij}^{(k)}]_{1 \leq i,j \leq m}$ and $\mathbf{R} = [r_{ij}]_{1 \leq i,j \leq m}$ as defined in equation 4 and equation 2 are the doubly-stochastic mixing matrices assigned to $\mathcal{G}^{(k)}$ and $\mathcal{G}$, respectively, we have

$$\left\|\mathbf{P}^{(k)}\mathbf{\Theta}^{(k)} - \mathbf{1}_m\bar{\theta}^{(k)}\right\|^2 \leq \left\|\mathbf{\Theta}^{(k)} - \mathbf{1}_m\bar{\theta}^{(k)}\right\|^2, \quad \left\|\mathbf{R}\mathbf{\Theta}^{(k)} - \mathbf{1}_m\bar{\theta}^{(k)}\right\|^2 \leq \rho_r^2 \cdot \left\|\mathbf{\Theta}^{(k)} - \mathbf{1}_m\bar{\theta}^{(k)}\right\|^2,$$

with $0 < \rho_r < 1$, where $\rho_r$ denotes the spectral radius of matrix $\mathbf{R} - \frac{1}{m}\mathbf{1}_m\mathbf{1}_m^T$.

### 4.2 MAIN RESULTS

As mentioned in Sec. 3.1, in order to find the optimal solution for equation 1, devices need to (i) reach consensus over the same solution, which will also (ii) be optimal with respect to the global cost function. We characterize the expected consensus error as $\mathbb{E}_{\mathbf{\Xi}^{(k)}}[\|\mathbf{\Theta}^{(k+1)} - \mathbf{1}_m\bar{\theta}^{(k+1)}\|^2]$, and the distance of the average model from the optimal solution as $\mathbb{E}_{\mathbf{\Xi}^{(k)}}[\|\bar{\theta}^{(k+1)} - \theta^\star\|^2]$. To characterize the convergence behavior of DSpodFL, we first provide an upper bound on the expected error in the average model at each iteration, i.e., $\mathbb{E}_{\mathbf{\Xi}^{(k)}}[\|\bar{\theta}^{(k+1)} - \theta^\star\|^2]$, in Lemma 4.1. Then, we also calculate an upper bound on the consensus error, i.e., $\mathbb{E}_{\mathbf{\Xi}^{(k)}}[\|\mathbf{\Theta}^{(k+1)} - \mathbf{1}_m\bar{\theta}^{(k+1)}\|^2]$, in Lemma 4.2.

To simplify the notations, we define $\mathbf{\Xi}^{(k)}$ as the collection of all random variables $v_i^{(r)}$, $\hat{v}_{ij}^{(r)}$ and $\epsilon_i^{(r)}$ for all $(i, j) \in \mathcal{M}^2$ and all iterations $0 \leq r \leq k$.

**Lemma 4.1 (Average model error)** *(See Appendix F for the proof.) For each iteration $k \geq 0$, we have the following bound on the expected average model error*

$\mathbb{E}_{\mathbf{\Xi}^{(k)}}[\|\bar{\theta}^{(k+1)} - \theta^\star\|^2] \leq \phi_{11}^{(k)}\mathbb{E}_{\mathbf{\Xi}^{(k-1)}}[\|\bar{\theta}^{(k)} - \theta^\star\|^2] + \phi_{12}^{(k)}\mathbb{E}_{\mathbf{\Xi}^{(k-1)}}[\|\mathbf{\Theta}^{(k)} - \mathbf{1}_m\bar{\theta}^{(k)}\|^2] + \psi_1^{(k)}$, where $\phi_{11}^{(k)} = 1 - \mu\alpha^{(k)}(1 + \mu\alpha^{(k)} - (\mu\alpha^{(k)})^2) + \frac{2\alpha^{(k)}}{\mu}(1 + \mu\alpha^{(k)})(1 - d_{\min}^{(k)})\beta^2$,

$\phi_{12}^{(k)} = (1 + \mu\alpha^{(k)})\frac{\alpha^{(k)}d_{\max}^{(k)}\beta^2}{m\mu}$, and $\psi_1^{(k)} = \frac{2\alpha^{(k)}}{\mu}(1 + \mu\alpha^{(k)})(1 - d_{\min}^{(k)})\delta^2 + \frac{(\alpha^{(k)})^2 d_{\max}^{(k)}\sigma^2}{m}$.

In Lemma 4.1, the upper bound on the expected error at iteration $k + 1$ is expressed in terms of the expected error $\mathbb{E}_{\mathbf{\Xi}^{(k-1)}}[\|\bar{\theta}^{(k)} - \theta^\star\|^2]$ at the previous iteration $k$, the consensus error (see Lemma 4.2) at iteration $k$, and the scalar $\psi_1^{(k)}$. Furthermore, the coefficients simplify when $d_{\min}^{(k)} = 1$,

which is essentially equivalent to the conventional DFL setup where devices perform SGDs at every iteration, i.e., $v_i^{(k)} = 1$ for all $i \in \mathcal{M}$ (see Lemma 5-b in Zehtabi et al. (2022b)).

We next bound the consensus error at each iteration, i.e., $\mathbb{E}_{\Xi^{(k)}}[\|\Theta^{(k+1)} - \mathbf{1}_m \bar{\theta}^{(k+1)}\|^2]$, which measures the deviation of ML model parameters of devices from the average ML model.

**Lemma 4.2 (Consensus error)** *(See Appendix I for the proof.) For each iteration $k \geq 0$, we have the following bound on the expected consensus error*

$$\mathbb{E}_{\Xi^{(k)}}[\|\Theta^{(k+1)} - \mathbf{1}_m \bar{\theta}^{(k+1)}\|^2] \leq \phi_{21}^{(k)} \mathbb{E}_{\Xi^{(k-1)}}[\|\bar{\theta}^{(k)} - \theta^\star\|^2] + \phi_{22}^{(k)} \mathbb{E}_{\Xi^{(k-1)}}[\|\Theta^{(k)} - \mathbf{1}_m \bar{\theta}^{(k)}\|^2] + \psi_2^{(k)},$$

*where $\phi_{21}^{(k)} = 3 \frac{1+\tilde{\rho}^{(k)}}{1-\tilde{\rho}^{(k)}} m d_{\max}^{(k)} (\alpha^{(k)})^2 (\zeta^2 + 2\beta^2 (1 - d_{min}^{(k)}))$,*
*$\phi_{22}^{(k)} = \frac{1+\tilde{\rho}^{(k)}}{2} + 3 \frac{1+\tilde{\rho}^{(k)}}{1-\tilde{\rho}^{(k)}} d_{\max}^{(k)} (\alpha^{(k)})^2 (\zeta^2 + 2\beta^2)$, and $\psi_2^{(k)} = m (\alpha^{(k)})^2 d_{\max}^{(k)} (3 \frac{1+\tilde{\rho}^{(k)}}{1-\tilde{\rho}^{(k)}} \delta^2 + \sigma^2)$.*

Note that $\tilde{\rho}$ is characterized as $\mathbb{E}_{\Xi^{(k)}}[\|\mathbf{P}^{(k)}\Theta^{(k)} - \mathbf{1}_m \bar{\theta}^{(k)}\|^2] \leq \tilde{\rho}^{(k)} \mathbb{E}_{\Xi^{(k-1)}}[\|\Theta^{(k)} - \mathbf{1}_m \bar{\theta}^{(k)}\|^2]$ (see Lemma C.4-(c)). In the conventional DFL setup, we will have (a) $\zeta = 0$ and (b) $d_{min}^{(k)} = 1$, resulting $d_i^{(k)} = 1$ for all $i \in \mathcal{M}$. This will result in $\phi_{21}^{(k)} = 0$.

Putting the results of Lemmas 4.1 and 4.2 together form the following linear system of inequalities:

$$\begin{bmatrix} \mathbb{E}_{\Xi^{(k)}}\left[\|\bar{\theta}^{(k+1)} - \theta^\star\|^2\right] \\ \mathbb{E}_{\Xi^{(k)}}\left[\|\Theta^{(k+1)} - \mathbf{1}_m \bar{\theta}^{(k+1)}\|^2\right] \end{bmatrix} \leq \mathbf{\Phi}^{(k)} \begin{bmatrix} \mathbb{E}_{\Xi^{(k-1)}}\left[\|\bar{\theta}^{(k)} - \theta^\star\|^2\right] \\ \mathbb{E}_{\Xi^{(k-1)}}\left[\|\Theta^{(k)} - \mathbf{1}_m \bar{\theta}^{(k)}\|^2\right] \end{bmatrix} + \mathbf{\Psi}^{(k)}, \quad (6)$$

with $\mathbf{\Phi}^{(k)} = [\phi_{ij}^{(k)}]_{1 \leq i,j \leq 2}$ and $\mathbf{\Psi}^{(k)} = [\psi_1^{(k)} \quad \psi_2^{(k)}]^T$. Recursively expanding the inequalities in equation 6 gives us an explicit relationship between the expected model error and consensus error at each iteration and their initial values:

$$\begin{bmatrix} \mathbb{E}_{\Xi^{(k)}}\left[\|\bar{\theta}^{(k+1)} - \theta^\star\|^2\right] \\ \mathbb{E}_{\Xi^{(k)}}\left[\|\Theta^{(k+1)} - \mathbf{1}_m \bar{\theta}^{(k+1)}\|^2\right] \end{bmatrix} \leq \mathbf{\Phi}^{(k:0)} \begin{bmatrix} \|\bar{\theta}^{(0)} - \theta^\star\|^2 \\ \|\Theta^{(0)} - \mathbf{1}_m \bar{\theta}^{(0)}\|^2 \end{bmatrix} + \sum_{r=1}^{k} \mathbf{\Phi}^{(k:r)} \mathbf{\Psi}^{(r-1)} + \mathbf{\Psi}^{(k)},$$
$$(7)$$

where we have defined $\mathbf{\Phi}^{(k:s)} = \mathbf{\Phi}^{(k)} \mathbf{\Phi}^{(k-1)} \cdots \mathbf{\Phi}^{(s)}$ for $k > s$, and $\mathbf{\Phi}^{(k:k)} = \mathbf{\Phi}^{(k)}$.

In order for us formalize the convergence bound of `DSpodFL`, we first have to show that the error vector in equation 6 decreases at each iteration. In other words, we have to show that the spectral radius of matrix $\mathbf{\Phi}^{(k)}$ is less than one, i.e., $\rho(\mathbf{\Phi}^{(k)}) < 1$.

**Proposition 4.1** *(See Appendix J for the proof.) If the step size satisfies the following condition for all $k \geq 0$*

$$\alpha^{(k)} < \min \left\{ \frac{1}{\mu}, \frac{1}{2\sqrt{3d_{max}^{(k)}}} \frac{1-\tilde{\rho}^{(k)}}{\sqrt{1+\tilde{\rho}^{(k)}}} \frac{1}{\sqrt{\zeta^2 + 2\beta^2}}, \left( \frac{\mu}{12 \left( \zeta^2 + 2\beta^2 \left( 1 - d_{min}^{(k)} \right) \right)} \right)^{1/3} \left( \frac{1-\tilde{\rho}^{(k)}}{2d_{max}^{(k)} \beta} \right)^{2/3} \right\},$$

*then we have $\rho(\mathbf{\Phi}^{(k)}) < 1$ for all $k \geq 0$, in which $\rho(\cdot)$ denotes the spectral radius of a given matrix, and $\mathbf{\Phi}^{(k)}$ is given in the linear system of inequalities of equation 6. $\rho(\mathbf{\Phi}^{(k)})$ follows as*

$$\rho(\mathbf{\Phi}^{(k)}) = \frac{3+\tilde{\rho}^{(k)}}{4} - A\alpha^{(k)} + B(\alpha^{(k)})^2 + \frac{1}{2}\sqrt{\left(\frac{1-\tilde{\rho}^{(k)}}{2} - 2(A\alpha^{(k)} + B(\alpha^{(k)})^2)\right)^2 + C(\alpha^{(k)})^3},$$

*where $A = \frac{1}{\mu}(1 + \Gamma_1^{(k)})(\Gamma_2^{(k)} - 1)(1 - d_{min}^{(k)})\beta^2$, $B = \frac{3}{2}\frac{1+\tilde{\rho}^{(k)}}{1-\tilde{\rho}^{(k)}} d_{\max}^{(k)}(\zeta^2 + 2\beta^2)$ and $C = 12\frac{(1+\Gamma_1^{(k)})\beta^2}{\mu}\frac{1+\tilde{\rho}^{(k)}}{1-\tilde{\rho}^{(k)}}(d_{\max}^{(k)})^2(\zeta^2 + 2\beta^2(1 - d_{min}^{(k)}))$.*

Proposition 4.1 implies that $\lim_{k \to \infty} \mathbf{\Phi}^{(k:0)} = 0$ in equation 7. However, this is only the asymptotic behaviour of $\mathbf{\Phi}^{(k:0)}$, and the exact convergence rate will depend on the choice of the step size $\alpha^{(k)}$. Furthermore, since the first expression in equation 7 asymptotically approaches zero, Proposition 4.1

also implies that the non-negative optimality gap is determined by the terms $\sum_{r=1}^{k} \mathbf{\Phi}^{(k:r)} \mathbf{\Psi}^{(r-1)} + \mathbf{\Psi}^{(k)}$, and it can be either zero or a positive value depending on the choice of $\alpha^{(k)}$ and $d_{min}^{(k)}$.

Proposition 4.1 outlines the necessary constraint on the step size $\alpha^{(k)}$ at each iteration $k \geq 0$. We next provide a corollary to Proposition 4.1, in which we show that under certain conditions, the above-mentioned constraint needs to be satisfied only on the initial value of the step size, i.e, $\alpha^{(0)}$.

**Corollary 4.1** *If the step size $\alpha^{(k)}$ is non-increasing, the probability of SGDs $d_i^{(k)}$ are constant, i.e., $\alpha^{(k+1)} \leq \alpha^{(k)}$, $d_i^{(k)} = d_i$ for all $k \geq 0$, and we have $\tilde{\rho}^{(k)} \leq \tilde{\rho} = \sup_{k=0,1,...} \tilde{\rho}^{(k)}$ for the spectral radius, then the constraints in Proposition 4.1 simplify to*

$$\alpha^{(0)} < \min \left\{ \frac{1}{\mu}, \frac{1}{2\sqrt{3}d_{max}} \frac{1-\tilde{\rho}}{\sqrt{1+\tilde{\rho}}} \frac{1}{\sqrt{\zeta^2+2\beta^2}}, \left( \frac{\mu}{12(\zeta^2+2\beta^2(1-d_{min}))} \right)^{1/3} \left( \frac{1-\tilde{\rho}}{2d_{max}\beta} \right)^{2/3} \right\},$$

*where $d_{max} = \max_{i \in \mathcal{M}} d_i$ and $d_{min} = \min_{i \in \mathcal{M}} d_i$.*

In the above Corollary, we obtained the constraints on the initial value of the step size, i.e., $\alpha^{(0)}$, that lead to the spectral radius of $\mathbf{\Phi}^{(k)}$ being less than 1, i.e., $\rho(\mathbf{\Phi}^{(k)}) < 1$. Using this result, we characterize the short-term behavior and also derive non-asymptotic convergence guarantees on `DSpodFL` in the following theorem.

**Theorem 4.1 (Convergence guarantees)** *(See Appendix K for the proof.) If a constant step size $\alpha^{(k)} = \alpha$ with $\alpha > 0$ satisfying the conditions outlined in Corollary 4.1 is employed, and the probability of SGDs and probability of aggregations are set to constant values, i.e., $d_i^{(k)} = d_i$ for all $i \in \mathcal{M}$ and $b_{ij}^{(k)} = b_{ij}$ for all $(i,j) \in \mathcal{M}^2$, respectively, then we can rewrite equation 7 as*

$$\begin{bmatrix} \mathbb{E}_{\boldsymbol{\Xi}^{(k)}} \left[ \left\| \bar{\theta}^{(k+1)} - \theta^\star \right\|^2 \right] \\ \mathbb{E}_{\boldsymbol{\Xi}^{(k)}} \left[ \left\| \mathbf{\Theta}^{(k+1)} - \mathbf{1}_m \bar{\theta}^{(k+1)} \right\|^2 \right] \end{bmatrix} \leq \rho(\mathbf{\Phi})^{k+1} \begin{bmatrix} \left\| \bar{\theta}^{(0)} - \theta^\star \right\|^2 \\ \left\| \mathbf{\Theta}^{(0)} - \mathbf{1}_m \bar{\theta}^{(0)} \right\|^2 \end{bmatrix} + \frac{1}{1-\rho(\mathbf{\Phi})} \mathbf{\Psi}, \quad (8)$$

*in which the entries of $\mathbf{\Phi}^{(k)} = \mathbf{\Phi}$ and $\mathbf{\Psi}^{(k)} = \mathbf{\Psi}$ are given in Lemmas 4.1 and 4.2, and they are time-invariant in this setup as all their parameters are constants. Letting $k \to \infty$, we get*

$$\limsup_{k \to \infty} \begin{bmatrix} \mathbb{E}_{\boldsymbol{\Xi}^{(k)}} \left[ \left\| \bar{\theta}^{(k+1)} - \theta^\star \right\|^2 \right] \\ \mathbb{E}_{\boldsymbol{\Xi}^{(k)}} \left[ \left\| \mathbf{\Theta}^{(k+1)} - \mathbf{1}_m \bar{\theta}^{(k+1)} \right\|^2 \right] \end{bmatrix} = \frac{\alpha}{1-\rho(\mathbf{\Phi})} \begin{bmatrix} \frac{2}{\mu}(1+\mu\alpha)(1-d_{min})\delta^2 + \frac{\alpha d_{max}\sigma^2}{m} \\ m\alpha d_{max} \left( 3\frac{1+\tilde{\rho}}{1-\tilde{\rho}}\delta^2 + \sigma^2 \right) \end{bmatrix},$$
$$(9)$$

*where $d_{max} = \max_{i \in \mathcal{M}} d_i$ and $d_{min} = \min_{i \in \mathcal{M}} d_i$. Note that Proposition 4.1 ensures $\rho(\mathbf{\Phi}) < 1$.*

The bound in equation 8 of Theorem 4.1 indicates that by using a constant step size, DSpodFL achieves a geometric convergence rate, and equation 9 characterizes the asymptotic optimality gap as $k \to \infty$. Furthermore, we observe that this optimality gap is proportional to the step size $\alpha$, but also depends on the connectivity of the communication graph, number of devices in the system, frequency of computations, frequency of communications and the dataset. Consequently, by choosing a smaller step size, we can linearly reduce this gap.

Note that equation 8 and equation 9 are derived for the consensus error and the average model error themselves. This is an improvement over previous work where bounds are provided only on cesaro sums of these error terms, e.g., Koloskova et al. (2020); Lian et al. (2017); Sundhar Ram et al. (2010).

## 5 NUMERICAL EVALUATION

### 5.1 EXPERIMENTAL SETUP

To evaluate our methodology, we consider an image classification task using the Fashion-MNIST dataset (Xiao et al., 2017). We consider two models: linear SVM (Support Vector Machine), which is convex and satisfies Assumption 4.2-(b), and a 5-layer CNN (Convolutional Neural Network), which is non-convex. We consider a network of 10 devices ($m = 10$), connected to each other in a decentralized manner via a random geometric graph with radius $0.4$ (Penrose, 2003). We set the

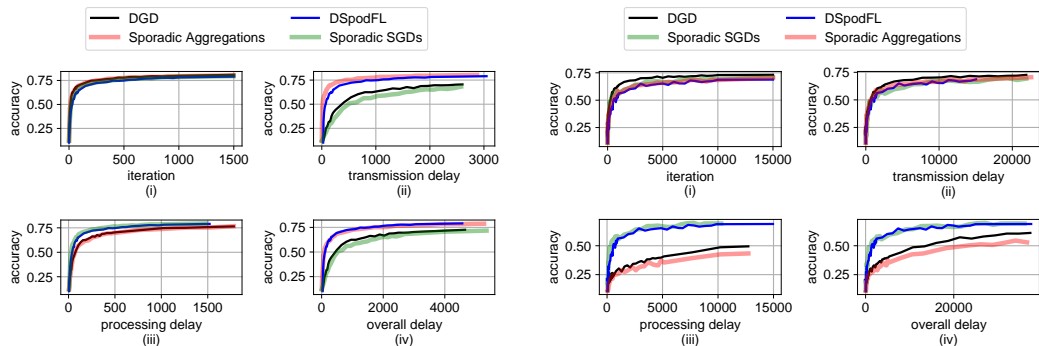

*(a) IID data split, with each device having data points from all of the 10 classes in the dataset.*

*(b) Non-IID data split, with each device having data points from only 1 of the classes in the dataset.*

Figure 1: *Results for an SVM model on the Fashion-MNIST image classification dataset. Overall, we see that `DSpodFL` matches the highest performing baselines in terms of overall delay, emphasizing the benefit of sporadicity in DFL for SGD iterations and model aggregations simultaneously.*

learning rate for all devices to a constant value $\alpha = 0.01$ to satisfy the conditions outlined in our theoretical analysis (see Theorem 4.1), and use a batch size of 16. The probability of SGD for each device is set to a constant value and is randomly chosen from the interval $[0, 0.1]$ for the SVM model and $[0, 1]$ for the CNN model, i.e., $d_i \sim \mathcal{U}[0, 0.1]$ and $d_i \sim \mathcal{U}[0, 1]$ for all $i \in \mathcal{M}$, respectively.

We compare our method, `DSpodFL`, with three baselines: (a) Distributed Gradient Descent (DGD), which is the conventional DFL algorithm, where SGDs and local aggregations occur at every iteration (Nedic & Ozdaglar, 2009); (b) the Randomized Gossip algorithm Koloskova et al. (2020), which we denote as Sporadic Aggregations (with constant SGDs); and (c) Sporadic SGDs (with constant aggregations). Note that all of these baselines can be viewed as special cases of the `DSpodFL` method, i.e., $d_i = 1$ in (a) and (b) for all $i \in \mathcal{M}$, and $b_{ij} = 1$ in (a) and (c) for all $(i, j) \in \mathcal{M}^2$.

We provide the results for two different data split scenarios common in the FL literature: IID and non-IID. In the IID case, each device gets samples coming from all of the 10 classes in the data set, while in the non-IID case, each device gets samples belonging to just 1 class. The probability of aggregations for each link is randomly chosen as $b_{ij} \sim \mathcal{U}[0, 1]$ for all $(i, j) \in \mathcal{M}$.

In Appendix P, we present additional results based on varying graph connectivity, data distributions, and communication/computation capabilities.

## 5.2 RESULTS AND DISCUSSION

Our results are shown in Figs. 1 & 2. The metrics we use for performance comparisons are testing accuracy achieved over different units: (i) training iterations, (ii) average transmission delay incurred across links, i.e., $\tau_{trans} = [\sum_{i=1}^{m} (1/|\mathcal{N}_i|) \sum_{j \in \mathcal{N}_i} 1/b_{ij}] / [\sum_{i=1}^{m} (1/|\mathcal{N}_i|) \sum_{j \in \mathcal{N}_i} \hat{v}_{ij}^{(k)}/b_{ij}]$, (iii) average processing delay incurred across devices, i.e., $\tau_{proc} = [\sum_{i=1}^{m} v_i^{(k)}/d_i] / [\sum_{i=1}^{m} 1/d_i]$, and (iv) average total delay incurred, i.e., $\tau_{total} = \tau_{trans} + \tau_{proc}$. Specifically, due to resource heterogeneity in DFL, having a device waiting for all other devices to complete their computations and communications to start the next local SGD iteration is prohibitively time-consuming. This motivates us to plot the accuracy reached per time unit of delay.

**Impact of sporadic aggregations.** In the IID case, since all devices have access to the distribution of the entire dataset, model aggregations are not as critical to DFL training performance. This can be observed in Figs. 1a-(ii) for SVM and 2a-(ii) for CNN, in which sporadic aggregations enable convergence within a lower average transmission time. In the non-IID case, by contrast, model aggregations are crucial to propagating global information, which is consistent with sporadic aggregations performing poorly in Fig. 2b-(ii) for CNN. In this case, the sporadic SGD component of `DSprodFL` prevents the models from getting stuck in local minima.

**Impact of sporadic SGDs.** Figs. 1a-(iii) and 1b-(iii) illustrate that sporadic SGDs in an SVM model enable the system to achieve a higher accuracy with lower average processing time. The

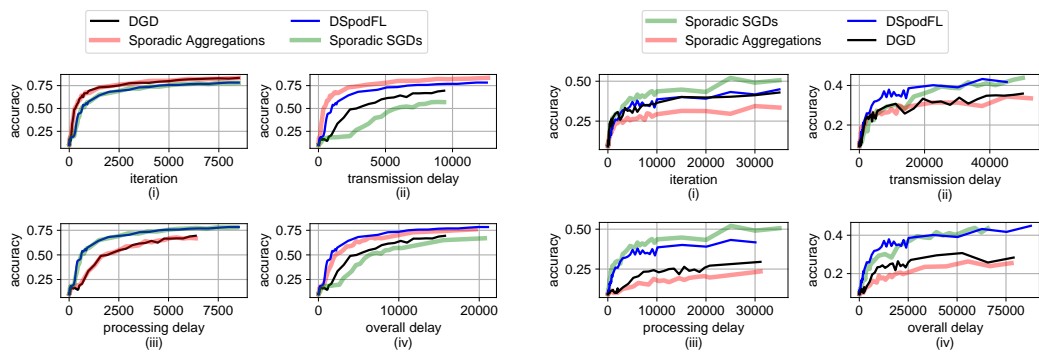

*(a) IID data split, with each device having data points from all of the 10 classes in the dataset.*

*(b) Non-IID data split, with each device having data points from only 1 of the classes in the dataset.*

Figure 2: Results of employing a CNN model on the Fashion-MNIST image classification dataset. We see that `DSpodFL` surpasses the baselines in terms of overall delay in this case of a non-convex model which violates Assumption 4.2-(b).

same holds for a CNN model as shown in Figs. 2a-(iii) and 2b-(iii). We observe that the non-IID case benefits more from sporadic SGDs in terms of processing delay as compared to the IID case. This is consistent with aggregations being more crucial in the non-IID case in order to prevent the local models of devices from drifting too far from each other, thereby allowing local SGDs to be more sporadic without compromising accuracy. Thus, the non-IID case benefits more from sporadic SGDs rather than sporadic aggregations.

**Comparison in terms of overall delay.** Figs. 1a-(iv), 1b-(iv), 2a-(iv), and 2b-(iv) contain the most important comparisons for `DSpodFL`, as they combine processing delay and transmission delay. We can see that, since `DSpodFL` method employs both sporadic SGDs and aggregations, it outperforms all baselines, for both data distributions and models. Consistent with the above discussion, we observe that in the IID cases, the sporadic aggregations component of our method is a larger driver of the total resource efficiency, while in non-IID case, it is the sporadic SGDs component. In Appendix P.2, we conduct further analysis on this through varying the label distributions.

**Comparison in terms of training iterations.** The DGD method outperforms other baselines in terms of the testing accuracy per iteration in Figs. 1a-(i) and 1b-(i) for the SVM case, and in Fig. 2a-(i) for the CNN case. This is due to DGD not taking into account the limited available resources at the devices, which leads to its low resource efficiency in the delay plots. Interestingly however, Fig. 2b-(i) shows an improvement from employing sporadic SGDs when a CNN model is used for the non-IID case, implying that sporadicity in SGDs is preventing the model from getting stuck in local minima.

## 6 CONCLUSION

We proposed a novel resource-efficient methodology for DFL, in which (i) stochastic gradient computations and (ii) device-to-device communications do not need to occur at every iteration. Contrarily, each device in our fully-decentralized system decides on (i) when to conduct SGDs based on its computational resource availability, and (ii) when to partake in inter-device communications based on its communication resource availability. We theoretically analyzed the convergence behavior of `DSpodFL`, and showed that when using a constant step size, a geometric convergence rate to a finite neighbourhood of the optimal solution can be achieved. We further illustrated that the optimality gap is proportional to the step size. Finally, through several experiments, we demonstrated that our method achieves significant delay savings compared to DFL baselines.

Our work also gives rise to interesting future directions. It would be instructive to introduce sporadicity in the gradient computations involved in decentralized gradient tracking methods, and subsequently assess the performance of some known algorithms that fall into this category. Another insightful offshoot of our methodology would be to investigate the effects of introducing temporal correlations in the patterns of sporadicity in both SGDs and local model aggregations.

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
