## A  NOTATION

Arguments for functions are denoted with parentheses, e.g., $f(x)$ implies $x$ is an argument for function $f$. The iteration index for a parameter is indicated via superscripts, e.g., $h^{(k)}$ is the value of the parameter $h$ at iteration $k$. Device indices are given via subscripts, e.g., $h_i^{(k)}$ refers to parameter belonging to device $i$. We write a graph $\mathcal{G}$ with a set of nodes (devices) $\mathcal{V}$ and a set of edges (links) $\mathcal{E}$ as $\mathcal{G} = (\mathcal{V}, \mathcal{E})$.

We denote vectors with lowercase boldface, e.g., $\mathbf{x}$, and matrices with uppercase boldface, e.g., $\mathbf{X}$. All vectors $\mathbf{x} \in \mathbb{R}^{d \times 1}$ are column vectors, except in certain cases where average vectors $\bar{\mathbf{x}} \in \mathbb{R}^{1 \times d}$ and optimal vectors $\mathbf{w}^\star \in \mathbb{R}^{1 \times d}$ are row vectors. $\langle \mathbf{x}, \mathbf{x}' \rangle$ and $\langle \mathbf{X}, \mathbf{X}' \rangle$ denote the inner product of two vectors $\mathbf{x}, \mathbf{x}'$ of equal dimensions and the Frobenius inner product of two matrices $\mathbf{X}, \mathbf{X}'$ of equal dimensions, respectively. Moreover, $\|\mathbf{x}\|$ and $\|\mathbf{X}\|$ denote the 2-norm of the vector $\mathbf{x}$, and the Frobenius norm of the matrix $\mathbf{X}$, respectively. The spectral norm of the matrix $\mathbf{X}$ is written as $\rho(\mathbf{X})$.

## B  ALGORITHM PSEUDOCODE

---
**Algorithm 1:** Decentralized Sporadic Federated Learning (DSpodFL)

---
**Input:** K, $\{\mathcal{G}^{(k)} = (\mathcal{M}, \mathcal{E}^{(k)})\}_{0 \leq k \leq K}$, $\{d_i^{(k)}\}_{i \in \mathcal{M}, 0 \leq k \leq K}$, $\{b_{ij}^{(k)}\}_{(i,j) \in \mathcal{E}^{(k)}, 0 \leq k \leq K}$,
$\qquad \{\alpha^{(k)}\}_{0 \leq k \leq K}$

**Output:** $\{\theta_i^{(K+1)}\}_{i \in \mathcal{M}}$

$k \leftarrow 0$, Initialize $\theta^{(0)}$, $\{\theta_i^{(0)} \leftarrow \theta^{(0)}\}_{i \in \mathcal{M}}$, $\{v_i^{(0)} \leftarrow 1\}_{i \in \mathcal{M}}$, $\{\hat{v}_{ij}^{(0)} \leftarrow 1\}_{(i,j) \in \mathcal{E}^{(0)}}$

**while** $k \leq K$ **do**
    **forall** $i \in \mathcal{M}$ **do**
        $g_i^{(k)} \leftarrow 0$, $\text{aggr}_i^{(k)} \leftarrow 0$
        **if** $v_i^{(k)} = 1$ **then**
            sample mini-batch $\xi_i^{(k)} \in \mathcal{D}_i$
            $g_i^{(k)} \leftarrow \nabla F_i(\theta_i^{(k)}; \xi_i^{(k)})$
        **end**
        **forall** $j \in \mathcal{E}_i^{(k)}$ **do**
            **if** $\hat{v}_{ij}^{(k)} = 1$ **then**
                $r_{ij} \leftarrow 1/(1 + \max\{|\mathcal{N}_i^{(k)}|, |\mathcal{N}_j^{(k)}|\})$
                $\text{aggr}_i^{(k)} \leftarrow \text{aggr}_i^{(k)} + r_{ij}\left(\theta_j^{(k)} - \theta_i^{(k)}\right)$
            **end**
        **end**
    **end**
    **forall** $i \in \mathcal{M}$ **do**
        $\theta_i^{(k+1)} \leftarrow \theta_i^{(k)} + \text{aggr}_i^{(k)} - \alpha^{(k)}\mathbf{g}_i^{(k)}$
        $v_i^{(k+1)} \leftarrow 0$
        **if** $random()\leq d_i^{(k)}$ **then**
            $v_i^{(k+1)} \leftarrow 1$
        **end**
        **forall** $j \in \mathcal{E}_i^{(k)}$ **do**
            $\hat{v}_{ij}^{(k+1)} \leftarrow 0$ **if** $random()\leq b_{ij}^{(k)}$ **then**
                $\hat{v}_{ij}^{(k+1)} \leftarrow 1$
            **end**
        **end**
    **end**
    $k \leftarrow k + 1$
**end**

---

## C  INTERMEDIARY LEMMAS

**Lemma C.1** *(See Appendix D for the proof.) Let Assumption 4.2 hold. We have*

*(a) The global objective function $F(\theta)$ is $\beta$-smooth and $\mu$-strongly convex, i.e.,*

$$\|\nabla F(\theta) - \nabla F(\theta')\| \leq \beta \|\theta - \theta'\|, \qquad \langle \nabla F(\theta) - \nabla F(\theta'), \theta - \theta' \rangle \geq \mu \|\theta - \theta'\|^2.$$

*(b) The gradients of the global and local objective functions, and the gradient of the local objective function at the optimal point are bounded as*

$$\|\nabla F(\theta)\| \leq \beta \|\theta - \theta^\star\|, \quad \|\nabla F_i(\theta)\|^2 \leq 2 \left( \beta_i^2 \|\theta - \theta^\star\|^2 + \delta_i^2 \right), \quad \|\nabla F_i(\theta^\star)\| \leq \delta_i.$$

Part-(a) of Lemma C.1 outlines the smoothness and convexity behaviour of the global objective function based on the measures of local objectives, and part (b) provides upper bounds on the gradients. Note how these show that we are not making the bounded gradients assumption for all $\theta \in \mathbb{R}^n$, but only bounded local gradients at the globally optimal point $\theta^\star$.

Next, we provide upper bounds on the expected Frobenius norms of the following quantities related to SGD noises.

**Lemma C.2** *(See Appendix E for the proof.) For every iteration $k \geq 0$, the average SGD noise and their deviation from this average can be bounded as*

$$\mathbb{E}_{\xi^{(k)}} \left[ \left\| \overline{\epsilon v}^{(k)} \right\|^2 \right] \leq d_{max}^{(k)} \sigma^2 / m, \qquad \mathbb{E}_{\xi^{(k)}} \left[ \left\| \mathbf{V}^{(k)} \mathbf{E}^{(k)} - \mathbf{1}_m \overline{\epsilon v}^{(k)} \right\|^2 \right] \leq m d_{max}^{(k)} \sigma^2,$$

*in which $\overline{\epsilon v}^{(k)} = \frac{1}{m} \sum_{i=1}^m \epsilon_i^{(k)} v_i^{(k)}$.*

Note that by setting $d_{max}^{(k)} = 1$ in C.2, we get back the well-known estimation bounds for these quantities (e.g., see Lemma 2 in Pu & Nedić (2021)).

Next, we find an upper bound on the expected deviation of the gradients from their average (similar to the second quantity in Lemma C.2).

**Lemma C.3** *(See Appendix G for the proof.) For each iteration $k \geq 0$, we have the following bound on the expected error of gradients from their average*

$$\mathbb{E}_{\mathbf{\Xi}^{(k)}} \left[ \left\| \mathbf{V}^{(k)} \nabla^{(k)} - \mathbf{1}_m \overline{\nabla v}^{(k)} \right\|^2 \right] \leq 3 d_{max}^{(k)} \left[ m\delta^2 + \left( \zeta^2 + 2\beta^2 \right) \mathbb{E}_{\mathbf{\Xi}^{(k-1)}} \left[ \left\| \mathbf{\Theta}^{(k)} - \mathbf{1}_m \bar{\theta}^{(k)} \right\|^2 \right] \right.$$

$$\left. + m \left( \zeta^2 + 2\beta^2 \left( 1 - d_{min}^{(k)} \right) \right) \mathbb{E}_{\mathbf{\Xi}^{(k-1)}} \left[ \left\| \bar{\theta}^{(k)} - \theta^\star \right\|^2 \right] \right]$$

*in which $\nabla^{(k)}$ is a matrix whose rows are comprised of the gradient vectors $\nabla F_i(\theta_i^{(k)})$, and $\overline{\nabla v}^{(k)} = \frac{1}{m} \sum_{i=1}^m \nabla F_i(\theta_i^{(k)}) v_i^{(k)}$.*

Finally, we analyze the behaviour of the random mixing matrix $\mathbf{P}^{(k)}$ defined in equation 3 and equation 4.

**Lemma C.4** *(See Appendix H for the proof.) For each iteration $k \geq 0$, we have*

*(a) The expected mixing matrix denoted as $\bar{\mathbf{R}}^{(k)}$, is irreducible and doubly-stochastic:*

$$\mathbb{E}_{\hat{\mathbf{V}}^{(k)}} \left[ \mathbf{P}^{(k)} \right] \triangleq \bar{\mathbf{R}}^{(k)} = \left[ \bar{r}_{ij}^{(k)} \right]_{1 \leq i,j \leq m}, \qquad \bar{r}_{ij}^{(k)} = \begin{cases} b_{ij}^{(k)} r_{ij} & i \neq j \\ 1 - \sum_{j=1}^m b_{ij}^{(k)} r_{ij} & i = j \end{cases}.$$

(b) $\mathbb{E}_{\hat{\mathbf{V}}^{(k)}}\left[\left(\mathbf{P}^{(k)}\right)^2\right] = \left(\bar{\mathbf{R}}^{(k)}\right)^2 + \mathbf{R}_0^{(k)} \triangleq \tilde{\mathbf{R}}^{(k)},$

where $\mathbf{R}_0^{(k)}$ is a matrix whose rows and columns sum to zero. Thus, $\tilde{\mathbf{R}}^{(k)}$ will be irreducible and doubly-stochastic.

(c) $\mathbb{E}_{\mathbf{\Xi}^{(k)}}\left[\left\|\mathbf{P}^{(k)}\mathbf{\Theta}^{(k)} - \mathbf{1}_m\bar{\theta}^{(k)}\right\|^2\right] \leq \tilde{\rho}^{(k)}\mathbb{E}_{\mathbf{\Xi}^{(k-1)}}\left[\left\|\mathbf{\Theta}^{(k)} - \mathbf{1}_m\bar{\theta}^{(k)}\right\|^2\right],$

in which $\tilde{\rho}^{(k)}$ is the spectral radius of the matrix $\tilde{\mathbf{R}}^{(k)} - \frac{1}{m}\mathbf{1}_m\mathbf{1}_m^T$.

# D    PROOF OF LEMMA C.1

(a) First, we use the smoothness property given in Assumption 4.2-(a) to get

$$\|\nabla F(\theta) - \nabla F(\theta')\| = \left\|\frac{1}{m}\sum_{j=1}^m (\nabla F_j(\theta) - \nabla F_j(\theta'))\right\| \leq \frac{1}{m}\sum_{j=1}^m \|\nabla F_j(\theta) - \nabla F_j(\theta')\|$$

$$\leq \frac{1}{m}\sum_{j=1}^m \beta_j \|\theta - \theta'\| = \bar{\beta}\|\theta - \theta'\| \leq \beta\|\theta - \theta'\|$$

Next, using the strong convexity property of Assumption 4.2-(b), we have

$$\langle\nabla F(\theta) - \nabla F(\theta'), \theta - \theta'\rangle = \frac{1}{m}\sum_{j=1}^m \langle\nabla F_j(\theta) - \nabla F_j(\theta'), \theta - \theta'\rangle$$

$$\geq \frac{1}{m}\sum_{j=1}^m \mu_j\|\theta - \theta'\|^2 = \bar{\mu}\|\theta - \theta'\|^2 \geq \mu\|\theta - \theta'\|^2$$

(b) Since $\nabla F(\theta^\star) = 0$ by definition, we can use the results of part (a) of this lemma to show that

$$\|\nabla F(\theta)\| \leq \beta\|\theta - \theta^\star\|.$$

Once again noting that $\nabla F(\theta^\star) = 0$, we next use the gradient diversity bound outlined in Assumption 4.2-(c) to get

$$\|\nabla F_i(\theta^\star)\| \leq \delta_i. \tag{10}$$

Finally, using equation 10 and Assumption 4.2-(a), we write

$$\|\nabla F_i(\theta)\|^2 \leq 2\left(\|\nabla F_i(\theta) - \nabla F_i(\theta^\star)\|^2 + \|\nabla F_i(\theta^\star)\|^2\right) \leq 2\left(\beta_i^2\|\theta - \theta^\star\|^2 + \delta_i^2\right),$$

finishing the proof.

To explain the statement written after Assumption 4.2 on how these measures relate to each other, we first have

$$\mu \leq \mu_i \leq \beta_i \leq \beta,$$

in which $\mu_i \leq \beta_i$ is a well-known fact (see Bottou et al. (2018) as a reference), and $\mu \leq \mu_i$ and $\beta_i \leq \beta$ follow from the definitions given in Assumption 4.2. Moreover, if we upper-bound the gradient diversity term $\|\nabla F(\theta) - \nabla F_i(\theta)\|$ without using Assumption 4.2-(c), we will have

$$\|\nabla F(\theta) - \nabla F_i(\theta)\| \leq \|\nabla F(\theta) - \nabla F_i(\theta^\star) + \nabla F_i(\theta^\star) - \nabla F_i(\theta)\|$$
$$\leq \|\nabla F(\theta)\| + \|\nabla F_i(\theta^\star)\| + \|\nabla F_i(\theta) - \nabla F_i(\theta^\star)\| \tag{11}$$
$$\leq \delta_i + \beta_i\|\theta - \theta^\star\| + \beta\|\theta - \theta^\star\| \leq \delta_i + (\beta_i + \beta)\|\theta - \theta^\star\|,$$

in which we used Assumption 4.2-(a) and the results of Lemma C.1-(b). Now comparing equation 11 with the assumption made in 4.2-(c) for the same expression, we conclude that

$$\zeta_i \leq \beta_i + \beta, \qquad \zeta \leq 2\beta.$$

# E  PROOF OF LEMMA C.2

We start by finding an upper bound for the average SGD noise, by expanding the terms using their definitions, and employing the properties given in Assumption 4.3.

$$
\mathbb{E}_{\xi^{(k)}}\left[\left\|\overline{\epsilon v}^{(k)}\right\|^2\right] = \mathbb{E}_{\xi^{(k)}}\left[\left\|\frac{1}{m}\sum_{i=1}^{m}\epsilon_i^{(k)}v_i^{(k)}\right\|^2\right] = \mathbb{E}_{\xi^{(k)}}\left[\frac{1}{m^2}\sum_{i=1}^{m}\sum_{j=1}^{m}\left\langle\epsilon_i^{(k)}v_i^{(k)},\epsilon_j^{(k)}v_j^{(k)}\right\rangle\right]
$$

$$
= \frac{1}{m^2}\sum_{i=1}^{m}\mathbb{E}_{\xi_i^{(k)}}\left[\left\|\epsilon_i^{(k)}v_i^{(k)}\right\|^2\right] + \frac{1}{m^2}\sum_{i=1}^{m}\sum_{\substack{j=1\\j\neq i}}^{m}\left\langle\mathbb{E}_{\xi_i^{(k)}}\left[\epsilon_i^{(k)}v_i^{(k)}\right],\mathbb{E}_{\xi_j^{(k)}}\left[\epsilon_j^{(k)}v_j^{(k)}\right]\right\rangle
$$

$$
= \frac{1}{m^2}\sum_{i=1}^{m}\mathbb{E}_{\xi_i^{(k)}}\left[\left\|\epsilon_i^{(k)}\right\|^2 v_i^{(k)}\right] + \frac{1}{m^2}\sum_{i=1}^{m}\sum_{\substack{j=1\\j\neq i}}^{m}\left\langle\mathbb{E}_{\epsilon_i^{(k)}}\left[\epsilon_i^{(k)}\right]\mathbb{E}_{v_i^{(k)}}\left[v_i^{(k)}\right],\right.
$$

$$
\left.\mathbb{E}_{\epsilon_j^{(k)}}\left[\epsilon_j^{(k)}\right]\mathbb{E}_{v_j^{(k)}}\left[v_j^{(k)}\right]\right\rangle
$$

$$
= \frac{1}{m^2}\sum_{i=1}^{m}\mathbb{E}_{\epsilon_i^{(k)}}\left[\left\|\epsilon_i^{(k)}\right\|^2\right]\mathbb{E}_{v_i^{(k)}}\left[v_i^{(k)}\right] = \frac{1}{m^2}\sum_{i=1}^{m}d_i^{(k)}\sigma_i^2 \leq \frac{d_{max}^{(k)}\sigma^2}{m}.
$$

Next, we found an upper bound for deviance of the error matrix from its average, using a similar approach as above. We have

$$
\mathbb{E}_{\xi^{(k)}}\left[\left\|\mathbf{V}^{(k)}\mathbf{E}^{(k)} - \mathbf{1}_m\overline{\epsilon v}^{(k)}\right\|^2\right]
$$

$$
= \mathbb{E}_{\xi^{(k)}}\left[\left\|\mathbf{V}^{(k)}\mathbf{E}^{(k)}\right\|^2 - 2\left\langle\mathbf{V}^{(k)}\mathbf{E}^{(k)},\mathbf{1}_m\overline{\epsilon v}^{(k)}\right\rangle + \left\|\mathbf{1}_m\overline{\epsilon v}^{(k)}\right\|^2\right]
$$

$$
= \mathbb{E}_{\xi^{(k)}}\left[\sum_{i=1}^{m}\left\|\epsilon_i^{(k)}v_i^{(k)}\right\|^2\right] - 2\mathbb{E}_{\xi^{(k)}}\left[\sum_{i=1}^{m}\left\langle\epsilon_i^{(k)}v_i^{(k)},\overline{\epsilon v}^{(k)}\right\rangle\right] + \mathbb{E}_{\xi^{(k)}}\left[m\left\|\overline{\epsilon v}^{(k)}\right\|^2\right]
$$

$$
= \sum_{i=1}^{m}\mathbb{E}_{\xi_i^{(k)}}\left[\left\|\epsilon_i^{(k)}\right\|^2 v_i^{(k)}\right] - \frac{2}{m}\sum_{i=1}^{m}\mathbb{E}_{\xi_i^{(k)}}\left[\left\|\epsilon_i^{(k)}\right\|^2 v_i^{(k)}\right]
$$

$$
- \frac{2}{m}\sum_{i=1}^{m}\left\langle\mathbb{E}_{\xi_i^{(k)}}\left[\epsilon_i^{(k)}v_i^{(k)}\right],\sum_{\substack{j=1\\j\neq i}}^{m}\mathbb{E}_{\xi_j^{(k)}}\left[\epsilon_j^{(k)}v_j^{(k)}\right]\right\rangle + m\mathbb{E}_{\xi^{(k)}}\left[\left\|\overline{\epsilon v}^{(k)}\right\|^2\right]
$$

$$
= \left(1 - \frac{2}{m} + \frac{1}{m}\right)\sum_{i=1}^{m}\mathbb{E}_{\epsilon_i^{(k)}}\left[\left\|\epsilon_i^{(k)}\right\|^2\right]\mathbb{E}_{v_i^{(k)}}\left[v_i^{(k)}\right]
$$

$$
- \frac{2}{m}\sum_{i=1}^{m}\sum_{\substack{j=1\\j\neq i}}^{m}\left\langle\mathbb{E}_{\epsilon_i^{(k)}}\left[\epsilon_i^{(k)}\right]\mathbb{E}_{v_i^{(k)}}\left[v_i^{(k)}\right],\mathbb{E}_{\epsilon_j^{(k)}}\left[\epsilon_j^{(k)}\right]\mathbb{E}_{v_j^{(k)}}\left[v_j^{(k)}\right]\right\rangle
$$

$$
= \left(1 - \frac{1}{m}\right)\sum_{i=1}^{m}d_i^{(k)}\sigma_i^2 \leq (m-1)d_{max}^{(k)}\sigma^2 \leq md_{max}^{(k)}\sigma^2.
$$

# F  PROOF OF LEMMA 4.1

Using Lemma C.1-(b) on $\bar{\theta}^{(k)}$, the average model parameters at iteration $k$, we get

$$
\left\|\nabla F_i\left(\bar{\theta}^{(k)}\right)\right\|^2 \leq 2\left(\beta_i^2\left\|\bar{\theta}^{(k)} - \theta^\star\right\|^2 + \delta_i^2\right). \tag{12}
$$

Now, if $0 < \alpha^{(k)} \leq \frac{2}{\mu+\beta}$, we can write

$$
\begin{aligned}
\left\|\bar{\theta}^{(k+1)} - \theta^\star\right\|^2 &= \left\|\bar{\theta}^{(k)} - \alpha^{(k)}\overline{\mathbf{g}v}^{(k)} - \theta^\star\right\|^2 \\
&= \left\|\bar{\theta}^{(k)} - \alpha^{(k)}\overline{\nabla v}^{(k)} - \theta^\star\right\|^2 - 2\left\langle \bar{\theta}^{(k)} - \alpha^{(k)}\overline{\nabla v}^{(k)} - \theta^\star, \alpha^{(k)}\overline{\epsilon v}^{(k)}\right\rangle + \left(\alpha^{(k)}\right)^2\left\|\overline{\epsilon v}^{(k)}\right\|^2 \\
&\leq \left(1 + \mu\alpha^{(k)}\right)\left\|\bar{\theta}^{(k)} - \alpha^{(k)}\nabla F\left(\bar{\theta}^{(k)}\right) - \theta^\star\right\|^2 \\
&\quad + \left(1 + \frac{1}{\mu\alpha^{(k)}}\right)\frac{\left(\alpha^{(k)}\right)^2}{m}\sum_{i=1}^{m}\left\|\nabla F_i\left(\bar{\theta}^{(k)}\right) - \nabla F_i\left(\theta_i^{(k)}\right)v_i^{(k)}\right\|^2 \\
&\quad - 2\left\langle \bar{\theta}^{(k)} - \alpha^{(k)}\overline{\nabla v}^{(k)} - \theta^\star, \alpha^{(k)}\overline{\epsilon v}^{(k)}\right\rangle + \left(\alpha^{(k)}\right)^2\left\|\overline{\epsilon v}^{(k)}\right\|^2 \\
&\leq \left(1 + \mu\alpha^{(k)}\right)\left(1 - \mu\alpha^{(k)}\right)^2\left\|\bar{\theta}^{(k)} - \theta^\star\right\|^2 \\
&\quad + \frac{\alpha^{(k)}}{m\mu}\left(1 + \mu\alpha^{(k)}\right)\sum_{\substack{i=1 \\ v_i^{(k)}=1}}^{m}\beta_i^2\left\|\bar{\theta}^{(k)} - \theta_i^{(k)}\right\|^2 \\
&\quad + \frac{2\alpha^{(k)}}{m\mu}\left(1 + \mu\alpha^{(k)}\right)\sum_{\substack{i=1 \\ v_i^{(k)}=0}}^{m}\left(\beta_i^2\left\|\bar{\theta}^{(k)} - \theta^\star\right\|^2 + \delta_i^2\right) \\
&\quad - 2\left\langle \bar{\theta}^{(k)} - \alpha^{(k)}\overline{\nabla v}^{(k)} - \theta^\star, \alpha^{(k)}\overline{\epsilon v}^{(k)}\right\rangle + \left(\alpha^{(k)}\right)^2\left\|\overline{\epsilon v}^{(k)}\right\|^2 \\
&= \left[\left(1 + \mu\alpha^{(k)}\right)\left(1 - \mu\alpha^{(k)}\right)^2 + \frac{2\alpha^{(k)}}{m\mu}\left(1 + \mu\alpha^{(k)}\right)\sum_{i=1}^{m}\beta_i^2\left(1 - v_i^{(k)}\right)\right]\left\|\bar{\theta}^{(k)} - \theta^\star\right\|^2 \\
&\quad + \frac{\alpha^{(k)}}{m\mu}\left(1 + \mu\alpha^{(k)}\right)\sum_{i=1}^{m}\beta_i^2\left\|\theta_i^{(k)} - \bar{\theta}^{(k)}\right\|^2 v_i^{(k)} \\
&\quad + \frac{2\alpha^{(k)}}{m\mu}\left(1 + \mu\alpha^{(k)}\right)\sum_{i=1}^{m}\delta_i^2\left(1 - v_i^{(k)}\right) \\
&\quad - 2\left\langle \bar{\theta}^{(k)} - \alpha^{(k)}\overline{\nabla v}^{(k)} - \theta^\star, \alpha^{(k)}\overline{\epsilon v}^{(k)}\right\rangle + \left(\alpha^{(k)}\right)^2\left\|\overline{\epsilon v}^{(k)}\right\|^2,
\end{aligned}
$$

in which the relationship in first four lines follow from (i) equation 5, (ii) $\mathbf{g}_i^{(k)} = \nabla_i^{(k)} + \epsilon_i^{(k)}$ for all $i \in \mathcal{M}$, (iii) Young's inequality, (iv) Lemma 10 in Qu & Li (2017), Assumption 4.2-(a) and equation 12. Next, we take the expected value of the above inequality and use Assumption 4.3 and Lemma C.2 to get

$$
\begin{aligned}
\mathbb{E}_{\boldsymbol{\Xi}^{(k)}}\left[\left\|\bar{\theta}^{(k+1)} - \theta^\star\right\|^2\right] &\leq \mathbb{E}_{\mathbf{v}^{(k)}}\left[\left(1 + \mu\alpha^{(k)}\right)\left(1 - \mu\alpha^{(k)}\right)^2\right. \\
&\quad \left. + \frac{2\alpha^{(k)}}{m\mu}\left(1 + \mu\alpha^{(k)}\right)\sum_{i=1}^{m}\beta_i^2\left(1 - v_i^{(k)}\right)\right]\mathbb{E}_{\boldsymbol{\Xi}^{(k-1)}}\left[\left\|\bar{\theta}^{(k)} - \theta^\star\right\|^2\right] \\
&\quad + \frac{\alpha^{(k)}}{m\mu}\left(1 + \mu\alpha^{(k)}\right)\sum_{i=1}^{m}\beta_i^2\mathbb{E}_{\boldsymbol{\Xi}^{(k-1)}}\left[\left\|\theta_i^{(k)} - \bar{\theta}^{(k)}\right\|^2\right]\mathbb{E}_{v_i^{(k)}}\left[v_i^{(k)}\right] \\
&\quad + \frac{2\alpha^{(k)}}{m\mu}\left(1 + \mu\alpha^{(k)}\right)\sum_{i=1}^{m}\delta_i^2\mathbb{E}_{v_i^{(k)}}\left[1 - v_i^{(k)}\right] \\
&\quad - 2\mathbb{E}_{\boldsymbol{\Xi}^{(k-1)}\cup\mathbf{v}^{(k)}}\left[\left\langle \bar{\theta}^{(k)} - \alpha^{(k)}\overline{\nabla v}^{(k)} - \theta^\star, \alpha^{(k)}\mathbb{E}_{\mathbf{E}^{(k)}}\left[\overline{\epsilon v}^{(k)}\right]\right\rangle\right] \\
&\quad + \left(\alpha^{(k)}\right)^2\mathbb{E}_{\xi^{(k)}}\left[\left\|\overline{\epsilon v}^{(k)}\right\|^2\right]
\end{aligned}
$$

$$\leq \left[ \left(1 + \mu\alpha^{(k)}\right)\left(1 - \mu\alpha^{(k)}\right)^2 + \frac{2\alpha^{(k)}}{m\mu}\left(1 + \mu\alpha^{(k)}\right)\sum_{i=1}^{m}\beta^2\left(1 - d_i^{(k)}\right)\right]$$
$$\mathbb{E}_{\boldsymbol{\Xi}^{(k-1)}}\left[\left\|\bar{\theta}^{(k)} - \theta^{\star}\right\|^2\right]$$
$$+ \frac{\alpha^{(k)}}{m\mu}\left(1 + \mu\alpha^{(k)}\right)\sum_{i=1}^{m}d_i^{(k)}\beta_i^2\mathbb{E}_{\boldsymbol{\Xi}^{(k-1)}}\left[\left\|\theta_i^{(k)} - \bar{\theta}^{(k)}\right\|^2\right]$$
$$+ \frac{2\alpha^{(k)}}{m\mu}\left(1 + \mu\alpha^{(k)}\right)\sum_{i=1}^{m}\delta_i^2\left(1 - d_i^{(k)}\right) + \frac{\left(\alpha^{(k)}\right)^2 d_{max}^{(k)}\sigma^2}{m}$$
$$- 2\mathbb{E}_{\boldsymbol{\Xi}^{(k-1)}\cup\mathbf{V}^{(k)}}\left[\left\langle \bar{\theta}^{(k)} - \alpha^{(k)}\overline{\nabla v}^{(k)} - \theta^{\star}, \alpha^{(k)}\mathbb{E}_{\mathbf{E}^{(k)}}\left[\overline{\epsilon v}^{(k)}\right]\right\rangle\right]$$
$$+ \left(\alpha^{(k)}\right)^2\mathbb{E}_{\xi^{(k)}}\left[\left\|\overline{\epsilon v}^{(k)}\right\|^2\right]$$
$$= \left[1 - \mu\alpha^{(k)}\left(1 + \mu\alpha^{(k)} - \left(\mu\alpha^{(k)}\right)^2\right) + \frac{2\alpha^{(k)}}{\mu}\left(1 + \mu\alpha^{(k)}\right)\left(1 - d_{min}^{(k)}\right)\beta^2\right]$$
$$\mathbb{E}_{\boldsymbol{\Xi}^{(k-1)}}\left[\left\|\bar{\theta}^{(k)} - \theta^{\star}\right\|^2\right]$$
$$+ \left(1 + \mu\alpha^{(k)}\right)\frac{\alpha^{(k)}d_{max}^{(k)}\beta^2}{m\mu}\mathbb{E}_{\boldsymbol{\Xi}^{(k-1)}}\left[\left\|\boldsymbol{\Theta}^{(k)} - \mathbf{1}_m\bar{\theta}^{(k)}\right\|^2\right]$$
$$+ \frac{2\alpha^{(k)}}{\mu}\left(1 + \mu\alpha^{(k)}\right)\left(1 - d_{min}^{(k)}\right)\delta^2 + \frac{\left(\alpha^{(k)}\right)^2 d_{max}^{(k)}\sigma^2}{m}$$

## G   PROOF OF LEMMA C.3

Noting that

$$\left\|\mathbf{V}^{(k)}\nabla^{(k)} - \mathbf{1}_m\overline{\nabla v}^{(k)}\right\|^2 = 2\left\|\mathbf{V}^{(k)}\nabla^{(k)} - \mathbf{V}^{(k)}\nabla\mathbf{F}^{(k)}\right\|^2 + 2\left\|\mathbf{V}^{(k)}\nabla\mathbf{F}^{(k)} - \mathbf{1}_m\overline{\nabla v}^{(k)}\right\|^2,$$

we first find an upper bound for each of the two terms above separately. For the first term, we have

$$\left\|\mathbf{V}^{(k)}\nabla^{(k)} - \mathbf{V}^{(k)}\nabla\mathbf{F}^{(k)}\right\|^2 = \sum_{i=1}^{m}\left\|\nabla F_i\left(\theta_i^{(k)}\right)v_i^{(k)} - \nabla F\left(\theta_i^{(k)}\right)v_i^{(k)}\right\|^2$$
$$= \sum_{i=1}^{m}\left\|\nabla F_i\left(\theta_i^{(k)}\right) - \nabla F\left(\theta_i^{(k)}\right)\right\|^2 v_i^{(k)} \leq \sum_{i=1}^{m}\left(\delta_i + \zeta_i\left\|\theta_i^{(k)} - \theta^{\star}\right\|\right)^2 v_i^{(k)}$$
$$\leq \sum_{i=1}^{m}\left(\delta_i + \zeta_i\left\|\theta_i^{(k)} - \bar{\theta}^{(k)}\right\| + \zeta_i\left\|\bar{\theta}^{(k)} - \theta^{\star}\right\|\right)^2 v_i^{(k)}$$
$$\leq 3\sum_{i=1}^{m}\left(\delta_i^2 + \zeta_i^2\left\|\theta_i^{(k)} - \bar{\theta}^{(k)}\right\|^2 + \zeta_i^2\left\|\bar{\theta}^{(k)} - \theta^{\star}\right\|^2\right)v_i^{(k)},$$

where Assumption 4.2-(c) was used in the first inequality. Next, for the second term, we use Assumption 4.2-(a) write

$$\left\|\mathbf{V}^{(k)}\nabla\mathbf{F}^{(k)} - \mathbf{1}_m\overline{\nabla v}^{(k)}\right\|^2 = \sum_{i=1}^{m}\left\|\nabla F\left(\theta_i^{(k)}\right)v_i^{(k)} - \overline{\nabla v}^{(k)}\right\|^2$$
$$= \sum_{i=1}^{m}\left\|\frac{1}{m}\sum_{j=1}^{m}\left(\nabla F_j\left(\theta_i^{(k)}\right)v_i^{(k)} - \nabla F_j\left(\theta_j^{(k)}\right)v_j^{(k)}\right)\right\|^2$$

$$\leq \sum_{i=1}^{m} \frac{1}{m} \sum_{j=1}^{m} \left\| \nabla F_j\left(\theta_i^{(k)}\right) v_i^{(k)} - \nabla F_j\left(\theta_j^{(k)}\right) v_j^{(k)} \right\|^2$$

$$= \frac{1}{m} \sum_{i=1}^{m} \sum_{j=1}^{m} \left\| \nabla F_j\left(\theta_i^{(k)}\right) v_i^{(k)} - \nabla F_j\left(\bar{\theta}^{(k)}\right) v_i^{(k)} + \nabla F_j\left(\bar{\theta}^{(k)}\right) v_i^{(k)} - \nabla F_j(\theta^\star) v_i^{(k)} \right.$$
$$\left. + \nabla F_j(\theta^\star) v_i^{(k)} - \nabla F_j\left(\bar{\theta}^{(k)}\right) v_j^{(k)} + \nabla F_j\left(\bar{\theta}^{(k)}\right) v_j^{(k)} - \nabla F_j\left(\theta_j^{(k)}\right) v_j^{(k)} \right\|^2$$

$$= \frac{3}{m} \sum_{i=1}^{m} \sum_{j=1}^{m} \left( \left\| \nabla F_j\left(\theta_i^{(k)}\right) - \nabla F_j\left(\bar{\theta}^{(k)}\right) \right\|^2 v_i^{(k)} + \left\| \nabla F_j\left(\bar{\theta}^{(k)}\right) - \nabla F_j(\theta^\star) \right\|^2 \left( v_i^{(k)} - v_j^{(k)} \right)^2 \right.$$
$$\left. + \left\| \nabla F_j\left(\bar{\theta}^{(k)}\right) - \nabla F_j\left(\theta_j^{(k)}\right) \right\|^2 v_j^{(k)} \right)$$

$$\leq \frac{3}{m} \sum_{i=1}^{m} \sum_{j=1}^{m} \left( \beta_j^2 \left\| \theta_i^{(k)} - \bar{\theta}^{(k)} \right\|^2 v_i^{(k)} + \beta_j^2 \left\| \bar{\theta}^{(k)} - \theta^\star \right\|^2 \left( v_i^{(k)} + v_j^{(k)} - 2 v_i^{(k)} v_j^{(k)} \right) \right.$$
$$\left. + \beta_j^2 \left\| \bar{\theta}^{(k)} - \theta_j^{(k)} \right\|^2 v_j^{(k)} \right)$$

$$\leq \frac{3}{m} \sum_{j=1}^{m} \beta_j^2 \sum_{i=1}^{m} \left( \left\| \theta_i^{(k)} - \bar{\theta}^{(k)} \right\|^2 v_i^{(k)} + \left\| \bar{\theta}^{(k)} - \theta^\star \right\|^2 \left( v_i^{(k)} + v_j^{(k)} - 2 v_i^{(k)} v_j^{(k)} \right) \right.$$
$$\left. + \left\| \bar{\theta}^{(k)} - \theta_j^{(k)} \right\|^2 v_j^{(k)} \right)$$

$$\leq \frac{3\beta^2}{m} \left[ m \sum_{i=1}^{m} \left\| \theta_i^{(k)} - \bar{\theta}^{(k)} \right\|^2 v_i^{(k)} + m \sum_{j=1}^{m} \left\| \bar{\theta}^{(k)} - \theta_j^{(k)} \right\|^2 v_j^{(k)} \right.$$
$$\left. + \left\| \bar{\theta}^{(k)} - \theta^\star \right\|^2 \left( m \sum_{i=1}^{m} v_i^{(k)} + m \sum_{j=1}^{m} v_j^{(k)} - 2 \sum_{i=1}^{m} \sum_{j=1}^{m} v_i^{(k)} v_j^{(k)} \right) \right]$$

$$\leq 6\beta^2 \sum_{i=1}^{m} \left\| \theta_i^{(k)} - \bar{\theta}^{(k)} \right\|^2 v_i^{(k)} + 6\beta^2 \left\| \bar{\theta}^{(k)} - \theta^\star \right\|^2 \left( \sum_{i=1}^{m} v_i^{(k)} - \frac{1}{m} \left( \sum_{i=1}^{m} v_i^{(k)} \right)^2 \right)$$

$$\leq 6\beta^2 \sum_{i=1}^{m} \left[ \left\| \theta_i^{(k)} - \bar{\theta}^{(k)} \right\|^2 + \left\| \bar{\theta}^{(k)} - \theta^\star \right\|^2 \left( 1 - \frac{1}{m} \sum_{j=1}^{m} v_j^{(k)} \right) \right] v_i^{(k)}.$$

Now, by combining the two components together we get

$$\left\| \mathbf{V}^{(k)} \nabla^{(k)} - \mathbf{1}_m \overline{\nabla v}^{(k)} \right\|^2 = 2 \left\| \mathbf{V}^{(k)} \nabla^{(k)} - \mathbf{V}^{(k)} \nabla \mathbf{F}^{(k)} \right\|^2 + 2 \left\| \mathbf{V}^{(k)} \nabla \mathbf{F}^{(k)} - \mathbf{1}_m \overline{\nabla v}^{(k)} \right\|^2$$
$$\leq 3 \sum_{i=1}^{m} \left[ \delta_i^2 + \left( \zeta_i^2 + 2\beta^2 \right) \left\| \theta_i^{(k)} - \bar{\theta}^{(k)} \right\|^2 \right.$$
$$\left. + \left( \zeta_i^2 + 2\beta^2 \left( 1 - \frac{1}{m} \sum_{j=1}^{m} v_j^{(k)} \right) \right) \left\| \bar{\theta}^{(k)} - \theta^\star \right\|^2 \right] v_i^{(k)}.$$

Finally, we take the expected value of the above inequality to conclude the proof.

$$\mathbb{E}_{\mathbf{\Xi}^{(k)}}\left[\left\|\mathbf{V}^{(k)}\nabla^{(k)} - \mathbf{1}_m\overline{\nabla v}^{(k)}\right\|^2\right]$$

$$\leq 3\sum_{i=1}^{m}\left[\left(\delta_i^2 + (\zeta_i^2 + 2\beta^2)\mathbb{E}_{\mathbf{\Xi}^{(k-1)}}\left[\left\|\theta_i^{(k)} - \bar{\theta}^{(k)}\right\|^2\right]\right)\mathbb{E}_{v_i^{(k)}}\left[v_i^{(k)}\right] + \left(\zeta_i^2\mathbb{E}_{v_i^{(k)}}\left[v_i^{(k)}\right]\right.\right.$$

$$\left.\left. +2\beta^2\left(\mathbb{E}_{v_i^{(k)}}\left[v_i^{(k)}\right] - \frac{1}{m}\mathbb{E}_{v_i^{(k)}}\left[v_i^{(k)}\right] - \frac{1}{m}\mathbb{E}_{v_i^{(k)}}\left[v_i^{(k)}\right]\sum_{\substack{j=1\\j\neq i}}^{m}\mathbb{E}_{v_j^{(k)}}\left[v_j^{(k)}\right]\right)\right)\right.$$

$$\left. \mathbb{E}_{\mathbf{\Xi}^{(k-1)}}\left[\left\|\bar{\theta}^{(k)} - \theta^\star\right\|^2\right]\right]$$

$$\leq 3\sum_{i=1}^{m}\left[\left(\delta_i^2 + (\zeta_i^2 + 2\beta^2)\mathbb{E}_{\mathbf{\Xi}^{(k-1)}}\left[\left\|\theta_i^{(k)} - \bar{\theta}^{(k)}\right\|^2\right]\right)d_i^{(k)} + \left(\zeta_i^2 d_i^{(k)}\right.\right.$$

$$\left.\left. +2\beta^2\left(d_i^{(k)} - \frac{1}{m}d_i^{(k)} - \frac{1}{m}d_i^{(k)}\sum_{\substack{j=1\\j\neq i}}^{m}d_j^{(k)}\right)\right)\mathbb{E}_{\mathbf{\Xi}^{(k-1)}}\left[\left\|\bar{\theta}^{(k)} - \theta^\star\right\|^2\right]\right]$$

$$\leq 3\sum_{i=1}^{m}d_i^{(k)}\left[\delta_i^2 + (\zeta_i^2 + 2\beta^2)\mathbb{E}_{\mathbf{\Xi}^{(k-1)}}\left[\left\|\theta_i^{(k)} - \bar{\theta}^{(k)}\right\|^2\right]\right.$$

$$\left. +\left(\zeta_i^2 + 2\beta^2\left(1 - \frac{1}{m}\left[1 + \sum_{\substack{j=1\\j\neq i}}^{m}d_j^{(k)}\right]\right)\right)\mathbb{E}_{\mathbf{\Xi}^{(k-1)}}\left[\left\|\bar{\theta}^{(k)} - \theta^\star\right\|^2\right]\right]$$

$$\leq 3d_{max}^{(k)}\left[m\delta^2 + (\zeta^2 + 2\beta^2)\mathbb{E}_{\mathbf{\Xi}^{(k-1)}}\left[\left\|\mathbf{\Theta}^{(k)} - \mathbf{1}_m\bar{\theta}^{(k)}\right\|^2\right]\right.$$

$$\left. +m\left(\zeta^2 + 2\beta^2\left(1 - \frac{1}{m}\left[1 + (m-1)d_{min}^{(k)}\right]\right)\right)\mathbb{E}_{\mathbf{\Xi}^{(k-1)}}\left[\left\|\bar{\theta}^{(k)} - \theta^\star\right\|^2\right]\right]$$

$$= 3d_{max}^{(k)}\left[m\delta^2 + (\zeta^2 + 2\beta^2)\mathbb{E}_{\mathbf{\Xi}^{(k-1)}}\left[\left\|\mathbf{\Theta}^{(k)} - \mathbf{1}_m\bar{\theta}^{(k)}\right\|^2\right]\right.$$

$$\left. +m\left(\zeta^2 + 2\beta^2\left(1 - \frac{1}{m}\right)\left(1 - d_{min}^{(k)}\right)\right)\mathbb{E}_{\mathbf{\Xi}^{(k-1)}}\left[\left\|\bar{\theta}^{(k)} - \theta^\star\right\|^2\right]\right]$$

$$\leq 3d_{max}^{(k)}\left[m\delta^2 + (\zeta^2 + 2\beta^2)\mathbb{E}_{\mathbf{\Xi}^{(k-1)}}\left[\left\|\mathbf{\Theta}^{(k)} - \mathbf{1}_m\bar{\theta}^{(k)}\right\|^2\right]\right.$$

$$\left. +m\left(\zeta^2 + 2\beta^2\left(1 - d_{min}^{(k)}\right)\right)\mathbb{E}_{\mathbf{\Xi}^{(k-1)}}\left[\left\|\bar{\theta}^{(k)} - \theta^\star\right\|^2\right]\right].$$

## H  PROOF OF LEMMA C.4

(a) We take the expected value of the matrix $\mathbf{P}^{(k)}$ by looking at its individual elements. We have

$$\mathbb{E}_{\hat{\mathbf{V}}^{(k)}}\left[\mathbf{P}^{(k)}\right] = \left[\mathbb{E}_{\hat{v}_{ij}^{(k)}}\left[p_{ij}^{(k)}\right]\right]_{1\leq i,j\leq m} = \begin{cases} \left[r_{ij}\mathbb{E}_{\hat{v}_{ij}^{(k)}}\left[\hat{v}_{ij}^{(k)}\right]\right]_{1\leq i,j\leq m} & i\neq j \\ \left[1 - \sum_{j=1}^{m}r_{ij}\mathbb{E}_{\hat{v}_{ij}^{(k)}}\left[\hat{v}_{ij}^{(k)}\right]\right]_{1\leq i\leq m} & i=j \end{cases}$$

$$= \begin{cases} \left[b_{ij}^{(k)}r_{ij}\right]_{1\leq i,j\leq m} & i\neq j \\ \left[1 - \sum_{j=1}^{m}b_{ij}^{(k)}r_{ij}\right]_{1\leq i\leq m} & i=j \end{cases} = \bar{\mathbf{R}}^{(k)}$$

(b) Similar to the proof of the previous part, we take the expected value of $\left(\mathbf{P}^{(k)}\right)^T \mathbf{P}^{(k)} = \left(\mathbf{P}^{(k)}\right)^2$ by looking at its individual elements. We can write

if $i \neq j$:

$$
\mathbb{E}_{\hat{\mathbf{V}}^{(k)}} \left[ \sum_{l=1}^{m} p_{il}^{(k)} p_{lj}^{(k)} \right]
$$

$$
= \mathbb{E}_{\hat{\mathbf{V}}^{(k)}} \left[ \sum_{\substack{l=1 \\ l \neq i,j}}^{m} r_{il} r_{lj} \hat{v}_{il}^{(k)} \hat{v}_{lj}^{(k)} + \left( 1 - \sum_{q=1}^{m} r_{iq} \hat{v}_{iq}^{(k)} \right) r_{ij} \hat{v}_{ij}^{(k)} + \left( 1 - \sum_{q=1}^{m} r_{jq} \hat{v}_{jq}^{(k)} \right) r_{ij} \hat{v}_{ij}^{(k)} \right]
$$

$$
= \mathbb{E}_{\hat{\mathbf{V}}^{(k)}} \left[ \sum_{l=1}^{m} r_{il} r_{lj} \hat{v}_{il}^{(k)} \hat{v}_{lj}^{(k)} + \left( 2 - \sum_{q=1}^{m} \left( r_{iq} \hat{v}_{iq}^{(k)} + r_{jq} \hat{v}_{jq}^{(k)} \right) \right) r_{ij} \hat{v}_{ij}^{(k)} \right]
$$

$$
= \mathbb{E}_{\hat{\mathbf{V}}^{(k)}} \left[ \sum_{l=1}^{m} r_{il} r_{lj} \hat{v}_{il}^{(k)} \hat{v}_{lj}^{(k)} + \left( 2 - \sum_{\substack{q=1 \\ q \neq i,j}}^{m} \left( r_{iq} \hat{v}_{iq}^{(k)} + r_{jq} \hat{v}_{jq}^{(k)} \right) \right) r_{ij} \hat{v}_{ij}^{(k)} \right.
$$

$$
\left. - \left( r_{ij} \hat{v}_{ij}^{(k)} + r_{ji} \hat{v}_{ji}^{(k)} \right) r_{ij} \hat{v}_{ij}^{(k)} \right]
$$

$$
= \sum_{l=1}^{m} r_{il} r_{lj} \mathbb{E}_{\hat{v}_{il}^{(k)}} \left[ \hat{v}_{il}^{(k)} \right] \mathbb{E}_{\hat{v}_{lj}^{(k)}} \left[ \hat{v}_{lj}^{(k)} \right]
$$

$$
+ \left( 2 - \sum_{\substack{q=1 \\ q \neq i,j}}^{m} \left( r_{iq} \mathbb{E}_{\hat{v}_{iq}^{(k)}} \left[ \hat{v}_{iq}^{(k)} \right] + r_{jq} \mathbb{E}_{\hat{v}_{jq}^{(k)}} \left[ \hat{v}_{jq}^{(k)} \right] \right) \right) r_{ij} \mathbb{E}_{\hat{v}_{ij}^{(k)}} \left[ \hat{v}_{ij}^{(k)} \right]
$$

$$
- 2 r_{ij}^2 \mathbb{E}_{\hat{v}_{ij}^{(k)}} \left[ \hat{v}_{ij}^{(k)} \right]
$$

$$
= \sum_{l=1}^{m} b_{il}^{(k)} r_{il} b_{lj}^{(k)} r_{lj} + \left( 2 - \sum_{\substack{q=1 \\ q \neq i,j}}^{m} \left( b_{iq}^{(k)} r_{iq} + b_{jq}^{(k)} r_{jq} \right) \right) b_{ij}^{(k)} r_{ij} - 2 b_{ij}^{(k)} r_{ij}^2
$$

On the other hand,

if $i = j$:

$$
\mathbb{E}_{\hat{\mathbf{V}}^{(k)}} \left[ \sum_{l=1}^{m} p_{il}^{(k)} p_{lj}^{(k)} \right] = \mathbb{E}_{\hat{\mathbf{V}}^{(k)}} \left[ \sum_{\substack{l=1 \\ l \neq i}}^{m} r_{il} r_{li} \hat{v}_{il}^{(k)} \hat{v}_{li}^{(k)} + \left( 1 - \sum_{q=1}^{m} r_{iq} \hat{v}_{iq}^{(k)} \right)^2 \right]
$$

$$
= \mathbb{E}_{\hat{\mathbf{V}}^{(k)}} \left[ \sum_{l=1}^{m} r_{il}^2 \hat{v}_{il}^{(k)} + 1 - 2 \sum_{q=1}^{m} r_{iq} \hat{v}_{iq}^{(k)} + \sum_{q=1}^{m} \sum_{t=1}^{m} r_{iq} r_{it} \hat{v}_{iq}^{(k)} \hat{v}_{it}^{(k)} \right]
$$

$$
= \mathbb{E}_{\hat{\mathbf{V}}^{(k)}} \left[ \sum_{l=1}^{m} r_{il}^2 \hat{v}_{il}^{(k)} + 1 - 2 \sum_{q=1}^{m} r_{iq} \hat{v}_{iq}^{(k)} + \sum_{q=1}^{m} \sum_{\substack{t=1 \\ t \neq q}}^{m} r_{iq} r_{it} \hat{v}_{iq}^{(k)} \hat{v}_{it}^{(k)} + \sum_{q=1}^{m} r_{iq}^2 \hat{v}_{iq}^{(k)} \right]
$$

$$
= 2 \sum_{l=1}^{m} r_{il}^2 \mathbb{E}_{\hat{v}_{il}^{(k)}} \left[ \hat{v}_{il}^{(k)} \right] + 1 - 2 \sum_{q=1}^{m} r_{iq} \mathbb{E}_{\hat{v}_{iq}^{(k)}} \left[ \hat{v}_{iq}^{(k)} \right] + \sum_{q=1}^{m} \sum_{\substack{t=1 \\ t \neq q}}^{m} r_{iq} r_{it} \mathbb{E}_{\hat{v}_{iq}^{(k)}} \left[ \hat{v}_{iq}^{(k)} \right] \mathbb{E}_{\hat{v}_{it}^{(k)}} \left[ \hat{v}_{it}^{(k)} \right]
$$

$$
= 2 \sum_{l=1}^{m} b_{il}^{(k)} r_{il}^2 + 1 - 2 \sum_{q=1}^{m} b_{iq}^{(k)} r_{iq} + \sum_{q=1}^{m} \sum_{\substack{t=1 \\ t \neq q}}^{m} b_{iq}^{(k)} r_{iq} b_{it}^{(k)} r_{it}
$$

Finally, comparing the above expression with the elements of $\left(\bar{\mathbf{R}}^{(k)}\right)^2$, we get

$$
\begin{aligned}
\mathbb{E}_{\hat{\mathbf{V}}^{(k)}}\left[\left(\mathbf{P}^{(k)}\right)^2\right] &= \left[\mathbb{E}_{\hat{\mathbf{V}}^{(k)}}\left[\sum_{l=1}^m p_{il}^{(k)} p_{lj}^{(k)}\right]\right]_{1\le i,j\le m} \\
&= \left(\bar{\mathbf{R}}^{(k)}\right)^2 + \begin{cases} \left[-2b_{ij}^{(k)}\left(1-b_{ij}^{(k)}\right) r_{ij}^2\right]_{1\le i,j\le m} & i\ne j \\ \left[\sum_{l=1}^m 2b_{il}^{(k)}\left(1-b_{il}^{(k)}\right) r_{il}^2\right]_{1\le i\le m} & i=j \end{cases} \\
&= \left(\bar{\mathbf{R}}^{(k)}\right)^2 + \mathbf{R}_0^{(k)} \triangleq \tilde{\mathbf{R}}^{(k)},
\end{aligned}
$$

in which $\mathbf{R}_0^{(k)}$ is a matrix whose rows and columns sum to zero.

(c) In order to prove this inequality, we expand the left-hand side frobenius norm by its columns and use the results of part (b) of this lemma. We have

$$
\begin{aligned}
\mathbb{E}_{\mathbf{\Xi}^{(k)}}\left[\left\|\mathbf{P}^{(k)}\mathbf{\Theta}^{(k)} - \mathbf{1}_m\bar{\theta}^{(k)}\right\|^2\right] &= \mathbb{E}_{\mathbf{\Xi}^{(k)}}\left[\sum_{j=1}^m \left\|\mathbf{P}^{(k)}\theta_j^{(k)} - \bar{\theta}_j^{(k)}\mathbf{1}_m\right\|^2\right] \\
&= \mathbb{E}_{\mathbf{\Xi}^{(k)}}\left[\sum_{j=1}^m \left\|\mathbf{P}^{(k)}\theta_j^{(k)} - \bar{\theta}_j^{(k)}\mathbf{P}^{(k)}\mathbf{1}_m\right\|^2\right] = \mathbb{E}_{\mathbf{\Xi}^{(k)}}\left[\sum_{j=1}^m \left\|\mathbf{P}^{(k)}\left(\theta_j^{(k)} - \bar{\theta}_j^{(k)}\mathbf{1}_m\right)\right\|^2\right] \\
&= \mathbb{E}_{\mathbf{\Xi}^{(k)}}\left[\sum_{j=1}^m \left(\theta_j^{(k)} - \bar{\theta}_j^{(k)}\mathbf{1}_m\right)^T \left(\mathbf{P}^{(k)}\right)^T \mathbf{P}^{(k)} \left(\theta_j^{(k)} - \bar{\theta}_j^{(k)}\mathbf{1}_m\right)\right] \\
&= \mathbb{E}_{\mathbf{\Xi}^{(k-1)}}\left[\sum_{j=1}^m \left(\theta_j^{(k)} - \bar{\theta}_j^{(k)}\mathbf{1}_m\right)^T \mathbb{E}_{\hat{\mathbf{V}}^{(k)}}\left[\mathbf{P}^{(k)}\mathbf{P}^{(k)}\right] \left(\theta_j^{(k)} - \bar{\theta}_j^{(k)}\mathbf{1}_m\right)\right] \\
&= \mathbb{E}_{\mathbf{\Xi}^{(k-1)}}\left[\sum_{j=1}^m \left(\theta_j^{(k)} - \bar{\theta}_j^{(k)}\mathbf{1}_m\right)^T \tilde{\mathbf{R}}^{(k)} \left(\theta_j^{(k)} - \bar{\theta}_j^{(k)}\mathbf{1}_m\right)\right] \\
&\le \mathbb{E}_{\mathbf{\Xi}^{(k-1)}}\left[\sum_{j=1}^m \tilde{\rho}^{(k)} \left\|\theta_j^{(k)} - \bar{\theta}_j^{(k)}\mathbf{1}_m\right\|^2\right] \le \tilde{\rho}^{(k)} \mathbb{E}_{\mathbf{\Xi}^{(k-1)}}\left[\left\|\mathbf{\Theta}^{(k)} - \mathbf{1}_m\bar{\theta}^{(k)}\right\|^2\right].
\end{aligned}
$$

# I  PROOF OF LEMMA 4.2

Using equation 3 and equation 5, we first expand the left-hand side norm, and then use Young's inequality to get

$$
\begin{aligned}
\left\|\mathbf{\Theta}^{(k+1)} - \mathbf{1}_m\bar{\theta}^{(k+1)}\right\|^2 &= \left\|\mathbf{P}^{(k)}\mathbf{\Theta}^{(k)} - \mathbf{1}_m\bar{\theta}^{(k)} - \alpha^{(k)}\left(\mathbf{V}^{(k)}\mathbf{G}^{(k)} - \mathbf{1}_m\overline{gv}^{(k)}\right)\right\|^2 \\
&\le \left\|\mathbf{P}^{(k)}\mathbf{\Theta}^{(k)} - \mathbf{1}_m\bar{\theta}^{(k)} - \alpha^{(k)}\left(\mathbf{V}^{(k)}\nabla^{(k)} - \mathbf{1}_m\overline{\nabla v}^{(k)}\right)\right\|^2 \\
&\quad -2\alpha^{(k)}\left\langle \mathbf{P}^{(k)}\mathbf{\Theta}^{(k)} - \mathbf{1}_m\bar{\theta}^{(k)} - \alpha^{(k)}\left(\mathbf{V}^{(k)}\nabla^{(k)} - \mathbf{1}_m\overline{\nabla v}^{(k)}\right),\right. \\
&\qquad\qquad\qquad\qquad\qquad\qquad \left. \mathbf{V}^{(k)}\mathbf{E}^{(k)} - \mathbf{1}_m\overline{\epsilon v}^{(k)}\right\rangle \\
&\quad + \left(\alpha^{(k)}\right)^2 \left\|\mathbf{V}^{(k)}\mathbf{E}^{(k)} - \mathbf{1}_m\overline{\epsilon v}^{(k)}\right\|^2
\end{aligned}
$$

$$\leq \left(1 + \frac{1 - \tilde{\rho}^{(k)}}{2\tilde{\rho}^{(k)}}\right) \left\|\mathbf{P}^{(k)}\mathbf{\Theta}^{(k)} - \mathbf{1}_m \bar{\theta}^{(k)}\right\|^2$$
$$+ \left(1 + \frac{2\tilde{\rho}^{(k)}}{1 - \tilde{\rho}^{(k)}}\right) \left(\alpha^{(k)}\right)^2 \left\|\mathbf{V}^{(k)}\nabla^{(k)} - \mathbf{1}_m \overline{\nabla v}^{(k)}\right\|^2$$
$$- 2\alpha^{(k)} \left\langle \mathbf{P}^{(k)}\mathbf{\Theta}^{(k)} - \mathbf{1}_m \bar{\theta}^{(k)} - \alpha^{(k)}\left(\mathbf{V}^{(k)}\nabla^{(k)} - \mathbf{1}_m \overline{\nabla v}^{(k)}\right),\right.$$
$$\left.\left(\mathbf{I}_m - \frac{1}{m}\mathbf{1}_m\mathbf{1}_m^T\right)\mathbf{V}^{(k)}\mathbf{E}^{(k)}\right\rangle$$
$$+ \left(\alpha^{(k)}\right)^2 \left\|\mathbf{V}^{(k)}\mathbf{E}^{(k)} - \mathbf{1}_m \overline{\epsilon v}^{(k)}\right\|^2.$$

Next, we take the expected value of the above inequality and use Lemmas C.2, C.3 and C.4-(c) to get

$$\mathbb{E}_{\mathbf{\Xi}^{(k)}}\left[\left\|\mathbf{\Theta}^{(k+1)} - \mathbf{1}_m \bar{\theta}^{(k+1)}\right\|^2\right] \leq \frac{1 + \tilde{\rho}^{(k)}}{2\tilde{\rho}^{(k)}}\tilde{\rho}^{(k)}\mathbb{E}_{\mathbf{\Xi}^{(k-1)}}\left[\left\|\mathbf{\Theta}^{(k)} - \mathbf{1}_m \bar{\theta}^{(k)}\right\|^2\right]$$
$$+3\frac{1 + \tilde{\rho}^{(k)}}{1 - \tilde{\rho}^{(k)}}d_{max}^{(k)}\left(\alpha^{(k)}\right)^2 \left(m\delta^2 + \left(\zeta^2 + 2\beta^2\right)\mathbb{E}_{\mathbf{\Xi}^{(k-1)}}\left[\left\|\mathbf{\Theta}^{(k)} - \mathbf{1}_m \bar{\theta}^{(k)}\right\|^2\right]\right.$$
$$\left.+ m\left(\zeta^2 + 2\beta^2\left(1 - d_{min}^{(k)}\right)\right)\mathbb{E}_{\mathbf{\Xi}^{(k-1)}}\left[\left\|\bar{\theta}^{(k)} - \theta^\star\right\|^2\right]\right)$$
$$-2\alpha^{(k)}\mathbb{E}_{\mathbf{\Xi}^{(k)}\setminus\mathbf{E}^{(k)}}\left[\left\langle \mathbf{P}^{(k)}\mathbf{\Theta}^{(k)} - \mathbf{1}_m \bar{\theta}^{(k)} - \alpha^{(k)}\left(\mathbf{V}^{(k)}\nabla^{(k)} - \mathbf{1}_m \overline{\nabla v}^{(k)}\right),\right.\right.$$
$$\left.\left.\left(\mathbf{I}_m - \frac{1}{m}\mathbf{1}_m\mathbf{1}_m^T\right)\mathbf{V}^{(k)}\mathbb{E}_{\mathbf{E}^{(k)}}\left[\mathbf{E}^{(k)}\right]\right\rangle\right]$$
$$+ m\left(\alpha^{(k)}\right)^2 d_{max}^{(k)}\sigma^2$$
$$\leq \left[\frac{1 + \tilde{\rho}^{(k)}}{2} + 3\frac{1 + \tilde{\rho}^{(k)}}{1 - \tilde{\rho}^{(k)}}d_{max}^{(k)}\left(\alpha^{(k)}\right)^2 \left(\zeta^2 + 2\beta^2\right)\right]\mathbb{E}_{\mathbf{\Xi}^{(k-1)}}\left[\left\|\mathbf{\Theta}^{(k)} - \mathbf{1}_m \bar{\theta}^{(k)}\right\|^2\right]$$
$$+ 3\frac{1 + \tilde{\rho}^{(k)}}{1 - \tilde{\rho}^{(k)}}md_{max}^{(k)}\left(\alpha^{(k)}\right)^2 \left(\zeta^2 + 2\beta^2\left(1 - d_{min}^{(k)}\right)\right)\mathbb{E}_{\mathbf{\Xi}^{(k-1)}}\left[\left\|\bar{\theta}^{(k)} - \theta^\star\right\|^2\right]$$
$$+ m\left(\alpha^{(k)}\right)^2 d_{max}^{(k)}\left(3\frac{1 + \tilde{\rho}^{(k)}}{1 - \tilde{\rho}^{(k)}}\delta^2 + \sigma^2\right)$$

## J   PROOF OF PROPOSITION 4.1

**Step 1: Setting up the proof.** We want to find the conditions under which we will have $\rho\left(\mathbf{\Phi}^{(k)}\right) < 1$. As we have $\mathbf{\Phi}^{(k)} = [\phi_{ij}]_{1\leq i,j\leq 2}$ and $\rho\left(\mathbf{\Phi}^{(k)}\right) = \max\left\{\left|\lambda_1^{(k)}\right|, \left|\lambda_2^{(k)}\right|\right\}$ where $\lambda_i^{(k)}$ are the eigenvalues of the matrix $\mathbf{\Phi}^{(k)}$ for $i = 1, 2$, we need to show that $\max\left\{\left|\lambda_1^{(k)}\right|, \left|\lambda_2^{(k)}\right|\right\} < 1$. Therefore, we first write the eigenvalue equation of the matrix as

$$\left(\lambda - \phi_{11}^{(k)}\right)\left(\lambda - \phi_{22}^{(k)}\right) - \phi_{12}^{(k)}\phi_{21}^{(k)} = 0 \quad \Rightarrow \quad \lambda^2 - \left(\phi_{11}^{(k)} + \phi_{22}^{(k)}\right)\lambda + \phi_{11}^{(k)}\phi_{22}^{(k)} - \phi_{12}^{(k)}\phi_{21}^{(k)} = 0.$$

Since this is a quadratic equation in the form of $a\lambda^2 + b\lambda + c = 0$, we know that if $b < 0$, and $a, c > 0$ and the determinant is positive, we will have $\max\left\{\left|\lambda_1^{(k)}\right|, \left|\lambda_2^{(k)}\right|\right\} = \frac{-b + \sqrt{b^2 - 4ac}}{2a}$. Therefore, we solve for $\frac{-b + \sqrt{b^2 - 4ac}}{2a} < 1$ as follows

$$\sqrt{b^2 - 4ac} < b + 2a \quad \Rightarrow \quad 4a\left(b + c\right) + 4a^2 > 0 \quad \Rightarrow \quad a + b + c > 0.$$

Now, rewriting the above inequality in terms of the actual coefficients, we get

$$1 - \phi_{11}^{(k)} - \phi_{22}^{(k)} + \phi_{11}^{(k)}\phi_{22}^{(k)} - \phi_{12}^{(k)}\phi_{21}^{(k)} > 0 \quad \Rightarrow \quad \left(1 - \phi_{11}^{(k)}\right)\left(1 - \phi_{22}^{(k)}\right) > \phi_{12}^{(k)}\phi_{21}^{(k)}. \quad (13)$$

Furthermore, note that $a = 1 > 0$, $b = -\left(\phi_{11}^{(k)} + \phi_{22}^{(k)}\right) < 0$ and $b^2 - 4ac = \left(\phi_{11}^{(k)} - \phi_{22}^{(k)}\right)^2 + 4\phi_{12}^{(k)}\phi_{21}^{(k)} > 0$ hold by definition, so we only need to check for

$$c = \phi_{11}^{(k)}\phi_{22}^{(k)} - \phi_{12}^{(k)}\phi_{21}^{(k)} > 0. \tag{14}$$

Equation 13 and equation 14 lay out the necessary conditions in order to get $\rho\left(\mathbf{\Phi}^{(k)}\right) < 1$.

**Step 2: Simplifying the conditions.** Starting off with the more important of the two, we first solve for equation 13. In order to simplify this inequality, we choose to have (i) $0 < \phi_{11}^{(k)} \leq 1$ and (ii) $0 < \phi_{22}^{(k)} \leq 1$ for the main diagonal entries. For $\phi_{11}^{(k)}$ as defined in Lemma 4.1, we have

$$\phi_{11}^{(k)} \leq 1 \quad \Rightarrow \quad \frac{1 + \mu\alpha^{(k)} - \left(\mu\alpha^{(k)}\right)^2}{1 + \mu\alpha^{(k)}} \geq \frac{2\beta^2}{\mu^2}\left(1 - d_{min}^{(k)}\right). \tag{15}$$

To better characterize the condition on $\alpha^{(k)}$ based on the above inequality, we put the following constraint on $\alpha^{(k)}$ to get

$$\text{Constraint 1: } \alpha^{(k)} \leq \frac{\Gamma_1^{(k)}}{\mu} \quad \Rightarrow \quad \mu\alpha^{(k)} \geq \frac{2\beta^2}{\mu^2}\left(1 - d_{min}^{(k)}\right)\left(1 + \Gamma_1^{(k)}\right) + \left(\Gamma_1^{(k)}\right)^2 - 1,$$

where $\Gamma_1^{(k)} > 0$ is a scalar. The above condition requires the step size $\alpha^{(k)}$ to be lower-bounded, which is something we want to avoid. Thus, if the right-hand side of the inequality is non-positive, this condition only requires us to choose a non-negative value for the step size, which is sensible. So, we have

$$\frac{2\beta^2}{\mu^2}\left(1 - d_{min}^{(k)}\right)\left(1 + \Gamma_1^{(k)}\right) + \left(\Gamma_1^{(k)}\right)^2 - 1 \leq 0$$

$$\Rightarrow \quad \left(\Gamma_1^{(k)}\right)^2 + \left(\frac{2\beta^2}{\mu^2}\left(1 - d_{min}^{(k)}\right)\right)\Gamma_1^{(k)} + \left(\frac{2\beta^2}{\mu^2}\left(1 - d_{min}^{(k)}\right) - 1\right) \leq 0$$

$$\Rightarrow \quad \left|\Gamma_1^{(k)} - \frac{\beta^2}{\mu^2}\left(1 - d_{min}^{(k)}\right)\right| \leq \left|\frac{\beta^2}{\mu^2}\left(1 - d_{min}^{(k)}\right) - 1\right|$$

$$\Rightarrow \quad \begin{cases} d_{min}^{(k)} < 1 - \frac{\mu^2}{\beta^2}: & 1 \leq \Gamma_1^{(k)} \leq \frac{2\beta^2}{\mu^2}\left(1 - d_{min}^{(k)}\right) - 1 \\ d_{min}^{(k)} > 1 - \frac{\mu^2}{\beta^2}: & \frac{2\beta^2}{\mu^2}\left(1 - d_{min}^{(k)}\right) - 1 \leq \Gamma_1^{(k)} \leq 1 \end{cases}$$

$$\Rightarrow \quad \min\left\{1, \frac{2\beta^2}{\mu^2}\left(1 - d_{min}^{(k)}\right) - 1\right\} \leq \Gamma_1^{(k)} \leq \max\left\{1, \frac{2\beta^2}{\mu^2}\left(1 - d_{min}^{(k)}\right) - 1\right\}.$$

We observe that we found a lower and upper bound for the choice of $\Gamma_1^{(k)}$. Note that this lower bound is not on the step size, as we only have $\alpha^{(k)} \leq \frac{\Gamma_1^{(K)}}{\mu}$.

Next, in order to simplify $\phi_{11}^{(k)}$ as defined in Lemma 4.1, we can use equation 15 to write

$$\text{Constraint 2: } \frac{1 + \mu\alpha^{(k)} - \left(\mu\alpha^{(k)}\right)^2}{1 + \mu\alpha^{(k)}} \geq \frac{2\beta^2}{\mu^2}\left(1 - d_{min}^{(k)}\right)\Gamma_2^{(k)},$$

in which $\Gamma_2^{(k)} \geq 1$ ensures that the constraint is satisfied, since we solved for equation 15 and found the conditions on $\alpha^{(k)}$ and $\Gamma_1^{(k)}$ to do so. Hence, for the bounds defined in Lemma 4.1, we get

$$\phi_{11}^{(k)} \leq 1 - \frac{2}{\mu}\left(1 + \Gamma_1^{(k)}\right)\left(\Gamma_2^{(k)} - 1\right)\left(1 - d_{min}^{(k)}\right)\beta^2\alpha^{(k)}, \qquad \phi_{12}^{(k)} \leq \frac{\left(1 + \Gamma_1^{(k)}\right)d_{max}^{(k)}\beta^2}{m\mu}\alpha^{(k)},$$

$$\psi_1^{(k)} \leq \frac{2\alpha^{(k)}}{\mu}\left(1 + \Gamma_1^{(k)}\right)\left(1 - d_{min}^{(k)}\right)\delta^2 + \frac{\left(\alpha^{(k)}\right)^2 d_{max}^{(k)}\sigma^2}{m},$$

and there are no changes to the upper bounds of $\phi_{21}^{(k)}$, $\phi_{22}^{(k)}$ and $\psi_2^{(k)}$, which were defined in Lemma 4.2. Note that matrix $\mathbf{\Phi}^{(k)}$ and vector $\mathbf{\Psi}^{(k)}$ in equation 6 were used as upper bounds, therefore we

can always replace their values with new upper bounds for them. Consequently, with this new value for $\phi_{11}^{(k)}$, we continue as

$$\phi_{11}^{(k)} > 0 \quad \Rightarrow \quad \alpha^{(k)} < \frac{\mu}{2\left(1 + \Gamma_1^{(k)}\right)\left(\Gamma_2^{(k)} - 1\right)\left(1 - d_{min}^{(k)}\right)\beta^2}.$$

Finally, we check the next conditions on $\phi_{22}^{(k)}$ defined in Lemma 4.2, i.e., $0 < \phi_{22}^{(k)} \leq 1$. Note that for $\phi_{11}^{(k)}$, $\phi_{12}^{(k)}$ and $\phi_{21}^{(k)}$ the lower bound is 0, but for $\phi_{22}^{(k)}$ it is $\frac{1+\tilde{\rho}^{(k)}}{2}$. Therefore, the lower-bound condition of $\phi_{22}^{(k)} > 0$ is already met. For the upper-bound condition $\phi_{22}^{(k)} \leq 1$, noting that we have $\frac{3+\tilde{\rho}^{(k)}}{4} < 1$, we can write $\phi_{22}^{(k)} \leq \frac{3+\tilde{\rho}^{(k)}}{4}$ to enforce this constraint. We have

$$\frac{1 + \tilde{\rho}^{(k)}}{2} \leq \phi_{22}^{(k)} \leq \frac{3 + \tilde{\rho}^{(k)}}{4} \quad \Rightarrow \quad 0 \leq \alpha^{(k)} \leq \frac{1}{2\sqrt{3d_{max}^{(k)}}}\frac{1 - \tilde{\rho}^{(k)}}{\sqrt{1 + \tilde{\rho}^{(k)}}}\frac{1}{\sqrt{\zeta^2 + 2\beta^2}}$$

**Step 3: Determining the constraints.** Now that we have made sure that (i) $0 < \phi_{11}^{(k)} \leq 1$ and (ii) $0 < \phi_{22}^{(k)} \leq 1$ in the previous step, we can continue to solve equation 13. For the left-hand side of the inequality, we have

$$\left(1 - \phi_{11}^{(k)}\right)\left(1 - \phi_{22}^{(k)}\right) = \left[\frac{2}{\mu}\left(1 + \Gamma_1^{(k)}\right)\left(\Gamma_2^{(k)} - 1\right)\left(1 - d_{min}^{(k)}\right)\beta^2\alpha^{(k)}\right]\left(1 - \phi_{22}^{(k)}\right)$$

$$\geq \left[\frac{2}{\mu}\left(1 + \Gamma_1^{(k)}\right)\left(\Gamma_2^{(k)} - 1\right)\left(1 - d_{min}^{(k)}\right)\beta^2\alpha^{(k)}\right]\frac{1 - \tilde{\rho}^{(k)}}{4}.$$

Now, putting this back to equation 13, we get

$$\left[\frac{2}{\mu}\left(1 + \Gamma_1^{(k)}\right)\left(\Gamma_2^{(k)} - 1\right)\left(1 - d_{min}^{(k)}\right)\beta^2\alpha^{(k)}\right]\frac{1 - \tilde{\rho}^{(k)}}{4} > \phi_{12}^{(k)}\phi_{21}^{(k)}$$

$$\Rightarrow \quad \left[\frac{\left(1 + \Gamma_1^{(k)}\right)d_{max}^{(k)}\beta^2}{m\mu}\alpha^{(k)}\right]\left[3md_{max}^{(k)}\frac{1 + \tilde{\rho}^{(k)}}{1 - \tilde{\rho}^{(k)}}\left(\zeta^2 + 2\beta^2\left(1 - d_{min}^{(k)}\right)\right)\left(\alpha^{(k)}\right)^2\right]$$

$$< \left[\frac{2}{\mu}\left(1 + \Gamma_1^{(k)}\right)\left(\Gamma_2^{(k)} - 1\right)\left(1 - d_{min}^{(k)}\right)\beta^2\alpha^{(k)}\right]\frac{1 - \tilde{\rho}^{(k)}}{4}$$

$$\Rightarrow \quad \alpha^{(k)} < \frac{\sqrt{\Gamma_2^{(k)} - 1}\sqrt{1 - d_{min}^{(k)}}\left(1 - \tilde{\rho}^{(k)}\right)}{\sqrt{6}d_{max}^{(k)}\sqrt{1 + \tilde{\rho}^{(k)}}\sqrt{\zeta^2 + 2\beta^2\left(1 - d_{min}^{(k)}\right)}}.$$

Finally, we solve for equation 14. Noting that by solving equation 13 we made sure that $1 - \phi_{11}^{(k)} - \phi_{22}^{(k)} + \phi_{11}^{(k)}\phi_{22}^{(k)} - \phi_{12}^{(k)}\phi_{21}^{(k)} > 0$, we can write

$$c > 0 \quad \Rightarrow \quad \phi_{11}^{(k)}\phi_{22}^{(k)} - \phi_{12}^{(k)}\phi_{21}^{(k)} > 0 \quad \Rightarrow \quad \phi_{11}^{(k)} + \phi_{22}^{(k)} - 1 > 0$$

$$\Rightarrow \quad 1 - \frac{2}{\mu}\left(1 + \Gamma_1^{(k)}\right)\left(\Gamma_2^{(k)} - 1\right)\left(1 - d_{min}^{(k)}\right)\beta^2\alpha^{(k)} + \frac{1 + \tilde{\rho}^{(k)}}{2} - 1 > 0$$

$$\Rightarrow \quad \alpha^{(k)} < \frac{\mu\left(1 + \tilde{\rho}^{(k)}\right)}{4\left(1 + \Gamma_1^{(k)}\right)\left(\Gamma_2^{(k)} - 1\right)\left(1 - d_{min}^{(k)}\right)\beta^2}.$$

**Step 4: Putting all the constraints together.** Reviewing all the constraints on $\alpha^{(k)}$ from the beginning of this appendix, we can collect all of the constraints together and simplify them as

$$\alpha^{(k)} < \min \left\{ \frac{\Gamma_1^{(k)}}{\mu}, \frac{1}{2\sqrt{3d_{max}^{(k)}}} \frac{1-\tilde{\rho}^{(k)}}{\sqrt{1+\tilde{\rho}^{(k)}}} \frac{1}{\sqrt{\zeta^2+2\beta^2}}, \frac{\mu}{2\left(1+\Gamma_1^{(k)}\right)\left(\Gamma_2^{(k)}-1\right)\left(1-d_{min}^{(k)}\right)\beta^2}, \right.$$
$$\frac{\mu\left(1+\tilde{\rho}^{(k)}\right)}{4\left(1+\Gamma_1^{(k)}\right)\left(\Gamma_2^{(k)}-1\right)\left(1-d_{min}^{(k)}\right)\beta^2},$$
$$\left. \frac{\sqrt{\Gamma_2^{(k)}-1}\sqrt{1-d_{min}^{(k)}}\left(1-\tilde{\rho}^{(k)}\right)}{\sqrt{6}d_{max}^{(k)}\sqrt{1+\tilde{\rho}^{(k)}}\sqrt{\zeta^2+2\beta^2\left(1-d_{min}^{(k)}\right)}} \right\}$$
$$= \min \left\{ \frac{\Gamma_1^{(k)}}{\mu}, \frac{1}{2\sqrt{3d_{max}^{(k)}}} \frac{1-\tilde{\rho}^{(k)}}{\sqrt{1+\tilde{\rho}^{(k)}}} \frac{1}{\sqrt{\zeta^2+2\beta^2}}, \frac{\mu\left(1+\tilde{\rho}^{(k)}\right)}{4\left(1+\Gamma_1^{(k)}\right)\left(\Gamma_2^{(k)}-1\right)\left(1-d_{min}^{(k)}\right)\beta^2}, \right.$$
$$\left. \frac{\sqrt{\Gamma_2^{(k)}-1}\sqrt{1-d_{min}^{(k)}}\left(1-\tilde{\rho}^{(k)}\right)}{\sqrt{6}d_{max}^{(k)}\sqrt{1+\tilde{\rho}^{(k)}}\sqrt{\zeta^2+2\beta^2\left(1-d_{min}^{(k)}\right)}} \right\},$$

while satisfying

$$\Gamma_1^{(k)} > 0, \qquad \min\left\{1, \frac{2\beta^2}{\mu^2}\left(1-d_{min}^{(k)}\right)-1\right\} \le \Gamma_1^{(k)} \le \max\left\{1, \frac{2\beta^2}{\mu^2}\left(1-d_{min}^{(k)}\right)-1\right\}, \tag{16}$$
$$\Gamma_2^{(k)} > 1.$$

Note that one of the terms in the above minimization function was trivially removed since $\frac{1+\tilde{\rho}^{(k)}}{2} < 1$. In order to simply the condition on $\alpha^{(k)}$ further, we take the minimum of these terms with respect to each variable separately to get

$$\alpha^{(k)} < \min \left\{ \min_{\Gamma_1^{(k)}} \left\{ \frac{\Gamma_1^{(k)}}{\mu}, \min_{\Gamma_2^{(k)}} \left\{ \frac{\mu\left(1+\tilde{\rho}^{(k)}\right)}{4\left(1+\Gamma_1^{(k)}\right)\left(\Gamma_2^{(k)}-1\right)\left(1-d_{min}^{(k)}\right)\beta^2}, \right. \right. \right.$$
$$\left. \left. \left. \frac{\sqrt{\Gamma_2^{(k)}-1}\sqrt{1-d_{min}^{(k)}}\left(1-\tilde{\rho}^{(k)}\right)}{\sqrt{6}d_{max}^{(k)}\sqrt{1+\tilde{\rho}^{(k)}}\sqrt{\zeta^2+2\beta^2\left(1-d_{min}^{(k)}\right)}} \right\} \right\}, \right. \tag{17}$$
$$\left. \frac{1}{2\sqrt{3d_{max}^{(k)}}} \frac{1-\tilde{\rho}^{(k)}}{\sqrt{1+\tilde{\rho}^{(k)}}} \frac{1}{\sqrt{\zeta^2+2\beta^2}} \right\}.$$

Solving for the inner minimization in equation 17 first using $\Gamma_2^{(k)}$ by defining $c_1^{(k)} = \frac{\sqrt{1-d_{min}^{(k)}}\left(1-\tilde{\rho}^{(k)}\right)}{\sqrt{6}d_{max}^{(k)}\sqrt{1+\tilde{\rho}^{(k)}}\sqrt{\zeta^2+2\beta^2\left(1-d_{min}^{(k)}\right)}}$ and $c_2^{(k)} = \frac{4\left(1+\Gamma_1^{(k)}\right)\left(1-d_{min}^{(k)}\right)\beta^2}{\mu\left(1+\tilde{\rho}^{(k)}\right)}$, we have

$$\begin{cases} c_1^{(k)}\sqrt{\Gamma_2^{(k)}-1} \le \frac{1}{c_2^{(k)}\left(\Gamma_2^{(k)}-1\right)}; & 1 < \Gamma_2^{(k)} \le \Gamma_2^{\star(k)} \\ \frac{1}{c_2^{(k)}\left(\Gamma_2^{(k)}-1\right)} \le c_1^{(k)}\sqrt{\Gamma_2^{(k)}-1}; & \Gamma_2^{(k)} \ge \Gamma_2^{\star(k)} \end{cases}, \tag{18}$$

in which $\Gamma_2^{(k)} > 1$ is due to equation 16. We can see that in equation 18, one of the expressions is increasing with respect to $\Gamma_2^{(k)}$, and the other one is decreasing. Thus, we find the optimal value for

it $\Gamma_2^{\star(k)}$ as

$$\sqrt{\Gamma_2^{\star(k)} - 1}^3 = \frac{1}{c_1^{(k)} c_2^{(k)}} \quad \Rightarrow \quad \Gamma_2^{\star(k)} = \frac{1}{\left(c_1^{(k)} c_2^{(k)}\right)^{2/3}} + 1$$

$$\Rightarrow \Gamma_2^{\star(k)} = \left( \frac{\sqrt{6} d_{max}^{(k)} \sqrt{1 + \tilde{\rho}^{(k)}} \sqrt{\zeta^2 + 2\beta^2 \left(1 - d_{min}^{(k)}\right)}}{\sqrt{1 - d_{min}^{(k)}} \left(1 - \tilde{\rho}^{(k)}\right)} \frac{\mu \left(1 + \tilde{\rho}^{(k)}\right)}{4 \left(1 + \Gamma_1^{(k)}\right) \left(1 - d_{min}^{(k)}\right) \beta^2} \right)^{2/3} + 1$$

$$= \frac{(1 + \tilde{\rho}^{(k)}) \sqrt[3]{3} \sqrt[3]{\zeta^2 + 2\beta^2 \left(1 - d_{min}^{(k)}\right)}}{2 \left(1 - d_{min}^{(k)}\right)} \left( \frac{d_{max}^{(k)} \mu}{\left(1 - \tilde{\rho}^{(k)}\right) \left(1 + \Gamma_1^{(k)}\right) \beta^2} \right)^{2/3} + 1$$

In order to simplify the above value for $\Gamma_2^{\star(k)}$, we bound it as

$$\left( \frac{\sqrt{3} \mu \zeta d_{min}^{(k)}}{2 \left(1 + \Gamma_1^{(k)}\right) \beta} \right)^{2/3} + 1 < \Gamma_2^{\star(k)} < \left( \frac{3\sqrt{2} d_{max}^{(k)}}{\left(1 - d_{max}^{(k)}\right)^{3/2} \left(1 - \tilde{\rho}^{(k)}\right) \left(1 + \Gamma_1^{(k)}\right)} \right)^{2/3} + 1.$$

Note that if $d_{min}^{(k)} \to 0$ and $d_{max}^{(k)} \to 1$, then we would have $1 < \Gamma_2^{\star(k)} < \infty$. Choosing $\Gamma_2^{(k)} = \Gamma_2^{(k)\star}$, we get

$$\min_{\Gamma_2^{(k)}} \left\{ \frac{\mu \left(1 + \tilde{\rho}^{(k)}\right)}{4 \left(1 + \Gamma_1^{(k)}\right) \left(\Gamma_2^{(k)} - 1\right) \left(1 - d_{min}^{(k)}\right) \beta^2}, \frac{\sqrt{\Gamma_2^{(k)} - 1} \sqrt{1 - d_{min}^{(k)}} \left(1 - \tilde{\rho}^{(k)}\right)}{\sqrt{6} d_{max}^{(k)} \sqrt{1 + \tilde{\rho}^{(k)}} \sqrt{\zeta^2 + 2\beta^2 \left(1 - d_{min}^{(k)}\right)}} \right\}$$

$$\geq \frac{\mu \left(1 + \tilde{\rho}^{(k)}\right) \left(c_1^{(k)} c_2^{(k)}\right)^{2/3}}{4 \left(1 + \Gamma_1^{(k)}\right) \left(1 - d_{min}^{(k)}\right) \beta^2} = \left( \frac{\left(c_1^{(k)}\right)^2}{c_2^{(k)}} \right)^{1/3}$$

$$= \left( \frac{\mu}{6 (1 + \Gamma_1^{(k)}) \left(\zeta^2 + 2\beta^2 \left(1 - d_{min}^{(k)}\right)\right)} \right)^{1/3} \left( \frac{1 - \tilde{\rho}^{(k)}}{2 d_{max}^{(k)} \beta} \right)^{2/3}$$

Moving on to the second minimization in equation 17 using $\Gamma_1^{(k)}$, we note that finding the optimal value $\Gamma_1^{\star(k)}$ would be analytically cumbersome due to the conditions that need to be satisfied for it; First, $\Gamma_1^{(k)} > 0$, and second, $\min\left\{1, \frac{2\beta^2}{\mu^2} \left(1 - d_{min}^{(k)}\right) - 1\right\} \leq \Gamma_1^{(k)} \leq \max\left\{1, \frac{2\beta^2}{\mu^2} \left(1 - d_{min}^{(k)}\right) - 1\right\}$. Thus, in order to get a more intuitive upper bound for $\alpha^{(k)}$, we settle for a possible suboptimal value for it. If we choose $\Gamma_1^{(k)} = 1$ which is the only point satisfying the conditions in equation 16 and it also does not rely on the value of $d_{min}^{(k)}$, we get

$$\alpha^{(k)} < \min \left\{ \frac{1}{\mu}, \left( \frac{\mu}{12 \left(\zeta^2 + 2\beta^2 \left(1 - d_{min}^{(k)}\right)\right)} \right)^{1/3} \left( \frac{1 - \tilde{\rho}^{(k)}}{2 d_{max}^{(k)} \beta} \right)^{2/3}, \right.$$

$$\left. \frac{1}{2\sqrt{3 d_{max}^{(k)}}} \frac{1 - \tilde{\rho}^{(k)}}{\sqrt{1 + \tilde{\rho}^{(k)}}} \frac{1}{\sqrt{\zeta^2 + 2\beta^2}} \right\}.$$

**Step 5: Obtaining $\rho(\mathbf{\Phi}^{(k)})$.** We established $\rho(\mathbf{\Phi}^{(k)}) < 1$ in the previous steps. The last step is to determine what $\rho(\mathbf{\Phi}^{(k)})$ is. We have

$$
\rho\left(\mathbf{\Phi}^{(k)}\right) = \frac{-b + \sqrt{b^2 - 4ac}}{2a} = \frac{\phi_{11}^{(k)} + \phi_{22}^{(k)} + \sqrt{\left(\phi_{11}^{(k)} + \phi_{22}^{(k)}\right)^2 - 4\left(\phi_{11}^{(k)}\phi_{22}^{(k)} - \phi_{12}^{(k)}\phi_{21}^{(k)}\right)}}{2}
$$

$$
= \frac{\phi_{11}^{(k)} + \phi_{22}^{(k)} + \sqrt{\left(\phi_{11}^{(k)} - \phi_{22}^{(k)}\right)^2 + 4\phi_{12}^{(k)}\phi_{21}^{(k)}}}{2}
$$

$$
= \frac{1}{2}\Bigg[ 1 - \frac{2}{\mu}\left(1 + \Gamma_1^{(k)}\right)\left(\Gamma_2^{(k)} - 1\right)\left(1 - d_{min}^{(k)}\right)\beta^2\alpha^{(k)} + \frac{1 + \tilde{\rho}^{(k)}}{2}
$$
$$
+ 3\frac{1 + \tilde{\rho}^{(k)}}{1 - \tilde{\rho}^{(k)}}d_{max}^{(k)}\left(\alpha^{(k)}\right)^2\left(\zeta^2 + 2\beta^2\right) \Bigg]
$$
$$
+ \frac{1}{2}\Bigg[ \left(1 - \frac{2}{\mu}\left(1 + \Gamma_1^{(k)}\right)\left(\Gamma_2^{(k)} - 1\right)\left(1 - d_{min}^{(k)}\right)\beta^2\alpha^{(k)}\right.
$$
$$
\left. - \frac{1 + \tilde{\rho}^{(k)}}{2} - 3\frac{1 + \tilde{\rho}^{(k)}}{1 - \tilde{\rho}^{(k)}}d_{max}^{(k)}\left(\alpha^{(k)}\right)^2\left(\zeta^2 + 2\beta^2\right)\right)^2
$$
$$
+ 4\frac{\left(1 + \Gamma_1^{(k)}\right)d_{max}^{(k)}\beta^2}{m\mu}\alpha^{(k)}3\frac{1 + \tilde{\rho}^{(k)}}{1 - \tilde{\rho}^{(k)}}md_{max}^{(k)}\left(\alpha^{(k)}\right)^2\left(\zeta^2 + 2\beta^2\left(1 - d_{min}^{(k)}\right)\right) \Bigg]^{1/2}
$$

$$
= \frac{3 + \tilde{\rho}^{(k)}}{4} - \frac{1}{\mu}\left(1 + \Gamma_1^{(k)}\right)\left(\Gamma_2^{(k)} - 1\right)\left(1 - d_{min}^{(k)}\right)\beta^2\alpha^{(k)}
$$
$$
+ \frac{3}{2}\frac{1 + \tilde{\rho}^{(k)}}{1 - \tilde{\rho}^{(k)}}d_{max}^{(k)}\left(\zeta^2 + 2\beta^2\right)\left(\alpha^{(k)}\right)^2
$$
$$
+ \frac{1}{2}\Bigg[ \left(\frac{1 - \tilde{\rho}^{(k)}}{2} - \frac{2}{\mu}\left(1 + \Gamma_1^{(k)}\right)\left(\Gamma_2^{(k)} - 1\right)\left(1 - d_{min}^{(k)}\right)\beta^2\alpha^{(k)}\right.
$$
$$
\left. - 3\frac{1 + \tilde{\rho}^{(k)}}{1 - \tilde{\rho}^{(k)}}d_{max}^{(k)}\left(\zeta^2 + 2\beta^2\right)\left(\alpha^{(k)}\right)^2\right)^2
$$
$$
+ 12\frac{\left(1 + \Gamma_1^{(k)}\right)\beta^2}{\mu}\frac{1 + \tilde{\rho}^{(k)}}{1 - \tilde{\rho}^{(k)}}\left(d_{max}^{(k)}\right)^2\left(\zeta^2 + 2\beta^2\left(1 - d_{min}^{(k)}\right)\right)\left(\alpha^{(k)}\right)^3 \Bigg]^{1/2}
$$

## K  PROOF OF THEOREM 4.1

Note that by the properties of spectral radius, we have that $\mathbf{\Phi}\|\cdot\| \le \rho(\mathbf{\Phi})\|\cdot\|$. Now, using equation 7, we can write

$$
\begin{bmatrix} \mathbb{E}_{\mathbf{\Xi}^{(k)}}\left[\left\|\bar{\theta}^{(k+1)} - \theta^\star\right\|^2\right] \\ \mathbb{E}_{\mathbf{\Xi}^{(k)}}\left[\left\|\mathbf{\Theta}^{(k+1)} - \mathbf{1}_m\bar{\theta}^{(k+1)}\right\|^2\right] \end{bmatrix} \le \rho(\mathbf{\Phi})^{k+1}\begin{bmatrix} \left\|\bar{\theta}^{(0)} - \theta^\star\right\|^2 \\ \left\|\mathbf{\Theta}^{(0)} - \mathbf{1}_m\bar{\theta}^{(0)}\right\|^2 \end{bmatrix} + \sum_{r=1}^{k}\rho(\mathbf{\Phi})^{k-r+1}\mathbf{\Psi} + \mathbf{\Psi}.
$$

We emphasize that the time index $k$ in $\mathbf{\Phi}^{(k)}$ and $\mathbf{\Psi}^{(k)}$ was dropped, since we are using a constant step size, constant probability of SGDs and aggregations. This results in the constant matrix $\mathbf{\Phi}^{(k)} = \mathbf{\Phi}$ and the constant vector $\mathbf{\Psi}^{(k)} = \mathbf{\Psi}$. Focusing on the term $\sum_{r=1}^{k}\rho(\mathbf{\Phi})^{k-r+1}\mathbf{\Psi} + \mathbf{\Psi}$, we get

$$
\sum_{r=1}^{k}\rho(\mathbf{\Phi})^{k-r+1}\mathbf{\Psi} + \mathbf{\Psi} = \sum_{r=1}^{k+1}\rho(\mathbf{\Phi})^{k-r+1}\mathbf{\Psi} = \left(\sum_{u=0}^{k}\rho(\mathbf{\Phi})^u\right)\mathbf{\Psi} \le \left(\sum_{u=0}^{\infty}\rho(\mathbf{\Phi})^u\right)\mathbf{\Psi}
$$
$$
= \frac{1}{1 - \rho(\mathbf{\Phi})}\mathbf{\Psi}.
$$

Putting the above inequalities together concludes the proof of equation 8. Finally, noting that $\rho(\mathbf{\Phi}) < 1$ following 4.1, We can let $k \to \infty$ to get equation 9.

## L  DIMINISHING STEP SIZE POLICY

In this appendix, we do the convergence analysis of our methodology under a diminishing step size policy, i.e., when $\alpha^{(k+1)} < \alpha^{(k)}$ for all $k \geq 0$. We will show that convergence to the globally optimal point is possible if the frequency of SGDs, i.e., $d_i^{(k)}$, is increasing over time. Thus, a few preliminary lemmas are first required, to re-derive counterparts of Proposition 4.1 and Corollary 4.1 for the increasing $d_i^{(k)}$ strategy.

**Proposition L.1** *(See Appendix M for the proof.)* *If the probability of SGDs is chosen as $d_i^{(k)} = 1 - \Gamma_3 \alpha^{(0)}$ with $0 \leq \Gamma_3 \leq \frac{1}{\alpha^{(0)}}$, and the step size satisfies the following condition for all $k \geq 0$*

$$\alpha^{(k)} < \min \left\{ \frac{\Gamma_1^{(k)}}{\mu}, \frac{1}{2\sqrt{3}} \frac{1 - \tilde{\rho}^{(k)}}{\sqrt{1 + \tilde{\rho}^{(k)}}} \frac{1}{\sqrt{\zeta^2 + 2\beta^2}}, \right.$$
$$\left. \left( \frac{\mu}{6\left(\zeta^2 + 2\Gamma_3\Gamma_1^{(k)}\frac{\beta^2}{\mu}\right)\left(1 + \Gamma_1^{(k)}\right)} \right)^{1/3} \left(\frac{1 - \tilde{\rho}^{(k)}}{2\beta}\right)^{2/3} \right\}.$$

*then we have $\rho\left(\mathbf{\Phi}^{(k)}\right) < 1$ for all $k \geq 0$, in which $\rho(\cdot)$ denotes the spectral radius of a given matrix, and $\mathbf{\Phi}^{(k)}$ is given in the linear system of inequalities of equation 6. $\rho\left(\mathbf{\Phi}^{(k)}\right)$ is given by*

$\rho\left(\mathbf{\Phi}^{(k)}\right) = 1 - h(\alpha^{(k)})$, *where* $h(\alpha^{(k)}) = \frac{1 - \tilde{\rho}^{(k)}}{4} + A\alpha^{(k)} - B(\alpha^{(k)})^2 - \frac{1}{2}\sqrt{\left(\frac{1 - \tilde{\rho}^{(k)}}{2} - 2(A\alpha^{(k)} + B(\alpha^{(k)})^2)\right)^2 + C(\alpha^{(k)})^3}$, *and* $A = \frac{\Gamma_3\Gamma_1^{(k)}}{\mu}(1 + \Gamma_1^{(k)})(\Gamma_2^{(k)} - 1)\beta^2$, $B = \frac{3}{2}\frac{1 + \tilde{\rho}^{(k)}}{1 - \tilde{\rho}^{(k)}}(\zeta^2 + 2\beta^2)$, *and* $C = 12\frac{(1 + \Gamma_1^{(k)})\beta^2}{\mu}\frac{1 + \tilde{\rho}^{(k)}}{1 - \tilde{\rho}^{(k)}}(\zeta^2 + 2\Gamma_3\Gamma_1^{(k)}\frac{\beta^2}{\mu})$.

Proposition L.1 implies that $\lim_{k \to \infty} \mathbf{\Phi}^{(k:0)} = 0$ in equation 7. However, note that this is only the asymptotic behaviour of $\mathbf{\Phi}^{(k:0)}$, and the exact convergence rate will depend on the choice of the step size $\alpha^{(k)}$. Furthermore, noting that the first expression in equation 7 asymptotically approaches zero, Proposition L.1 also implies that the optimality gap is determined by the terms $\sum_{r=1}^{k} \mathbf{\Phi}^{(k:r)}\mathbf{\Psi}^{(r-1)} + \mathbf{\Psi}^{(k)}$, and it can be made zero if the step size $\alpha^{(k)}$ satisfies certain conditions, which we will discuss in Theorem L.1.

Proposition L.1 outlines the necessary constraint on the step size $\alpha^{(k)}$ at each iteration $k \geq 0$. We next provide a corollary to Proposition L.1, in which we show that under certain conditions, the above-mentioned constraint needs to be satisfied only on the initial value of the step size, i.e, $\alpha^{(0)}$.

**Corollary L.1** *If the step size $\alpha^{(k)}$ is non-increasing, i.e., $\alpha^{(k+1)} \leq \alpha^{(k)}$, the probability of SGDs are determined as $d_i^{(k)} = 1 - \Gamma_3 \alpha^{(k)}$ with $0 \leq \Gamma_3 \leq \frac{1}{\alpha^{(0)}}$, for all $k \geq 0$, and we have $\tilde{\rho}^{(k)} \leq \tilde{\rho} = \sup_{k=0,1,\dots} \tilde{\rho}^{(k)}$ for the spectral radius, then the constraints in Proposition L.1 simplify to*

$$\alpha^{(0)} < \min \left\{ \frac{\Gamma_1}{\mu}, \frac{1}{2\sqrt{3}} \frac{1 - \tilde{\rho}}{\sqrt{1 + \tilde{\rho}}} \frac{1}{\sqrt{\zeta^2 + 2\beta^2}}, \left( \frac{\mu}{6\left(\zeta^2 + 2\Gamma_3\Gamma_1\frac{\beta^2}{\mu}\right)\left(1 + \Gamma_1\right)} \right)^{1/3} \left(\frac{1 - \tilde{\rho}}{2\beta}\right)^{2/3} \right\}.$$

In the above Corollary, we obtained the constraints on the initial value of the step size, i.e., $\alpha^{(0)}$, that lead to the spectral radius of $\mathbf{\Phi}^{(k)}$ being less than 1, i.e., $\rho(\mathbf{\Phi}^{(k)}) < 1$. We need one more ingredient given in the subsequent lemma in order ultimately characterize the short-term behavior and also derive non-asymptotic convergence guarantees on `DSpodFL` in Theorem L.1.

Two final building blocks are necessary for the proof of Theorem L.1. We present these in the following lemmas.

**Lemma L.1** *(See Lemma 1 in Zehtabi et al. (2022b) for the proof.) Let $\{\zeta_r\}_{r=0}^{\infty}$ be a scalar sequence where $0 < \zeta_r \leq 1$, $\forall r \geq 0$. For any $p \geq 1$, we have*

$$\prod_{r=s}^{k} (1 - \zeta_r)^p \leq \frac{1}{p \sum_{r=s}^{k} \zeta_r}.$$

Next, we outline another crucial lemma for our analysis.

**Lemma L.2** *(See Appendix N for the proof.) Let a diminishing step size $\alpha^{(k)} = \alpha^{(0)}/\sqrt{1 + k/\gamma}$ be used, which satisfies the properties*

$$\alpha^{(k+1)} < \alpha^{(k)}, \qquad \sum_{k=0}^{\infty} \alpha^{(k)} = \infty, \qquad \sum_{k=0}^{\infty} \left(\alpha^{(k)}\right)^2 < \infty^3. \tag{19}$$

*Under the setup of Proposition L.1 for the probability of SGDs, i.e., $d_i^{(k)}$, if the probability of aggregations are constant, i.e., $b_{ij}^{(k)} = b_{ij}$ for all $(i, j) \in \mathcal{M}^2$, then the following bounds hold*

*(a)* $\sum_{q=r}^{k} \alpha^{(q)} \geq 2\alpha^{(0)}(\sqrt{1 + \frac{k}{\gamma}} - \sqrt{1 + \frac{r}{\gamma}})$,

*(b)* $\lim_{k \to \infty} \sum_{q=r}^{k} h(\alpha^{(q)}) \geq 4A\alpha^{(0)} \left(\sqrt{1 + \frac{k}{\gamma}} - \sqrt{1 + \frac{r}{\gamma}}\right)$,

*(c)* $\lim_{k \to \infty} \sum_{r=1}^{k} \frac{(\alpha^{(r-1)})^2}{\sum_{q=r}^{k} h(\alpha^{(q)})}$

$\leq \frac{\alpha^{(0)}}{4A} \left[ \frac{1}{\sqrt{1 + \frac{k}{\gamma}} - \sqrt{1 + \frac{1}{\gamma}}} + \frac{2(\gamma+1)}{\sqrt{1 + \frac{k}{\gamma}}} \left( \ln \sqrt{1 + \frac{k}{\gamma}} + \ln \frac{1}{\sqrt{1 + \frac{1}{\gamma+k-1}} - 1} \right) + \frac{2\sqrt{1 + \frac{k}{\gamma}}}{1 + \frac{k-1}{\gamma}} \right]$.

*where $h(\alpha^{(k)})$ and the constant $A$ were defined in Proposition L.1.*

Using Corollary L.1 and Lemmas L.1 and L.2, our main theorem follows.

**Theorem L.1** *(See Appendix O for the proof.) If a diminishing step size policy $\alpha^{(k)} = \alpha^{(0)}/\sqrt{1 + k/\gamma}$ with $\gamma > 0$ satisfying the conditions outlined in Corollary 4.1 is employed, and the probability of SGDs are all set to same value as $d_i^{(k)} = 1 - \alpha^{(k)}/\alpha^{(0)}$ for all $i \in \mathcal{M}$, while probability of aggregations are set to constant values, i.e., $b_{ij}^{(k)} = b_{ij}$ for all $(i, j) \in \mathcal{M}^2$, then we can rewrite equation 7 as*

$$\lim_{k \to \infty} \begin{bmatrix} \mathbb{E}_{\Xi^{(k)}} \left[ \left\| \bar{\theta}^{(k+1)} - \theta^\star \right\|^2 \right] \\ \mathbb{E}_{\Xi^{(k)}} \left[ \left\| \Theta^{(k+1)} - \mathbf{1}_m \bar{\theta}^{(k+1)} \right\|^2 \right] \end{bmatrix} \leq \mathcal{O}\left(\frac{1}{\sqrt{k}}\right) \begin{bmatrix} \left\| \bar{\theta}^{(0)} - \theta^\star \right\|^2 \\ \left\| \Theta^{(0)} - \mathbf{1}_m \bar{\theta}^{(0)} \right\|^2 \end{bmatrix}$$
$$+ \left(\alpha^{(0)}\right)^2 \left( 3\mathcal{O}\left(\frac{1}{\sqrt{k}}\right) + \mathcal{O}\left(\frac{\ln k}{\sqrt{k}}\right) + \mathcal{O}\left(\frac{1}{k}\right) \right) \begin{bmatrix} \frac{2}{\mu \alpha^{(0)}} \left(1 + \mu \alpha^{(0)}\right) \delta^2 + \frac{\sigma^2}{m} \\ m \left( 3\frac{1+\tilde{\rho}}{1-\tilde{\rho}} \delta^2 + \sigma^2 \right) \end{bmatrix}. \tag{20}$$

*Letting $k \to \infty$, we get*

$$\limsup_{k \to \infty} \begin{bmatrix} \mathbb{E}_{\Xi^{(k)}} \left[ \left\| \bar{\theta}^{(k+1)} - \theta^\star \right\|^2 \right] \\ \mathbb{E}_{\Xi^{(k)}} \left[ \left\| \Theta^{(k+1)} - \mathbf{1}_m \bar{\theta}^{(k+1)} \right\|^2 \right] \end{bmatrix} = 0. \tag{21}$$

---

[3]Note that the last condition implies $\lim_{k \to \infty} \alpha^{(k)} = 0$

The bound in equation 20 of Theorem L.1 indicates that by using a diminishing step size policy of $\alpha^{(k)} = \alpha^{(0)}/\sqrt{1 + k/\gamma}$, DSpodFL achieves a sub-linear convergence rate of $\mathcal{O}(\ln k/\sqrt{k})$, and equation 21 shows that asymptotic zero optimality gap as $k \to \infty$ can be achieved.

However, it is worth noting that choosing the probability of SGDs based on the step size, i.e., $d_i^{(k)} = 1 - \alpha^{(k)}/\alpha^{(0)}$ for all $i \in \mathcal{M}$ and $k \geq 0$, is only of theoretical value in this paper. This is because our motivation of introducing the notion of probability of SGDs was to capture computational capabilities of heterogeneous devices in real-world settings, therefore, it is an independent uncontrollable parameter and cannot be chosen based on the step size.

Finally, note that setting $d_i^{(k)} = 1 - \alpha^{(k)}/\alpha^{(0)}$ is equivalent to having all devices in the decentralized system to conduct SGD at each iteration as $k \to \infty$. This result is akin to Wang & Nedic (2022), in which an increasing similarity between the step sizes of devices is needed for convergence, despite them being initially uncoordinated.

## M  PROOF OF PROPOSITION L.1

Let the probability of SGDs $d_i^{(k)}$ be chosen as the following for all $i \in \mathcal{M}$:

$$d_i^{(k)} = 1 - \Gamma_3 \alpha^{(k)}, \qquad 0 \leq \Gamma_3 \leq \frac{1}{\alpha^{(0)}},$$

where $\alpha^{(k)}$ is the step size with a diminishing policy, i.e., $\lim_{k\to\infty} \alpha^{(k)} = 0$. Note that $d_i^{(0)} = 1 - \Gamma_3 \alpha^{(0)}$ and $\lim_{k\to\infty} d_i^{(k)} = 1$, which means that all devices will basically do SGDs at every iteration for large enough values of $k$. Based on this relationship that we put between the probability of SGDs and the step size, we first rewrite the bounds for matrices $\mathbf{\Phi}^{(k)}$ and $\mathbf{\Psi}^{(k)}$ which were given in Lemmas 4.1 and 4.2. We have

$$\phi_{11}^{(k)} = 1 - \mu\alpha^{(k)}\left(1 + \mu\alpha^{(k)} - \left(\mu\alpha^{(k)}\right)^2\right) + \frac{2\Gamma_3\left(\alpha^{(k)}\right)^2}{\mu}\left(1 + \mu\alpha^{(k)}\right)\beta^2,$$

$$\phi_{12}^{(k)} = \left(1 + \mu\alpha^{(k)}\right)\left(1 - \Gamma_3\alpha^{(k)}\right)\frac{\alpha^{(k)}\beta^2}{m\mu},$$

$$\phi_{21}^{(k)} = 3\frac{1 + \tilde{\rho}^{(k)}}{1 - \tilde{\rho}^{(k)}}m\left(1 - \Gamma_3\alpha^{(k)}\right)\left(\alpha^{(k)}\right)^2\left(\zeta^2 + 2\beta^2\Gamma_3\alpha^{(k)}\right)$$

$$\phi_{22}^{(k)} = \frac{1 + \tilde{\rho}^{(k)}}{2} + 3\frac{1 + \tilde{\rho}^{(k)}}{1 - \tilde{\rho}^{(k)}}\left(1 - \Gamma_3\alpha^{(k)}\right)\left(\alpha^{(k)}\right)^2\left(\zeta^2 + 2\beta^2\right),$$

$$\psi_1^{(k)} = \left(\alpha^{(k)}\right)^2\left[\frac{2\Gamma_3}{\mu}\left(1 + \mu\alpha^{(k)}\right)\delta^2 + \left(1 - \Gamma_3\alpha^{(k)}\right)\frac{\sigma^2}{m}\right],$$

$$\psi_2^{(k)} = m\left(1 - \Gamma_3\alpha^{(k)}\right)\left(\alpha^{(k)}\right)^2\left(3\frac{1 + \tilde{\rho}^{(k)}}{1 - \tilde{\rho}^{(k)}}\delta^2 + \sigma^2\right).$$

$$(22)$$

The important difference with the terms in equation 22 and the corresponding ones outlined in Lemmas 4.1 and 4.2 is the fact that we get a $\left(\alpha^{(k)}\right)^2$ factor for $\psi_1^{(k)}$ and $\psi_2^{(k)}$. This factor will help us show in Theorem L.1 that zero optimality gap can be reached, which follows mainly from equation 19.

Next, we do an analysis similar to the proof of Proposition 4.1, which was given in Appendix J.

**Step 1: Setting up the proof.** We skip repeating the explanations for this step, as they are exactly the same as step 1 in Appendix J.

**Step 2: Simplifying the conditions.** Recall that we have to ensure (i) $0 < \phi_{11}^{(k)} \leq 1$ and (ii) $0 < \phi_{22}^{(k)} \leq 1$. For $\phi_{11}^{(k)}$ as defined in equation 22, we have

$$\phi_{11}^{(k)} \leq 1 \qquad \Rightarrow \qquad \frac{1 + \mu\alpha^{(k)} - \left(\mu\alpha^{(k)}\right)^2}{\left(1 + \mu\alpha^{(k)}\right)\alpha^{(k)}} \geq \frac{2\beta^2\Gamma_3}{\mu^2}. \qquad (23)$$

We then put the following constraint on $\alpha^{(k)}$ to get a tighter lower bound for equation 23. We have

$$\text{Constraint 1: } \alpha^{(k)} \leq \frac{\Gamma_1^{(k)}}{\mu}, \qquad \Rightarrow \qquad \mu\alpha^{(k)} > \frac{2\beta^2\Gamma_3}{\mu^2}\Gamma_1^{(k)}\left(1+\Gamma_1^{(k)}\right) + \left(\Gamma_1^{(k)}\right)^2 - 1,$$

where $\Gamma_1^{(k)} > 0$ is a scalar. In order to avoid a positive lower-bound on the step size $\alpha^{(k)}$, we find the conditions under which the right-hand side of the above inequality is negative. We have

$$\frac{2\beta^2\Gamma_3}{\mu^2}\Gamma_1^{(k)}\left(1+\Gamma_1^{(k)}\right) + \left(\Gamma_1^{(k)}\right)^2 - 1 < 0$$

$$\Rightarrow \qquad \left(1+\frac{2\beta^2\Gamma_3}{\mu^2}\right)\left(\Gamma_1^{(k)}\right)^2 + \frac{2\beta^2\Gamma_3}{\mu^2}\Gamma_1^{(k)} - 1 < 0,$$

$$\Rightarrow \qquad \left(\left(1+\frac{2\beta^2\Gamma_3}{\mu^2}\right)\Gamma_1^{(k)} - 1\right)\left(\Gamma_1^{(k)} + 1\right) < 0 \quad \Rightarrow \quad -1 < \Gamma_1^{(k)} < \frac{1}{1+\frac{2\beta^2\Gamma_3}{\mu^2}}.$$

Next, in order to simplify $\phi_{11}^{(k)}$ further, we add another constraint using equation 23 to parameterize the lower bound in equation 23. We have

$$\frac{1+\mu\alpha^{(k)} - \left(\mu\alpha^{(k)}\right)^2}{1+\mu\alpha^{(k)}} \geq \frac{2\beta^2\Gamma_3}{\mu^2}\Gamma_1^{(k)},$$

$$\text{Constraint 2: } \frac{1+\mu\alpha^{(k)} - \left(\mu\alpha^{(k)}\right)^2}{1+\mu\alpha^{(k)}} > \frac{2\beta^2\Gamma_3}{\mu^2}\Gamma_1^{(k)}\Gamma_2^{(k)}, \qquad \Gamma_2^{(k)} > 1,$$

in which $\Gamma_2^{(k)} > 1$ makes sure that the constraint in equation 23 is satisfied. Hence, we can update the entries of matrices $\boldsymbol{\Phi}^{(k)}$ and $\boldsymbol{\Psi}^{(k)}$ as

$$\phi_{11}^{(k)} \leq 1 - \frac{2\Gamma_3\Gamma_1^{(k)}}{\mu}\left(1+\Gamma_1^{(k)}\right)\left(\Gamma_2^{(k)} - 1\right)\beta^2\alpha^{(k)},$$

$$\phi_{12}^{(k)} \leq \frac{\left(1+\Gamma_1^{(k)}\right)\beta^2}{m\mu}\alpha^{(k)}, \qquad \phi_{21}^{(k)} \leq 3\frac{1+\tilde{\rho}^{(k)}}{1-\tilde{\rho}^{(k)}}m\left(\alpha^{(k)}\right)^2\left(\zeta^2 + 2\Gamma_3\Gamma_1^{(k)}\frac{\beta^2}{\mu}\right) \tag{24}$$

$$\phi_{22}^{(k)} \leq \frac{1+\tilde{\rho}^{(k)}}{2} + 3\frac{1+\tilde{\rho}^{(k)}}{1-\tilde{\rho}^{(k)}}\left(\alpha^{(k)}\right)^2\left(\zeta^2 + 2\beta^2\right),$$

$$\psi_1^{(k)} \leq \left(\alpha^{(k)}\right)^2\left[\frac{2\Gamma_3}{\mu}\left(1+\Gamma_1^{(k)}\right)\delta^2 + \frac{\sigma^2}{m}\right], \quad \psi_2^{(k)} \leq m\left(\alpha^{(k)}\right)^2\left(3\frac{1+\tilde{\rho}^{(k)}}{1-\tilde{\rho}^{(k)}}\delta^2 + \sigma^2\right).$$

Note that matrix $\boldsymbol{\Phi}^{(k)}$ and vector $\boldsymbol{\Psi}^{(k)}$ in equation 6 were used as upper bounds, therefore we can always replace their values with new upper bounds for them. Consequently, with this new value for $\phi_{11}^{(k)}$, we continue as

$$\phi_{11}^{(k)} > 0 \qquad \Rightarrow \qquad \alpha^{(k)} < \frac{\mu}{2\Gamma_3\Gamma_1^{(k)}\left(1+\Gamma_1^{(k)}\right)\left(\Gamma_2^{(k)} - 1\right)\beta^2}.$$

Finally, we check the next condition $0 < \phi_{22}^{(k)} \leq 1$. Noting that we have $\frac{3+\tilde{\rho}^{(k)}}{4} < 1$, we can enforce $\phi_{22}^{(k)} \leq 1$ by setting $\phi_{22}^{(k)} \leq \frac{3+\tilde{\rho}}{4}$. We have

$$\frac{1+\tilde{\rho}^{(k)}}{2} \leq \phi_{22}^{(k)} \leq \frac{3+\tilde{\rho}^{(k)}}{4} \qquad \Rightarrow \qquad 0 \leq \alpha^{(k)} \leq \frac{1}{2\sqrt{3}}\frac{1-\tilde{\rho}^{(k)}}{\sqrt{1+\tilde{\rho}^{(k)}}}\frac{1}{\sqrt{\zeta^2+2\beta^2}}.$$

**Step 3: Determining the constraints.** Having made sure that (i) $0 < \phi_{11}^{(k)} \leq 1$ and (ii) $0 < \phi_{22}^{(k)} \leq 1$ in the previous step, we can continue to solve equation 13. For the left-hand side of the inequality, we have

$$\left(1-\phi_{11}^{(k)}\right)\left(1-\phi_{22}^{(k)}\right) = \left[\frac{2\Gamma_3\Gamma_1^{(k)}}{\mu}\left(1+\Gamma_1^{(k)}\right)\left(\Gamma_2^{(k)}-1\right)\beta^2\alpha^{(k)}\right]\left(1-\phi_{22}^{(k)}\right)$$

$$\geq \left[\frac{2\Gamma_3\Gamma_1^{(k)}}{\mu}\left(1+\Gamma_1^{(k)}\right)\left(\Gamma_2^{(k)}-1\right)\beta^2\alpha^{(k)}\right]\frac{1-\tilde{\rho}^{(k)}}{4}$$

Now, putting this back to equation 13, we get

$$\left[\frac{2\Gamma_3\Gamma_1^{(k)}}{\mu}\left(1+\Gamma_1^{(k)}\right)\left(\Gamma_2^{(k)}-1\right)\beta^2\alpha^{(k)}\right]\frac{1-\tilde{\rho}^{(k)}}{4} > \phi_{12}^{(k)}\phi_{21}^{(k)}$$

$$\Rightarrow \quad \left[\frac{\left(1+\Gamma_1^{(k)}\right)\beta^2}{m\mu}\alpha^{(k)}\right]\left[3m\frac{1+\tilde{\rho}^{(k)}}{1-\tilde{\rho}^{(k)}}\left(\zeta^2+2\Gamma_3\Gamma_1^{(k)}\frac{\beta^2}{\mu}\right)\left(\alpha^{(k)}\right)^2\right]$$

$$< \left[\frac{2\Gamma_3\Gamma_1^{(k)}}{\mu}\left(1+\Gamma_1^{(k)}\right)\left(\Gamma_2^{(k)}-1\right)\beta^2\alpha^{(k)}\right]\frac{1-\tilde{\rho}^{(k)}}{4}$$

$$\Rightarrow \quad \alpha^{(k)} < \sqrt{\frac{\Gamma_3\Gamma_1^{(k)}\left(\Gamma_2^{(k)}-1\right)\left(1-\tilde{\rho}^{(k)}\right)^2}{6\left(1+\tilde{\rho}^{(k)}\right)\left(\zeta^2+2\Gamma_3\Gamma_1^{(k)}\frac{\beta^2}{\mu}\right)}}.$$

Finally, we solve for equation 14, i.e., $c = \phi_{11}^{(k)}\phi_{22}^{(k)} - \phi_{12}^{(k)}\phi_{21}^{(k)} > 0$. Noting that by solving equation 13 we made sure that $1 - \phi_{11}^{(k)} - \phi_{22}^{(k)} + \phi_{11}^{(k)}\phi_{22}^{(k)} - \phi_{12}^{(k)}\phi_{21}^{(k)} > 0$, we can write

$$c > 0 \quad \Rightarrow \quad \phi_{11}^{(k)} + \phi_{22}^{(k)} - 1 > 0$$

$$\Rightarrow \quad 1 - \frac{2\Gamma_3\Gamma_1^{(k)}}{\mu}\left(1+\Gamma_1^{(k)}\right)\left(\Gamma_2^{(k)}-1\right)\beta^2\alpha^{(k)} + \frac{1+\tilde{\rho}^{(k)}}{2} - 1 > 0$$

$$\Rightarrow \quad \alpha^{(k)} < \frac{\mu\left(1+\tilde{\rho}^{(k)}\right)}{4\Gamma_3\Gamma_1^{(k)}\left(1+\Gamma_1^{(k)}\right)\left(\Gamma_2^{(k)}-1\right)\beta^2},$$

in which we have used the value of $\phi_{11}^{(k)}$ itself, but the lower bound of $\phi_{22}^{(k)}$ in the second line.

**Step 4: Putting all the constraints together.** Reviewing all the constraints on $\alpha^{(k)}$ from the beginning of this appendix, we can collect all of the constraints together and simplify them as

$$\alpha^{(k)} < \min\left\{\frac{\Gamma_1^{(k)}}{\mu}, \frac{1}{2\sqrt{3}}\frac{1-\tilde{\rho}^{(k)}}{\sqrt{1+\tilde{\rho}^{(k)}}}\frac{1}{\sqrt{\zeta^2+2\beta^2}}, \frac{\mu}{2\Gamma_3\Gamma_1^{(k)}\left(1+\Gamma_1^{(k)}\right)\left(\Gamma_2^{(k)}-1\right)\beta^2},\right.$$

$$\left.\frac{\mu\left(1+\tilde{\rho}^{(k)}\right)}{4\Gamma_3\Gamma_1^{(k)}\left(1+\Gamma_1^{(k)}\right)\left(\Gamma_2^{(k)}-1\right)\beta^2}, \sqrt{\frac{\Gamma_3\Gamma_1^{(k)}\left(\Gamma_2^{(k)}-1\right)\left(1-\tilde{\rho}^{(k)}\right)^2}{6\left(1+\tilde{\rho}^{(k)}\right)\left(\zeta^2+2\Gamma_3\Gamma_1^{(k)}\frac{\beta^2}{\mu}\right)}}\right\}$$

$$= \min\left\{\frac{\Gamma_1^{(k)}}{\mu}, \frac{1}{2\sqrt{3}}\frac{1-\tilde{\rho}^{(k)}}{\sqrt{1+\tilde{\rho}^{(k)}}}\frac{1}{\sqrt{\zeta^2+2\beta^2}}, \frac{\mu\left(1+\tilde{\rho}^{(k)}\right)}{4\Gamma_3\Gamma_1^{(k)}\left(1+\Gamma_1^{(k)}\right)\left(\Gamma_2^{(k)}-1\right)\beta^2},\right.$$

$$\left.\sqrt{\frac{\Gamma_3\Gamma_1^{(k)}\left(\Gamma_2^{(k)}-1\right)\left(1-\tilde{\rho}^{(k)}\right)^2}{6\left(1+\tilde{\rho}^{(k)}\right)\left(\zeta^2+2\Gamma_3\Gamma_1^{(k)}\frac{\beta^2}{\mu}\right)}}\right\}, \tag{25}$$

while satisfying

$$\max\{-1,0\} = 0 < \Gamma_1^{(k)} < \min\left\{1, \frac{1}{1+\frac{2\beta^2\Gamma_3}{\mu^2}}\right\} = \frac{1}{1+\frac{2\beta^2\Gamma_3}{\mu^2}}, \qquad \Gamma_2^{(k)} > 1,$$

$$0 \le \Gamma_3 \le \frac{1}{\alpha^{(0)}}. \tag{26}$$

Note that one of the terms in equation 25 was trivially removed since $\frac{1+\tilde{\rho}^{(k)}}{2} < 1$. Consequently, we obtain

$$
\alpha^{(k)} < \min \left\{ \frac{\Gamma_1^{(k)}}{\mu}, \frac{1}{2\sqrt{3}} \frac{1-\tilde{\rho}^{(k)}}{\sqrt{1+\tilde{\rho}^{(k)}}} \frac{1}{\sqrt{\zeta^2+2\beta^2}}, \frac{\mu\left(1+\tilde{\rho}^{(k)}\right)}{4\Gamma_3\Gamma_1^{(k)}\left(1+\Gamma_1^{(k)}\right)\left(\Gamma_2^{(k)}-1\right)\beta^2}, \right.
$$
$$
\left. \sqrt{\frac{\Gamma_3\Gamma_1^{(k)}\left(\Gamma_2^{(k)}-1\right)\left(1-\tilde{\rho}^{(k)}\right)^2}{6\left(1+\tilde{\rho}^{(k)}\right)\left(\zeta^2+2\Gamma_3\Gamma_1^{(k)}\frac{\beta^2}{\mu}\right)}} \right\}
$$
$$
= \min_{\Gamma_1^{(k)}} \left\{ \frac{\Gamma_1^{(k)}}{\mu}, \frac{1}{2\sqrt{3}} \frac{1-\tilde{\rho}^{(k)}}{\sqrt{1+\tilde{\rho}^{(k)}}} \frac{1}{\sqrt{\zeta^2+2\beta^2}}, \min_{\Gamma_2^{(k)},\Gamma_3} \left\{ \frac{\mu\left(1+\tilde{\rho}^{(k)}\right)}{4\Gamma_3\Gamma_1^{(k)}\left(1+\Gamma_1^{(k)}\right)\left(\Gamma_2^{(k)}-1\right)\beta^2}, \right. \right.
$$
$$
\left. \left. \sqrt{\frac{\Gamma_3\Gamma_1^{(k)}\left(\Gamma_2^{(k)}-1\right)\left(1-\tilde{\rho}^{(k)}\right)^2}{6\left(1+\tilde{\rho}^{(k)}\right)\left(\zeta^2+2\Gamma_3\Gamma_1^{(k)}\frac{\beta^2}{\mu}\right)}} \right\} \right\},
$$
$$
\tag{27}
$$

First, we focus on minimizing the inner expression in equation 27 using $\Gamma_2^{(k)}$ by defining $c_1^{(k)} = \sqrt{\frac{\Gamma_3\Gamma_1^{(k)}\left(1-\tilde{\rho}^{(k)}\right)^2}{6\left(1+\tilde{\rho}^{(k)}\right)\left(\zeta^2+2\Gamma_3\Gamma_1^{(k)}\frac{\beta^2}{\mu}\right)}}$ and $c_2^{(k)} = \frac{4\Gamma_3\Gamma_1^{(k)}\left(1+\Gamma_1^{(k)}\right)\beta^2}{\mu\left(1+\tilde{\rho}^{(k)}\right)}$. We can see that one of the above expressions is increasing with respect to $\Gamma_2^{(k)}$, and the other one is decreasing. Thus, we have

$$
\begin{cases}
c_1^{(k)}\sqrt{\Gamma_2^{(k)}-1} \leq \frac{1}{c_2^{(k)}\left(\Gamma_2^{(k)}-1\right)}; & 1 < \Gamma_2^{(k)} \leq \Gamma_2^{\star(k)} \\
\frac{1}{c_2^{(k)}\left(\Gamma_2^{(k)}-1\right)} \leq c_1^{(k)}\sqrt{\Gamma_2^{(k)}-1}; & \Gamma_2^{(k)} \geq \Gamma_2^{\star(k)}.
\end{cases}
$$

in which $\Gamma_2^{(k)} > 1$ is due to equation 26. Hence, we find the optimal value for it, i.e., $\Gamma_2^{\star(k)}$, as

$$
\sqrt{\Gamma_2^{\star(k)}-1}^3 = \frac{1}{c_1^{(k)}c_2^{(k)}} \qquad \Rightarrow \qquad \Gamma_2^{\star(k)} = \frac{1}{\left(c_1^{(k)}c_2^{(k)}\right)^{2/3}} + 1
$$

$$
\Rightarrow \Gamma_2^{\star(k)} = \left( \sqrt{\frac{6\left(1+\tilde{\rho}^{(k)}\right)\left(\zeta^2+2\Gamma_3\Gamma_1^{(k)}\frac{\beta^2}{\mu}\right)}{\Gamma_3\Gamma_1^{(k)}\left(1-\tilde{\rho}^{(k)}\right)^2}} \frac{\mu\left(1+\tilde{\rho}^{(k)}\right)}{4\Gamma_3\Gamma_1^{(k)}\left(1+\Gamma_1^{(k)}\right)\beta^2} \right)^{2/3} + 1
$$

$$
= \frac{1+\tilde{\rho}^{(k)}}{2\Gamma_3\Gamma_1^{(k)}\beta} \left( \frac{3\left(\zeta^2+2\Gamma_3\Gamma_1^{(k)}\frac{\beta^2}{\mu}\right)}{\beta} \right)^{1/3} \left( \frac{\mu}{\left(1-\tilde{\rho}^{(k)}\right)\left(1+\Gamma_1^{(k)}\right)} \right)^{2/3} + 1
$$

We choose $\Gamma_2^{(k)} = \Gamma_2^{\star(k)}$ (see the explanation given in related step of Appendix J) to get

$$
\min_{\Gamma_2^{(k)}} \left\{ \frac{\mu\left(1+\tilde{\rho}^{(k)}\right)}{4\Gamma_3\Gamma_1^{(k)}\left(1+\Gamma_1^{(k)}\right)\left(\Gamma_2^{(k)}-1\right)\beta^2}, \sqrt{\frac{\Gamma_3\Gamma_1^{(k)}\left(\Gamma_2^{(k)}-1\right)\left(1-\tilde{\rho}^{(k)}\right)^2}{6\left(1+\tilde{\rho}^{(k)}\right)\left(\zeta^2+2\Gamma_3\Gamma_1^{(k)}\frac{\beta^2}{\mu}\right)}} \right\}
$$

$$
\geq \frac{\mu\left(1+\tilde{\rho}^{(k)}\right)\left(c_1^{(k)}c_2^{(k)}\right)^{2/3}}{4\Gamma_3\Gamma_1^{(k)}\left(1+\Gamma_1^{(k)}\right)\beta^2} = \left( \frac{\left(c_1^{(k)}\right)^2}{c_2^{(k)}} \right)^{1/3}
$$

$$
= \left( \frac{\mu}{6\left(\zeta^2+2\Gamma_3\Gamma_1^{(k)}\frac{\beta^2}{\mu}\right)\left(1+\Gamma_1^{(k)}\right)} \right)^{1/3} \left( \frac{1-\tilde{\rho}^{(k)}}{2\beta} \right)^{2/3}.
$$

Note that in the process of minimizing equation 27 over $\Gamma_2^{(k)}$, two out of the three dependencies on $\Gamma_3$, and two out of four dependencies on $\Gamma_1^{(k)}$ were removed. Hence, we get

$$
\alpha^{(k)} < \min \left\{ \frac{\Gamma_1^{(k)}}{\mu}, \frac{1}{2\sqrt{3}} \frac{1 - \tilde{\rho}^{(k)}}{\sqrt{1 + \tilde{\rho}^{(k)}}} \frac{1}{\sqrt{\zeta^2 + 2\beta^2}}, \right.
$$
$$
\left. \left( \frac{\mu}{6 \left( \zeta^2 + 2\Gamma_3 \Gamma_1^{(k)} \frac{\beta^2}{\mu} \right) \left( 1 + \Gamma_1^{(k)} \right)} \right)^{1/3} \left( \frac{1 - \tilde{\rho}^{(k)}}{2\beta} \right)^{2/3} \right\}. \tag{28}
$$

Finally, we make a remark that we do not minimize over $\Gamma_3$ here, as we take it as a given deterministic value based on the choice of $d_i^{(k)} = 1 - \Gamma_3 \alpha^{(k)}$.

**Step 5: Obtaining $\rho(\boldsymbol{\Phi}^{(k)})$.** We established $\rho(\boldsymbol{\Phi}^{(k)}) < 1$ in the previous steps. The last step is to determine what $\rho(\boldsymbol{\Phi}^{(k)})$ is. We have

$$
\rho\left(\boldsymbol{\Phi}^{(k)}\right) = \frac{-b + \sqrt{b^2 - 4ac}}{2a} = \frac{\phi_{11}^{(k)} + \phi_{22}^{(k)} + \sqrt{\left(\phi_{11}^{(k)} + \phi_{22}^{(k)}\right)^2 - 4\left(\phi_{11}^{(k)}\phi_{22}^{(k)} - \phi_{12}^{(k)}\phi_{21}^{(k)}\right)}}{2}
$$
$$
= \frac{\phi_{11}^{(k)} + \phi_{22}^{(k)} + \sqrt{\left(\phi_{11}^{(k)} - \phi_{22}^{(k)}\right)^2 + 4\phi_{12}^{(k)}\phi_{21}^{(k)}}}{2}
$$
$$
= \frac{1 - \frac{2\Gamma_3\Gamma_1^{(k)}}{\mu}\left(1 + \Gamma_1^{(k)}\right)\left(\Gamma_2^{(k)} - 1\right)\beta^2\alpha^{(k)} + \frac{1 + \tilde{\rho}^{(k)}}{2} + 3\frac{1 + \tilde{\rho}^{(k)}}{1 - \tilde{\rho}^{(k)}}\left(\alpha^{(k)}\right)^2\left(\zeta^2 + 2\beta^2\right)}{2}
$$
$$
+ \frac{1}{2}\left[ \left(1 - \frac{2\Gamma_3\Gamma_1^{(k)}}{\mu}\left(1 + \Gamma_1^{(k)}\right)\left(\Gamma_2^{(k)} - 1\right)\beta^2\alpha^{(k)} - \frac{1 + \tilde{\rho}^{(k)}}{2} \right. \right.
$$
$$
\left. - 3\frac{1 + \tilde{\rho}^{(k)}}{1 - \tilde{\rho}^{(k)}}\left(\alpha^{(k)}\right)^2\left(\zeta^2 + 2\beta^2\right)\right)^2
$$
$$
\left. + 4\frac{\left(1 + \Gamma_1^{(k)}\right)\beta^2}{m\mu}\alpha^{(k)}3\frac{1 + \tilde{\rho}^{(k)}}{1 - \tilde{\rho}^{(k)}}m\left(\alpha^{(k)}\right)^2\left(\zeta^2 + 2\Gamma_3\Gamma_1^{(k)}\frac{\beta^2}{\mu}\right) \right]^{1/2}
$$
$$
= \frac{3 + \tilde{\rho}^{(k)}}{4} - \frac{\Gamma_3\Gamma_1^{(k)}}{\mu}\left(1 + \Gamma_1^{(k)}\right)\left(\Gamma_2^{(k)} - 1\right)\beta^2\alpha^{(k)} + \frac{3}{2}\frac{1 + \tilde{\rho}^{(k)}}{1 - \tilde{\rho}^{(k)}}\left(\alpha^{(k)}\right)^2\left(\zeta^2 + 2\beta^2\right)
$$
$$
+ \frac{1}{2}\left[ \left( \frac{1 - \tilde{\rho}^{(k)}}{2} - \frac{2\Gamma_3\Gamma_1^{(k)}}{\mu}\left(1 + \Gamma_1^{(k)}\right)\left(\Gamma_2^{(k)} - 1\right)\beta^2\alpha^{(k)} \right. \right.
$$
$$
\left. - 3\frac{1 + \tilde{\rho}^{(k)}}{1 - \tilde{\rho}^{(k)}}\left(\alpha^{(k)}\right)^2\left(\zeta^2 + 2\beta^2\right)\right)^2
$$
$$
\left. + 12\frac{\left(1 + \Gamma_1^{(k)}\right)\beta^2}{\mu}\frac{1 + \tilde{\rho}^{(k)}}{1 - \tilde{\rho}^{(k)}}\left(\alpha^{(k)}\right)^3\left(\zeta^2 + 2\Gamma_3\Gamma_1^{(k)}\frac{\beta^2}{\mu}\right) \right]^{1/2}
$$
$$
= \frac{3 + \tilde{\rho}^{(k)}}{4} - A\alpha^{(k)} + B\left(\alpha^{(k)}\right)^2
$$
$$
+ \frac{1}{2}\sqrt{\left(\frac{1 - \tilde{\rho}^{(k)}}{2} - 2\left(A\alpha^{(k)} + B\left(\alpha^{(k)}\right)^2\right)\right)^2 + C\left(\alpha^{(k)}\right)^3}
$$

# N    PROOF OF LEMMA L.2

(a) Since $\alpha^{(k)} = \frac{\alpha^{(0)}}{\sqrt{1+\frac{k}{\gamma}}}$, we have

$$\sum_{q=r}^{k} \alpha^{(q)} = \sum_{q=r}^{k} \frac{\alpha^{(0)}}{\sqrt{1+\frac{q}{\gamma}}} \geq \int_{r}^{k} \frac{\alpha^{(0)}\,dx}{\sqrt{1+\frac{x}{\gamma}}} \geq 2\alpha^{(0)} \left( \sqrt{1+\frac{k}{\gamma}} - \sqrt{1+\frac{r}{\gamma}} \right).$$

(b) Based on the equation $\rho(\mathbf{\Phi}^{(k)}) = 1 - h(\alpha^{(k)})$ given in Proposition L.1, $h(\alpha^{(k)})$ was given as

$$h(\alpha^{(k)}) = \frac{1-\tilde{\rho}^{(k)}}{4} + A\alpha^{(k)} - B\left(\alpha^{(k)}\right)^2$$
$$- \frac{1}{2}\sqrt{\left(\frac{1-\tilde{\rho}^{(k)}}{2} - 2\left(A\alpha^{(k)} + B\left(\alpha^{(k)}\right)^2\right)\right)^2 + C\left(\alpha^{(k)}\right)^3}.$$

Further note that since we established $0 \leq \rho\left(\mathbf{\Phi}^{(k)}\right) < 1$ in Proposition L.1, it would mean $0 < h(\alpha^{(k)}) \leq 1$. Next, since we have $b_{ij}^{(k)} = b_{ij}$ for all $(i,j) \in \mathcal{M}^2$ and $k \geq 0$, it follows that $\tilde{\rho}^{(k)} = \tilde{\rho}$, where $\tilde{\rho}^{(k)}$ was defined in Lemma C.4-(c). Consequently, we have

$$\sum_{q=r}^{k} h(\alpha^{(q)}) \geq \frac{1-\tilde{\rho}}{4}(k-r+1) + A\sum_{q=r}^{k} \alpha^{(q)} - B\sum_{q=r}^{k}\left(\alpha^{(k)}\right)^2$$
$$- \frac{1}{2}\sqrt{k-r+1}\Bigg[ \frac{(1-\tilde{\rho})^2}{4}(k-r+1) - 2A(1-\tilde{\rho})\sum_{q=r}^{k}\alpha^{(q)}$$
$$+ \sum_{q=r}^{k}\Bigg[ 4\left(A^2\left(\alpha^{(q)}\right)^2 + B^2\left(\alpha^{(q)}\right)^4 + 2AB\left(\alpha^{(q)}\right)^3\right) - 2B(1-\tilde{\rho})\left(\alpha^{(q)}\right)^2$$
$$+ C\left(\alpha^{(q)}\right)^3\Bigg]\Bigg]^{1/2}$$

$$= \frac{1-\tilde{\rho}}{4}(k-r+1) + A\sum_{q=r}^{k}\alpha^{(q)} - B\sum_{q=r}^{k}\left(\alpha^{(k)}\right)^2$$
$$- \frac{1}{2}\sqrt{k-r+1}\Bigg[ \frac{(1-\tilde{\rho})^2}{4}(k-r+1) - 2A(1-\tilde{\rho})\sum_{q=r}^{k}\alpha^{(q)}$$
$$+ 2\left(2A^2 - B(1-\tilde{\rho})\right)\sum_{q=r}^{k}\left(\alpha^{(q)}\right)^2 + (8AB+C)\sum_{q=r}^{k}\left(\alpha^{(q)}\right)^3$$
$$+ 4B^2\sum_{q=r}^{k}\left(\alpha^{(q)}\right)^4\Bigg]^{1/2},$$

where Cauchy-Schwartz inequality was used to move the sum inside the square root term. Next, noting that $\sum_{k=0}^{\infty}\alpha^{(k)} = \infty$ but $\sum_{k=0}^{\infty}\left(\alpha^{(k)}\right)^2 < \infty$, we can take the limit of the above inequality

as $k \to \infty$ to get

$$
\begin{aligned}
\lim_{k \to \infty} \sum_{q=r}^{k} h(\alpha^{(q)}) &\geq \frac{1-\tilde{\rho}}{4}(k-r+1) + A \sum_{q=r}^{k} \alpha^{(q)} \\
&\qquad - \frac{1}{2}\sqrt{k-r+1}\sqrt{\frac{(1-\tilde{\rho})^2}{4}(k-r+1) - 2A(1-\tilde{\rho})\sum_{q=r}^{k} \alpha^{(q)}} \\
&= \frac{1-\tilde{\rho}}{4}(k-r+1) + A2\alpha^{(0)}\left(\sqrt{1+\frac{k}{\gamma}} - \sqrt{1+\frac{r}{\gamma}}\right) \\
&\qquad - \frac{1-\tilde{\rho}}{4}(k-r+1)\sqrt{1 - \frac{8A}{1-\tilde{\rho}}\frac{2\alpha^{(0)}}{k-r+1}\left(\sqrt{1+\frac{k}{\gamma}} - \sqrt{1+\frac{r}{\gamma}}\right)} \\
&\geq \frac{1-\tilde{\rho}}{4}(k-r+1) + 2A\alpha^{(0)}\left(\sqrt{1+\frac{k}{\gamma}} - \sqrt{1+\frac{r}{\gamma}}\right) \\
&\qquad - \frac{1-\tilde{\rho}}{4}(k-r+1)\left[1 - \frac{8A\alpha^{(0)}}{(1-\tilde{\rho})(k-r+1)}\left(\sqrt{1+\frac{k}{\gamma}} - \sqrt{1+\frac{r}{\gamma}}\right)\right] \\
&= 2A\alpha^{(0)}\left(\sqrt{1+\frac{k}{\gamma}} - \sqrt{1+\frac{r}{\gamma}}\right) + 2A\alpha^{(0)}\left(\sqrt{1+\frac{k}{\gamma}} - \sqrt{1+\frac{r}{\gamma}}\right) \\
&= 4A\alpha^{(0)}\left(\sqrt{1+\frac{k}{\gamma}} - \sqrt{1+\frac{r}{\gamma}}\right),
\end{aligned}
$$

in which Binomial approximation was used.

(c) First, we break the summation to two parts, i.e.,

$$
\sum_{r=1}^{k} \frac{(\alpha^{(r-1)})^2}{\sum_{q=r}^{k} h(\alpha^{(q)})} = \sum_{r=1}^{k-1} \frac{(\alpha^{(r-1)})^2}{\sum_{q=r}^{k} h(\alpha^{(q)})} + \frac{(\alpha^{(k-1)})^2}{h(\alpha^{(k)})}. \tag{29}
$$

On the other hand, note that using the definition of $h(\alpha^{(k)})$ as given in Proposition L.1, we have

$$
\begin{aligned}
\lim_{k \to \infty} h\left(\alpha^{(k)}\right) &= \frac{1-\tilde{\rho}^{(k)}}{4} + A\alpha^{(k)} - \frac{1}{2}\sqrt{\left(\frac{1-\tilde{\rho}^{(k)}}{2}\right)^2 - 2A\left(1-\tilde{\rho}^{(k)}\right)\alpha^{(k)}} \\
&= \frac{1-\tilde{\rho}^{(k)}}{4} + A\alpha^{(k)} - \frac{1-\tilde{\rho}^{(k)}}{4}\sqrt{1 - \frac{8A}{1-\tilde{\rho}^{(k)}}\alpha^{(k)}} \\
&= \frac{1-\tilde{\rho}^{(k)}}{4} + A\alpha^{(k)} - \frac{1-\tilde{\rho}^{(k)}}{4}\left(1 - \frac{4A}{1-\tilde{\rho}^{(k)}}\alpha^{(k)}\right) = 2A\alpha^{(k)}.
\end{aligned}
$$

Therefore, when taking the limit of equation 29, the second term becomes

$$
\lim_{k \to \infty} \frac{(\alpha^{(k-1)})^2}{h(\alpha^{(k)})} = \frac{(\alpha^{(k-1)})^2}{2A\alpha^{(k)}} = \frac{1}{2A}\frac{\frac{\alpha^{(0)2}}{1+\frac{k-1}{\gamma}}}{\frac{\alpha^{(0)}}{\sqrt{1+\frac{k}{\gamma}}}} = \frac{\alpha^{(0)}}{2A}\frac{\sqrt{1+\frac{k}{\gamma}}}{1+\frac{k-1}{\gamma}}.
$$

Now, moving on to the second term in equation 29, we can write

$$\lim_{k\to\infty}\sum_{r=1}^{k-1}\frac{\left(\alpha^{(r-1)}\right)^2}{\sum_{q=r}^k h(\alpha^{(q)})} = \sum_{r=1}^{\infty}\frac{\left(\alpha^{(r-1)}\right)^2}{\sum_{q=r}^{\infty} h(\alpha^{(q)})} = \sum_{r=1}^{\infty}\lim_{k\to\infty}\frac{\left(\alpha^{(r-1)}\right)^2}{\sum_{q=r}^k h(\alpha^{(q)})}$$

$$\leq \sum_{r=1}^{\infty}\lim_{k\to\infty}\frac{\left(\alpha^{(r-1)}\right)^2}{4A\alpha^{(0)}\left(\sqrt{1+\frac{k}{\gamma}}-\sqrt{1+\frac{r}{\gamma}}\right)} = \lim_{k\to\infty}\sum_{r=1}^{k-1}\frac{\frac{\left(\alpha^{(0)}\right)^2}{1+\frac{r-1}{\gamma}}}{4A\alpha^{(0)}\left(\sqrt{1+\frac{k}{\gamma}}-\sqrt{1+\frac{r}{\gamma}}\right)}$$

$$= \frac{\alpha^{(0)}}{4A}\lim_{k\to\infty}\sum_{r=1}^{k-1}\frac{1}{\left(1+\frac{r-1}{\gamma}\right)\left(\sqrt{1+\frac{k}{\gamma}}-\sqrt{1+\frac{r}{\gamma}}\right)}$$

$$\leq \frac{\alpha^{(0)}}{4A}\lim_{k\to\infty}\sum_{r=1}^{k-1}\frac{1+\frac{1}{\gamma}}{\left(1+\frac{r}{\gamma}\right)\left(\sqrt{1+\frac{k}{\gamma}}-\sqrt{1+\frac{r}{\gamma}}\right)}$$

$$\leq \frac{\alpha^{(0)}}{4A}\left[\frac{1}{\sqrt{1+\frac{k}{\gamma}}-\sqrt{1+\frac{1}{\gamma}}}+\int_1^{k-1}\frac{\left(1+\frac{1}{\gamma}\right)dx}{\left(1+\frac{x}{\gamma}\right)\left(\sqrt{1+\frac{k}{\gamma}}-\sqrt{1+\frac{x}{\gamma}}\right)}\right].$$

Focusing only on the integral and defining $u(x)=\sqrt{1+\frac{x}{\gamma}}$, we have

$$\int_1^{k-1}\frac{\left(1+\frac{1}{\gamma}\right)dx}{\left(1+\frac{x}{\gamma}\right)\left(\sqrt{1+\frac{k}{\gamma}}-\sqrt{1+\frac{x}{\gamma}}\right)} = \int_1^{k-1}\frac{2\gamma\left(1+\frac{1}{\gamma}\right)du(x)}{u(x)\left(u(k)-u(x)\right)}$$

$$= \int_1^{k-1}\frac{2\left(\gamma+1\right)}{u(k)}\left(\frac{1}{u(x)}+\frac{1}{u(k)-u(x)}\right)du(x)$$

$$= \frac{2\left(\gamma+1\right)}{u(k)}\left(\ln\frac{u(k-1)}{u(1)}+\ln\frac{u(k)-u(1)}{u(k)-u(k-1)}\right) = \frac{2\left(\gamma+1\right)}{u(k)}\ln\frac{u(k-1)u(k)}{u(k)-u(k-1)}$$

$$= \frac{2\left(\gamma+1\right)}{u(k)}\left(\ln u(k)+\ln\frac{1}{\frac{u(k)}{u(k-1)}-1}\right)$$

$$= \frac{2\left(\gamma+1\right)}{\sqrt{1+\frac{k}{\gamma}}}\left(\ln\sqrt{1+\frac{k}{\gamma}}+\ln\frac{1}{\sqrt{1+\frac{1}{\gamma+k-1}}-1}\right).$$

Putting everything back together concludes the proof.

## O    PROOF OF THEOREM L.1

First, using the fact that $\mathbf{\Phi}^{(k)}\|\cdot\| \leq \rho(\mathbf{\Phi^{(k)}})\|\cdot\|$ for each iteration $k\geq 0$, we can rewrite equation 7 to get

$$\begin{bmatrix}\mathbb{E}_{\mathbf{\Xi}^{(k)}}\left[\left\|\bar{\theta}^{(k+1)}-\theta^\star\right\|^2\right]\\\mathbb{E}_{\mathbf{\Xi}^{(k)}}\left[\left\|\mathbf{\Theta}^{(k+1)}-\mathbf{1}_m\bar{\theta}^{(k+1)}\right\|^2\right]\end{bmatrix} \leq \left(\prod_{q=0}^k\rho\left(\mathbf{\Phi}^{(q)}\right)\right)\begin{bmatrix}\left\|\bar{\theta}^{(0)}-\theta^\star\right\|^2\\\left\|\mathbf{\Theta}^{(0)}-\mathbf{1}_m\bar{\theta}^{(0)}\right\|^2\end{bmatrix}$$
$$+\sum_{r=1}^k\left(\prod_{q=r}^k\rho\left(\mathbf{\Phi}^{(q)}\right)\right)\left(\alpha^{(r-1)}\right)^2\begin{bmatrix}\frac{2}{\mu\alpha^{(0)}}\left(1+\mu\alpha^{(0)}\right)\delta^2+\frac{\sigma^2}{m}\\m\left(3\frac{1+\tilde{\rho}}{1-\tilde{\rho}}\delta^2+\sigma^2\right)\end{bmatrix}+\mathbf{\Psi}^{(k)},$$
(30)

where the $\mathbf{\Psi}^{(k)}$ matrix was written using equation 22.

Next, in order to obtain equation 20 when $k\to\infty$, we need to simplify each of the three terms in equation 30. The easiest one to show is the last term, i.e., $\mathbf{\Psi}^{(k)}$. Based on equation 22, both of its

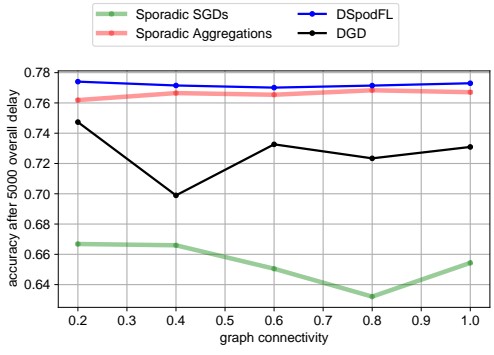 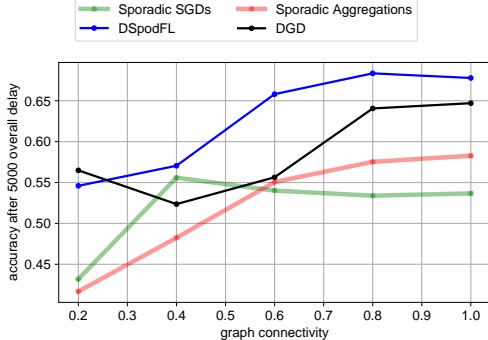

*(a) IID data split, with each device having data points from all of the 10 classes in the dataset.*

*(b) Non-IID data split, with each device having data points from only 1 of the classes in the dataset.*

Figure 3: Results of employing an SVM model on the Fashion-MNIST image classification dataset, with varying radius for the generated random geometric network graph.

entries $\psi_1^{(k)}$ and $\psi_2^{(k)}$ have a factor $\left(\alpha^{(k)}\right)^2$ multiplied by a value that can be upper-bounded by a constant. Thus, we have

$$\boldsymbol{\Psi}^{(k)} \leq \frac{\left(\alpha^{(0)}\right)^2}{1+\frac{k}{\gamma}} \left[ \frac{\frac{2\Gamma_3(1+\Gamma_1)}{\mu}\delta^2 + \frac{\sigma^2}{m}}{m\left(3\frac{1+\tilde{\rho}}{1-\tilde{\rho}}\delta^2 + \sigma^2\right)} \right].$$

Regarding the first and the second term, i.e., $\prod_{q=0}^{k} \rho\left(\boldsymbol{\Phi}^{(q)}\right)$ and $\sum_{r=1}^{k}\left(\prod_{q=r}^{k}\rho\left(\boldsymbol{\Phi}^{(q)}\right)\right)\left(\alpha^{(r-1)}\right)^2$, respectively, we have

$$\prod_{q=0}^{k} \rho\left(\boldsymbol{\Phi}^{(q)}\right) = \prod_{q=0}^{k}\left(1 - h(\alpha^{(q)})\right) \leq \frac{1}{\sum_{q=0}^{k} h(\alpha^{(q)})},$$

$$\sum_{r=1}^{k}\left(\prod_{q=r}^{k}\rho\left(\boldsymbol{\Phi}^{(q)}\right)\right)\left(\alpha^{(r-1)}\right)^2 = \sum_{r=1}^{k}\left(\prod_{q=r}^{k}\left(1 - h(\alpha^{(q)})\right)\right)\left(\alpha^{(r-1)}\right)^2 \leq \sum_{r=1}^{k}\frac{\left(\alpha^{(r-1)}\right)^2}{\sum_{q=r}^{k} h(\alpha^{(q)})},$$

in both of which Lemma L.1 was used, since $0 \leq h(\alpha^{(k)}) < 1$. Next, we employ Lemma L.2 on the above expressions. The proof easily follows. Note that $\ln \frac{1}{\sqrt{1+\frac{1}{\gamma+k-1}}-1} \leq \mathcal{O}(\ln k)$.

## P  FURTHER EXPERIMENTS

### P.1  EFFECT OF GRAPH CONNECTIVITY

We examine the effect of graph connectivity in this section. The underlying network topology that we use in this paper are generated via a random geometric graph with radius $r$ Penrose (2003). In the following numerical simulations, we compare our method with the baselines for different values of the radius, i.e., $r = 0.2, 0.4, 0.6, 0.8, 1$, where $r = 1$ corresponds to a fully-connected graph. We focus on the achieved accuracy after a specific amount of overall delay on the $y$-axis, for both cases of IID and non-IID data distributions among devices.

We have demonstrated the results for the IID case for the SVM model in Fig. 3a, and the CNN model in 4a. Notice how the graph connectivity does not have much effect on the outcome of any of the methods in the IID case. This is due to all devices already having data points from the distribution of the whole dataset, making inter-device communications insignificant. To express differently, if each device has access to the distribution of the whole dataset, it can solely conduct SGDs without any aggregations, and it would still perform notably well.

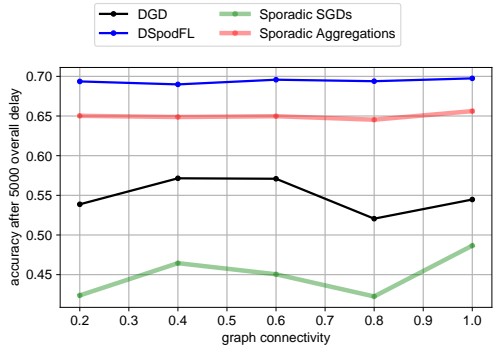 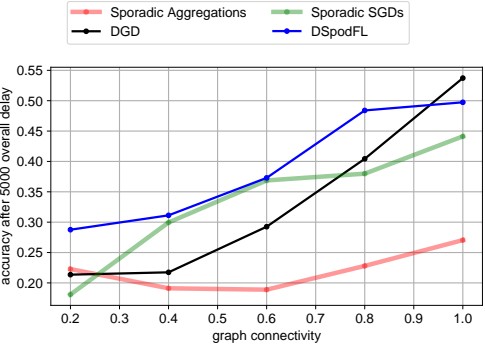

*(a) IID data split, with each device having data points from all of the 10 classes in the dataset.*

*(b) Non-IID data split, with each device having data points from only 1 of the classes in the dataset.*

Figure 4: *Results of employing a CNN model on the Fashion-MNIST image classification dataset, with varying radius for the generated random geometric network graph.*

However, we note that `DSpodFL` is able to outperform other baselines across all graph connectivity levels in Figs. 3a and 4a. Our method's efficiency becomes more prominent for the CNN model, as shown in 4a.

As illustrated in Fig. 3b for the SVM model and 4b for the CNN model, we see that graph connectivity plays a major role in performance of all methods in the non-IID case. We observe that higher graph connectivity helps devices in the fully-decentralized setup to reach convergence in less time. This is due to the fact that each device has access to only a portion of the distribution of the whole dataset, thus making inter-device communications very critical to reach the globally optimal solution.

### P.2 EFFECT OF DATA DISTRIBUTION

Next, we look into the effect of data distribution among devices, and compare our method with the baselines. We vary the number of labels from which each device gets samples. For example, if the number of labels is denoted as $k$ in Fig. 5, it means that each device has samples randomly chosen from $k$ of the classes in the whole dataset. Since the Fashion-MNIST dataset consists of 10 classes, we run our experiments with $k = 1, 2, ..., 10$, and look at the achieved accuracy per a fixed overall delay for all methods.

We observe that for both the SVM model in Fig. 5a and the CNN model in 5b, our `DSpodFL` method outperforms the baselines for any number of labels per devices. Moreover, an increasing trend with the number of labels can be seen for all algorithms, which is expected since with more labels, devices would have access to data points from the distribution of the whole dataset, and not just a portion of it.

Finally, the intriguing conclusion of this experiment is that for low number of labels per each device, i.e., the non-IID case, the Sporadic SGDs method outperforms Sporadic Aggregations method. This is because inter-device communication is critical for convergence in non-IID setups, and we cannot cut back on them arbitrarily. Therefore, Sporadic Aggregations perform very poorly in non-IID setups. Finally, note that this means in a low-label regime, our `DSpodFL` method benefits most from its Sporadic SGDs component in achieving resource efficiency.

On the other hand, however, when the number of labels per each device is high enough, Sporadic Aggregations tends to out perform Sporadic SGDs, due to the converse of the argument we made in the previous paragraph. This is clearly visualized in Fig. 5, where in 5a the two above-mentioned methods cross each other at 5 labels per device, and in 5b that happens in 3 labels per device.

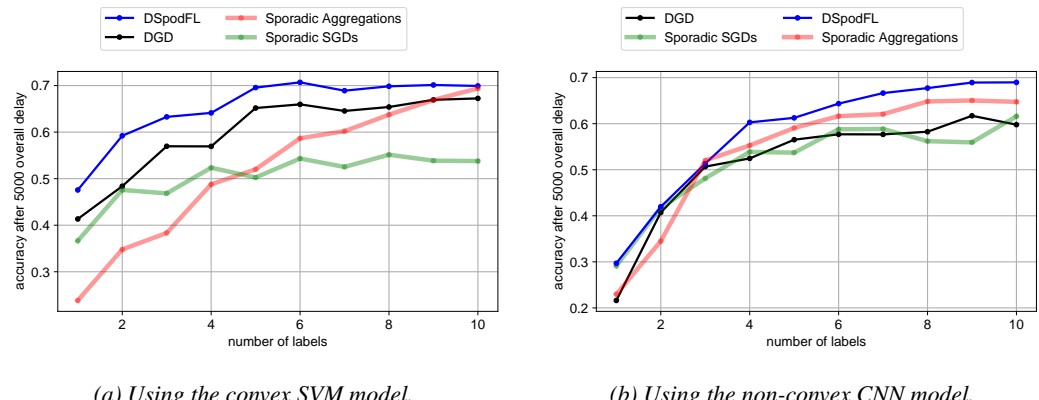

*(a) Using the convex SVM model.*  *(b) Using the non-convex CNN model.*

Figure 5: Results for the Fashion-MNIST image classification dataset, with varying number of labels per device.

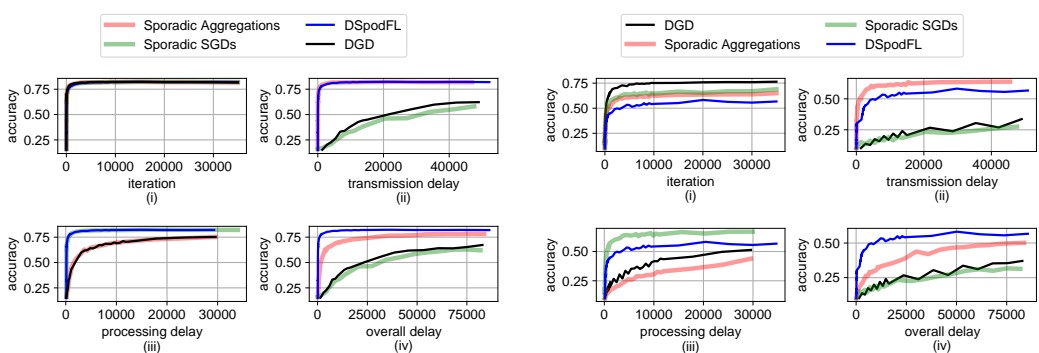

*(a) IID data split, with each device having data points from all of the 10 classes in the dataset.*  *(b) Non-IID data split, with each device having data points from only 1 of the classes in the dataset.*

Figure 6: Results of employing an SVM model on the Fashion-MNIST image classification dataset, with device and link capabilities coming from a beta distribution with $\alpha = \beta = 0.5$.

### P.3    EFFECT OF DEVICE AND LINK CAPABILITIES

In all of our experiments so far, the probability of SGDs and aggregations were sampled from a uniform distribution, e.g., $d_i^{(k)} \sim \mathcal{U}[0,1]$ and $b_{ij}^{(k)} \sim \mathcal{U}[0,1]$. In this section, we investigate sampling these probabilities from the beta distribution, denoted as $\text{Beta}(\alpha, \beta)$, where the values of $\alpha$ and $\beta$ determine its probability density function (PDF). Specifically, we choose $\alpha = \beta = 0.5$, which corresponds to an inverted bell curve PDF. In this regime, the probabilities are either extremely close to $1$ or extremely close to $0$.

We assign the probability of SGDs and aggregations as $d_i \sim \text{Beta}(0.5, 0.5)$ and $b_{ij} \sim \text{Beta}(0.5, 0.5)$, respectively. This regime is in accordance with practical settings where the devices/links in a decentralized network are either vastly abundant in resources, or are overly poor in them.

We have provided experimental results for both the IID and non-IID cases in Fig. 6 for the SVM model and Fig. 7 for the CNN model. It can be observed that the findings discussed in Sec. 5.2 also hold here. In other words, our `DSpodFL` method outperforms the baselines in terms of accuracy per overall delay. However, we observe that our method is able to compellingly surpass other methods when link and device resource capabilities come from $\text{Beta}(0.5, 0.5)$, compared to the uniform distribution as discussed in Sec. 5.

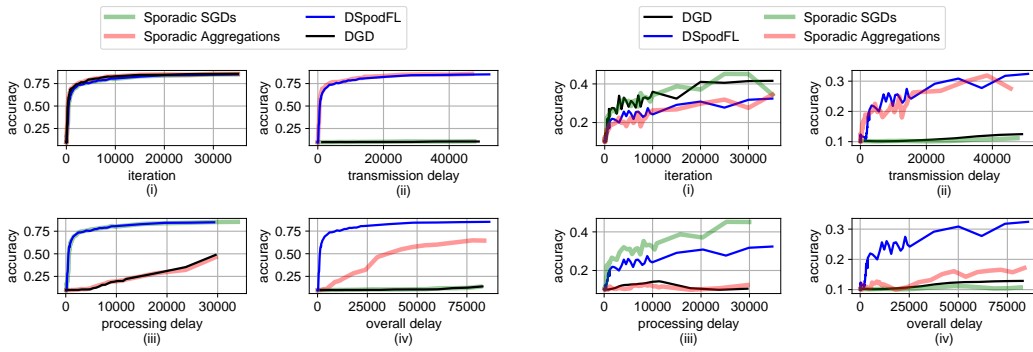

*(a) IID data split, with each device having data points from all of the 10 classes in the dataset.*

*(b) Non-IID data split, with each device having data points from only 1 of the classes in the dataset.*

Figure 7: Results of employing a CNN model on the Fashion-MNIST image classification dataset, with device and link capabilities coming from a beta distribution.

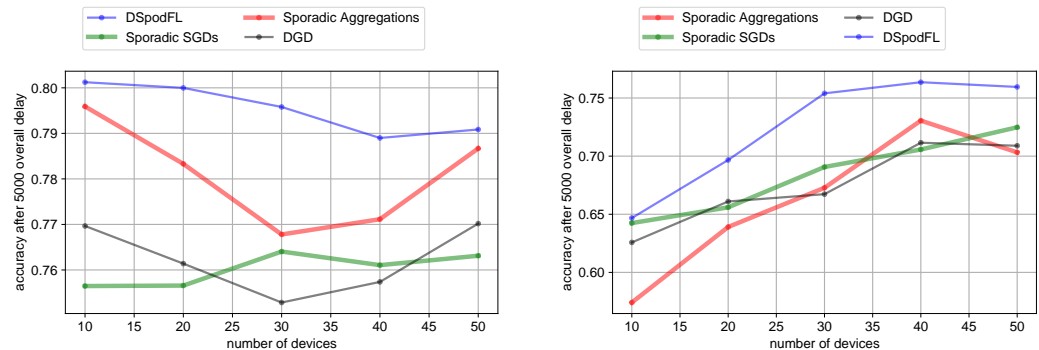

*(a) IID data split, with each device having data points from all of the 10 classes in the dataset.*

*(b) Non-IID data split, with each device having data points from only 1 of the classes in the dataset.*

Figure 8: Results of employing an SVM model on the Fashion-MNIST image classification dataset, with varying the number of devices in the network.

We will explain the intuitive reason for this drastic improvement with an example. Let $\mathcal{G} = (\mathcal{M}, \mathcal{E})$ be given a network graph, and assume there exits two paths between nodes $i$ and $j$. Let one of these paths have a communication cost $k$ times more than the other path, where $k \gg 1$. In our DSpodFL method, the path with lower cost will be utilized roughly $k$ times more than the other path, thus resulting in lower communication overhead while still preserving information flow between nodes $i$ and $j$. Meanwhile, the DGD method does not take this into account.

## P.4 EFFECT OF NUMBER OF DEVICES

Next, we do an ablation study on the number of devices, i.e., $m$. We compare DSpodFL with the baselines for $m = 10, 20, 30, 40, 50$. Similar to the previous subsections in this appendix, we plot the accuracy reached after a specific amount of total average delay. For each $m$, we generate a new random geometric graph with radius $r = 0.4$.

Figs. 8 and 9 showcase that our approach outperforms all other baselines regardless of the number of devices available in the system, with an SVM and a CNN model, respectively. Using either a

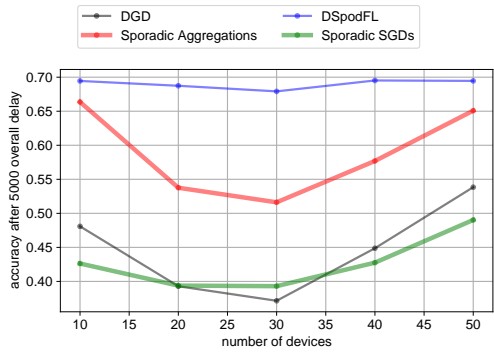 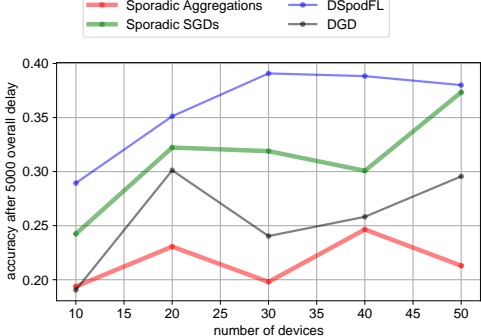

*(a) IID data split, with each device having data points from all of the* 10 *classes in the dataset.*

*(b) Non-IID data split, with each device having data points from only* 1 *of the classes in the dataset.*

*Figure 9: Results of employing a CNN model on the Fashion-MNIST image classification dataset, with varying the number of devices in the network.*

convex or a non-convex model, The superiority of our approach can be seen for both the IID case as illustrated in Figs. 8a and 9a, and the non-IID case as shown in Figs. 8b and 9b.

However, note that in the non-IID cases of Figs. 8b and 9b, increasing the number of devices from 10 to 50 results in an increase the achieved accuracy. This is because if the total number of devices in a decentralized network increases, the number of nodes containing data from a particular class in the dataset increases as well. In other words, there are multiple sources in the system where data distribution from each class in the dataset can propagate through the network.

## P.5 EFFECT OF PARAMETERS ALPHA AND BETA IN BETA DISTRIBUTION

As we discussed in Sec. P.3, sampling the probability of SGDs and aggregations, i.e., $d_i^{(k)}$ and $b_{ij}^{(k)}$, from the beta distribution $\text{Beta}(\alpha, \beta)$, will correspond to various real-world scenarios if we vary its parameters $\alpha$ and $\beta$. For example, we employed a uniform distribution $\mathcal{U}[0, 1]$ in Sec. 5, which is equivalent to using $\alpha = \beta = 1$ in the beta distribution. Moreover, the choice of $\alpha = \beta = 0.5$ in Sec. P.3 gives rise to an inverted bell curve PDF. In this section, we will analyze the effect of choosing different values for $\alpha$ and $\beta$ in the distribution, and compare our method's performance with other baselines in each scenario.

We have laid out the results for the SVM model In Fig. 10 and for the CNN model in Fig. 11. In all setups, we plot the achieved testing accuracy after 5000 overall delay units where $\alpha$ and $\beta$ in the beta distribution $\text{Beta}(\alpha, \beta)$ come from $\alpha, \beta = 0.5, 1, 2, 3, 4, 5$.

We observe that in all figures, our DSpodFL method outperforms other baselines for all values of $\alpha$ and $\beta$. Most notably, in setups with lower values of $\alpha$ for a fixed $\beta$, which corresponds to scenarios where most of the devices/links have low processing units/bandwidth, the efficiency of our method in converging faster than other baselines becomes more prominent.

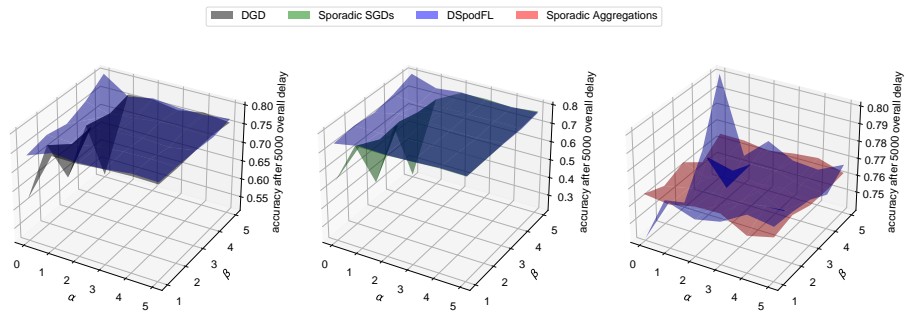

*(a) IID data split, with each device having data points from all of the 10 classes in the dataset.*

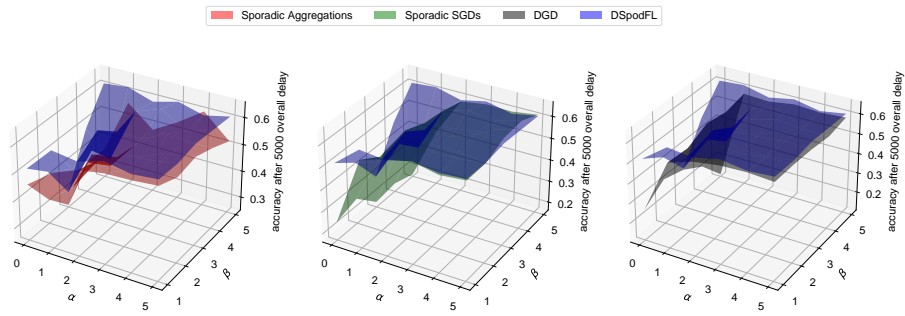

*(b) Non-IID data split, with each device having data points from only 1 of the classes in the dataset.*

Figure 10: *Results of employing a SVM model on the Fashion-MNIST image classification dataset, with device and link capabilities coming from a beta distribution* $\text{Beta}(\alpha, \beta)$ *with varying values for* $\alpha$ *and* $\beta$.

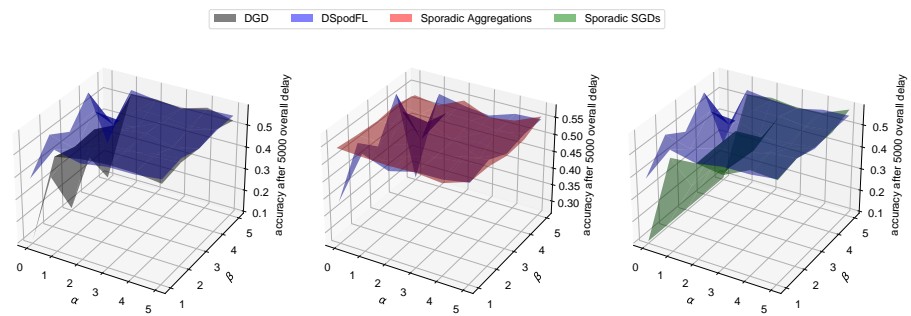

*(a) IID data split, with each device having data points from all of the 10 classes in the dataset.*

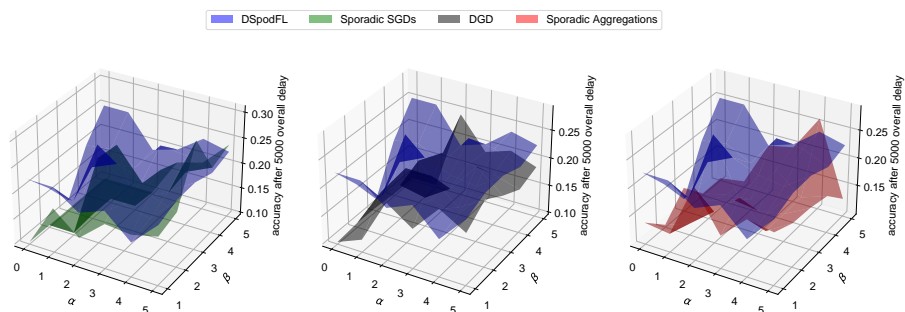

*(b) Non-IID data split, with each device having data points from only 1 of the classes in the dataset.*

*Figure 11: Results of employing a CNN model on the Fashion-MNIST image classification dataset, with device and link capabilities coming from a beta distribution ($\mathrm{Beta}(\alpha, \beta)$) with varying values for $\alpha$ and $\beta$.*