# OpenReview forum: "Sporadicity in Decentralized Federated Learning: Theory and Algorithm"
_ICLR.cc/2024/Conference — Submitted to ICLR 2024_

### Official Review · Reviewer_Mn6u · 2023-10-26

**Soundness:** 2 fair
**Presentation:** 2 fair
**Contribution:** 1 poor
**Rating:** 3
**Confidence:** 4

**Summary:**

This study expands the decentralized SGD method to accommodate sporadic cases in which nodes might randomly omit certain gradient computations or communications during iterations. The primary contribution lies in introducing this method and providing a convergence analysis for the strongly-convex setting.

**Strengths:**

Modelling and analysis of DSGD with random communication and computation skipping.

**Weaknesses:**

The idea of this work is incremental, and the literature review seems to omit pivotal and relevant works that address similar issues. Notably:

Srivastava, Kunal, and Angelia Nedic. "Distributed asynchronous constrained stochastic optimization." IEEE journal of selected topics in signal processing 5.4 (2011): 772-790.

Zhao, Xiaochuan, and Ali H. Sayed. "Asynchronous adaptation and learning over networks—Part I: Modeling and stability analysis." IEEE Transactions on Signal Processing 63.4 (2014): 811-826.

Lian, Xiangru, et al. "Asynchronous decentralized parallel stochastic gradient descent." International Conference on Machine Learning. PMLR, 2018.

Wang, Chengcheng, et al. "Coordinate-descent diffusion learning by networked agents." IEEE Transactions on Signal Processing 66.2 (2017): 352-367.

Liu, Wei, Li Chen, and Weidong Wang. "General decentralized federated learning for communication-computation tradeoff." IEEE INFOCOM 2022-IEEE Conference on Computer Communications Workshops (INFOCOM WKSHPS). IEEE, 2022.

The paper operates under the assumption of strong convexity. Furthermore, the present framework appears incremental and, due to the assumption of uncorrelated random variables, doesn't introduce significant challenges to the analysis.
The study doesn't showcase any theoretical advantages achieved through sporadic updates.

The pseudo code for the method should be incorporated into the main text. Clear definitions of the notation are necessary, such as those referenced in equation (3)
There's a lack of clarity and order in presenting equations and notations; they come across as disorganized.
The paper only offers asymptotic rates, neglecting to provide transient rates, especially where the error approaches zero, requiring the stepsize to diminish accordingly.

**Questions:**

How does the problem parameters influence the convergence bound, and how does this compare to the standard DSGD case? To understand this, we require a bound on $\rho(\Phi)$ that's expressed in terms of the network graph and random probabilities.
Does your analysis highlight any theoretical benefits from using sporadic updates?
I recommend deriving results for the nonconvex scenario and clearly comparing the bounds of the proposed method with those from other studies.

---

> ### Author Response · Authors · 2023-11-19
>
> ### **Related Work**
>
> Thanks a lot for pointing us towards these works. There are indeed a lot of papers that we were not able to reference in our first submission due to space constraints. However, we have included important papers that are cardinal to our work. There is a great overlap of ideas between the ones we have cited, and the ones the reviewer has mentioned here. However, we have now referenced these relevant papers in the related work section (Sec. 2) of our paper.
>
> ### **Theoretical Analysis of DSpodFL**
>
> The assumption of uncorrelated random variables arises out of the nature of the problem, and is not a strict assumption that we forcefully make to simplify our analysis. To explain this, note that there are three groups of random variables in our paper, 1) $v_i^{(k)}$ which indicate whether device i conducts local updates at iteration $k$, 2) $\hat{v}_{ij}^{(k)}$ which indicates whether the link between device $i$ and $j$ is being utilized for communication at iteration $k$, and 3) $\epsilon_i^{(k)}$ which are the stochastic gradient noise. None of these random variables have to do anything with one another, as the gradient noise depends on the minibatch size and nothing else. Furthermore, local update indicator depends only on processing capabilities of a device, and is not related in any way to link utilization indicator variable, which in term depends on a communication link’s bandwidth availability.
>
> The theoretical challenges of our paper were in deriving Lemmas 4.1, 4.2 and C.3. The significance of these Lemmas lie in the fact that they can recover theoretical results from previous papers in the literature in the degenerated cases of $d_{min}^{(k)} = 1$ and $\xi=0$.
>
> Regarding the comment of achieving theoretical advantages through sporadic updates, we would like to point out that we do not make such claims. The central message of our paper is that sporadicity can result in reducing the time in seconds it takes to complete each iteration of training, which we illustrate in our experiments. Alongside that, our theoretical analysis guarantees that our ML models of devices converge if they employ our algorithm. To shed light on the advantages of DSpodFL, one would need to theoretically incorporate resource efficiency analysis as well alongside convergence analysis.
>
> ### **Organization of the Paper, and Transient Rates**
>
> We understand how presenting the pseudocode and some notation in the appendix might be confusing. We had to defer the pseudocode and notations to the appendix due to the space constraints of this conference, as they would have taken a whole page of our main text. However, we have explained the steps of our algorithm in Sec. 3.1, and definitions for all used terms and expressions are given whenever an equation is defined, such as equation (3).
>
> Thank you for your comment on the analysis of our algorithm using diminishing step sizes. We have now added a new section in Appendix L, focusing on the convergence behavior of our approach when a diminishing step size policy is employed. In Theorem L.1, we show that a sub-linear rate of $O(\ln{k}/\sqrt{k})$ to the globally optimal solution can be achieved. The convergence rate of $O(\ln{k}/\sqrt{k})$ puts us in line with an extensive body of literature in Distributed Gradient Descent algorithms (e.g., see [1] for a discussion on this.)
>
> [1] Nedić, Angelia, and Alex Olshevsky. "Stochastic gradient-push for strongly convex functions on time-varying directed graphs." IEEE Transactions on Automatic Control 61, no. 12 (2016): 3936-3947.

---

> ### Author Response · Authors · 2023-11-19
>
> ### **Bound for $\rho{(\Phi)}$, and Non-Convex Analysis**
>
> We appreciate you pointing this out. We have now added an upper bound on $\rho(\Phi)$ in Proposition 1, which shows the dependence of the rate on all problem-related parameters, namely step size, number of devices, frequency of computations and communications, and the connectivity of the network graph. Note that $\rho(\Phi)$ is determined by the entries of the  matrix, all of which we present individually and discuss in detail on how they are dependent on problem-related parameters as well..
>
> Please see our response for weakness 2 regarding our theoretical contributions, and the convexity assumption.

---

> > ### Comment · Reviewer_Mn6u · 2023-11-19
> >
> > I have read the reviewer's response, and I maintain my original review.

---

### Official Review · Reviewer_Sqyj · 2023-10-31

**Soundness:** 3 good
**Presentation:** 2 fair
**Contribution:** 1 poor
**Rating:** 3
**Confidence:** 4

**Summary:**

Paper proposes a Decentralized Sporadic Federated Learning (DSpoFL) algorithm which allows both sporadic (i) local gradient updates and (ii) model averaging. Using a constant step-size (when local gradient updates do occur), theoretical  results for DSpoFL are provided to detail the convergence rates of average model error and consensus error. These results require strong convexity assumptions as well as expected sporadicity variables for local gradient updates and communication. Empirical results are provided for simple datasets in convex and non-convex settings.

**Strengths:**

1. Nice theoretical analysis is provided in terms of convergence rates (for consensus and model errors) that are often left out in many FL papers.
2. Tackling heterogeneity in FL is one of the final frontiers that need to be addressed, and so this paper is looking to solve important issues.

**Weaknesses:**

1. The novelty of the idea behind DSpoFL is lacking to me. There is a lot of work in FL on sporadic communication as well as compression and asynchronous methods to speed up the actual communication process. Sporadic local updates is more interesting but just allowing local gradient updates to either happen or not happen (without specifying what is the optimal rate or schedule for this) does not seem to be a large improvement in the field.
2. Strong convexity assumption is ok, but not that realistic in ML today (where most training is non-convex).
3. Empirical testing was on only one smaller dataset, with no relevant baselines (FedAvg is a must to include), and not much improvement is showcased overall (especially when other asynchronous or compression methods can better reduce delay).

**Questions:**

1. How does the limit (sub-optimality gap) that is approached in Equation (9) compare to other classical works like FedAvg (when convex assumptions are used)? This is very important as much of the analysis showcases convergence to a first-order stationary point (FOSP) and not the true optimality gap.
    - If convexity isn't that strong (small $\mu$), then the suboptimality grows inversely $\mathcal{O}(\frac{2}{\mu})$ in Equation (9)
3. Can convergence be shown in the standard FOSP manner with the non-convexity (relaxing convexity assumptions)?
4. There is no guidance for how to know or select the expected connectivity of links between devices as well as the expected sporadicity of local gradient updates. How should they be selected to optimize training and boost performance? Ablation studies are necessary for studying this phenomena.
5. DSpoFL does not seem to outperform Distributed Gradient Descent (DGD) in raw iteration testing. Comparison against FedAvg is a must since multiple local updates before communication also reduces communication costs and is shown to improve accuracy. DSpoFL should be compared with a range of similar communication-efficient algorithms (and should hopefully outperform them).

---

> ### Author Response · Authors · 2023-11-19
>
> ### **Novelty of DSpodFL**
> There is indeed a lot of work on sporadic communication, asynchronous communication and compression in the literature. We have referenced some of the prominent ones in Section 2 of the paper. As the reviewer has correctly mentioned, the sporadic local updates component of our method is the main novelty of our approach. Our main theoretical contribution, on analyzing sporadicity in local updates and communications in a single framework, was actually very difficult to accomplish: the introduction of indicator variables into the update expression (3) of the manuscript, to capture device participation, required us to differentiate between devices which conduct SGD and which do not on a per-iteration basis to arrive at our analytical results in Section 4. Throughout the proofs of Lemmas 4.1&4.2, Proposition 4.1, and Theorem 4.1, different bounds are needed for the corresponding error terms of participating and non-participating devices, which must then be combined together while trying to preserve tight bounds to encapsulate prior theoretical results as special cases. For example, the proof of Proposition 4.1 in Appendix J was perhaps the most involved from the original manuscript, where step by step we simplify the upper bounds of the entries of the matrix $\mathbf{\Phi}^{(k)}$, in order to obtain tight constraints on the step size $\alpha^{(k)}$ that result in the spectral radius of $\mathbf{\Phi}^{(k)}$ being less than $1$. In the revised manuscript, we have also added a new result for the case of using diminishing step sizes in Appendix L: here, we demonstrate that under a diminishing step size policy and an increasing frequency of local updates, a sublinear rate of $\mathcal{O}(\ln{k}/\sqrt{k})$ can be achieved to the globally optimal solution.
>
> It is true that we do not seek to specify an optimal rate or schedule for sporadicity across devices. As we are the first work to introduce sporadicity in local updates to distributed optimization methods, our work provides a foundation for future work which can investigate such scheduling strategies for distributed learning systems. However, note that any scheduling strategy would require a centralized server to coordinate all devices, to determine a jointly optimal rate for all of them to compute local stochastic gradients. Strictly speaking, in the fully-decentralized system setting we consider, this is impractical, as each device needs to locally decide when to perform local updates. In our paper, each device conducts local updates whenever it has sufficient available computational resources to do so, which we model via random indicator variables corresponding to each device’s resource availability.
>
> ### **Strong Convexity Unrealistic**
> We agree that non-convex is more realistic for ML today. This is the reason we have included experiments with non-convex CNN models in our paper. We have shown that all of our results on improving resource efficiency hold for both the convex and the non-convex cases. We have followed an extensive body of literature on distributed ML when doing this, where the theory is derived for the convex or strongly convex case, while numerical experiments are done using both convex and non-convex models.
>
> In terms of theoretical analysis, we have provided the convex analysis to gain intuition on how the consensus error and the optimality error behave over the iterations. Beyond the strong convexity parameter $\mu$, our results show how the convergence rate and the optimality gap depend on step size, number of devices, frequency of computations, frequency of communications, and the connectivity of the communication graph.
>
> Nonetheless, extending these results to the non-convex case should be straightforward, as one would need to form a new error matrix similar to equation (6). Instead of having a term for the distance between the average model and the optimal model, an upper bound on the norm of gradients of the average model should be added. We are working on this and will update the response again once it is completed.

---

> ### Author Response · Authors · 2023-11-19
>
> ### **Datasets and Baselines**
> Regarding baselines, note that FedAvg is intended for the client-server federated setup, i.e., with a central server being present in the system which aggregates models from end devices. It is therefore not a feasible algorithm in this paper, since we are focusing on a fully-decentralized setup. The counterpart of FedAvg here would be the standard DGD (Distributed Gradient Descent) method, which we have included as a baseline.
>
> Regarding datasets, we note that Fashion-MNIST is a very popular choice for evaluating distributed learning, e.g., [2]. Nonetheless, we agree with the reviewer that an additional dataset would strengthen our results. We are in the process of adding this and will update the response again once it is completed.
>
> Across Figures 1&2, the key point we wanted to make is that DSprodFL is the only algorithm that performs consistently well in terms of both transmission delay and processing delay, since it accounts for both sporadic aggregations and sporadic SGDs. Additionally, we would like to point the reviewer to Figures 6&7 in Appendix P.3, where the improvements in terms of transmission, processing, and overall delay are more substantial than in Figures 1&2, for both models and both data distributions. The difference here is that the device and link capabilities are sampled according to beta distributions, rather than uniform distributions, with the resulting probabilities being extremely close to $0$ or $1$, rather than spread between $0$ and $1$. This is more realistic in many decentralized network settings, where devices are extremely heterogeneous in their resources.
>
> Finally, we do agree that compression techniques can help reduce transmission delay further. We mention such techniques, as well as sparsification and quantization methods, in Section 2 of the manuscript. We view these as orthogonal to our work, as our focus is on regulating sporadicity in when devices will communicate (as well as when they will conduct SGD iterations), as opposed to reducing the delay incurred when they actually do communicate. Compression in particular could be applied on top of our method, as well as the baselines we consider, to reduce the transmission delays associated with participating in each training round, and in effect allowing the values of $b_{ij}^{(k)}$ to increase.
>
> ### **Optimality Gap**
> Our analysis shows convergence to the optimality gap, i.e., the difference between the model parameters of devices and the optimal global model. As FedAvg is designed for federated setups with a central server being present in the system, we do not believe this is the right comparison point, as our approach is for a fully-decentralized system. Instead, we can compare our analysis with other seminal works in the distributed optimization literature. In particular, we can verify the dependence on $\mathcal{O}(1/\mu)$ by checking Theorem 2 in Sec. 6.1 of [1]: it can be observed that for the similar strongly-convex case, there is a dependence on $1/\mu$ in the upper bound.
>
> Note additionally that most works (like [1]) in DSGD literature provide bounds for Cesaro sums of consensus errors and optimization errors (see Theorem 2 of [1]), while we present upper bounds for the actual errors themselves (see equations (6)-(9) in our paper). Hence, our results are arguably stronger in the sense that the convergence of model parameters (which we establish) implies convergence of their Cesaro sums, but not vice-versa.
>
> ### **Relaxing Convexity Assumptions**
> Yes, this is possible. Many papers in distributed ML showing convergence for the convex case have been later generalized to the non-convex case. We originally intended to keep the non-convex analysis for future work, as the theoretical results included in the paper are already comprehensive. However, showing that the norm of the gradient of the average model reaches $0$ as the iterations approach infinity should be straightforward, by having an upper bound on the norm of gradients of the average model instead of the distance between the average model and the optimal model. We are working on this and will update the response again once it is completed.

---

> ### Author Response · Authors · 2023-11-19
>
> ### **Parameter Selection**
> The premise of our algorithm is to reduce the processing and communication delays given a fixed network configuration, with specific device and link capabilities, i.e., processing units and bandwidth, and a particular distribution of data among devices. Consequently, we do not seek to optimize the inter-device link connections or the local gradient updates to boost performance, but rather to utilize the existing links and device capabilities better in order to reduce delay, through steps 18-27 of our algorithm.
>
> Despite this, we did provide an ablation study on the effect of graph connectivity in Appendix P.1, where we see (perhaps unsurprisingly) that a fully-connected network graph would always be preferable. An ablation study on the expected sporadicity of local gradient updates was also done, and it is given in Appendix P.3 and Appendix P.5.
>
> ### **DGD and FedAvg**
> We do not see a way to fairly compare FedAvg with DSpodFL, as FedAvg is tailored for the federated setup where a centralized server is present. In terms of performance compared to DGD in raw iterations testing, we had attempted to explain in Sec. 5.2 why this is the case, due to each device participating in each iteration irrespective of the time taken to complete each iteration. In practice, the overall required time in seconds is a better measure to compare different algorithms with each other, to determine which one can be executed faster. We illustrate the superiority of our algorithm in this sense in Figures 1, 2, 7, and 8.
>
> It is true that conducting multiple local updates before communication reduces communication costs. A few papers which introduce distributed algorithms doing that are referenced in the Related work section (Sec. 2). In fact, DSpodFL’s theoretic and algorithmic framework implicitly allows for multiple local updates before communication as well, on a per-device basis, since the local updates and the communications are both sporadic. Say for example a device does not communicate with its neighbors for $k$ iterations due to the sporadic communications component of DSpodFL, but conducts local updates in $d \le k$ of these iterations. This corresponds to this device conducting $d$ local updates before communicating with its neighbors. We have clarified this in Sec. 2.
>
> [1] Anastasia Koloskova, Nicolas Loizou, Sadra Boreiri, Martin Jaggi, and Sebastian Stich. A unified theory of decentralized sgd with changing topology and local updates. In International Conference on Machine Learning, pp. 5381–5393. PMLR, 2020.
>
> [2] Huang, Yan, Ying Sun, Zehan Zhu, Changzhi Yan, and Jinming Xu. "Tackling Data Heterogeneity: A New Unified Framework for Decentralized SGD with Sample-induced Topology." In International Conference on Machine Learning, pp. 9310-9345. PMLR, 2022.

---

> > ### Comment · Reviewer_Sqyj · 2023-11-20
> > **Rebuttal Response**
> >
> > Thank you to the authors for their extensive reply. Apologies for my delayed response.
> >
> > I first want to mention that it is incorrect to state that FedAvg is only for centralized settings. The premise of FedAvg is that devices can take multiple local update steps before averaging. In DSGD, only one gradient step is taken before averaging. Multiple local gradient steps both has been shown to improve performance and reduce communication costs. An example of such an algorithm is DFedAvgM in "Decentralized Federated Averaging Sun et al. 2021". A decentralized version of FedAvg can be compared DSpodFL, and should be within the empirical results. Furthermore, I believe it is still worrying that DSGD outperforms DSpodFL in pure raw iteration testing. I would assume the gap would grow further when comparing against DFedAvgM.
> >
> > Second, the lack of non-convex results is still a large weakness as many relevant and related works have theoretical results for non-convex settings (including the paper I provided above). Just having small-scale empirical results on non-convex problems is not sufficient.
> >
> > Third, it still is not provided what the comparable convergence rates are for related works as well as the assumptions that they use (e.g, non-convexity). Just providing papers in the related works is not sufficient and makes it difficult for the reader to digest the novelty of the theoretical results. While the proofs may be very involved, the results must be clearly compared to related papers to justify the overall novelty of the paper.
> >
> > I maintain my original review and score.

---

### Official Review · Reviewer_RbX5 · 2023-11-03

**Soundness:** 3 good
**Presentation:** 3 good
**Contribution:** 2 fair
**Rating:** 5
**Confidence:** 4

**Summary:**

This paper considers computing and communication resource efficiency problems in decentralzied federated learning over undirected communication networks. To solve this problem, the authors propose the DSpodFL algorithm, which introduces sporadic local stochastic gradient descent (SGD) computations and model aggregations among nodes. Specifically, at each iteration, each node conducts local SGD with some probability and each communication link activates with some probability. Their analysis shows that the proposed DSpodFL achieves a geometric convergence rate to a neighborhood of the globally optimal solution for a constant SGD step size, in a strongly convex and smooth setting. They conduct some numerical experiments to demonstrate the superior performance of DSpodFL compared to the existing baselines.

**Strengths:**

1. Their proposed algorithm employing sporadic SGDs and sporadic aggregations, seems new to the reviewer.

2. They provide linear convergence results for their proposed DSpodFL and show that the final optimality gap (steady-state error) is dimishing with the step size.

3. The manuscript is well-organized and is easy to follow.

**Weaknesses:**

1. The dependence of linear convergence rate $\rho \left( \Phi \right)$ on problem-related parameters such as step size , number of devices, frequency of computations, frequency of communications and the connectivity of the communication graph is not discussed in their theoretical results, which is the main limitation in the convergence property of DSpodFL.

2. Moreover, there are no theoretical comparisons to the baselines: e.g., to [1].

3. Limited node scability: Only 10 nodes is considered in their numerical experiments. How does the proposed DspodFL scale with repect to the node number is not clear.

4. Some baselines are missing in experiments, such as AD-PSGD [2] and OSGP [3].

[1] Anastasia Koloskova, Nicolas Loizou, Sadra Boreiri, Martin Jaggi, and Sebastian Stich. A unified theory of decentralized sgd with changing topology and local updates. In International Conference on Machine Learning, pp. 5381–5393. PMLR, 2020.

[2] Xiangru Lian, Wei Zhang, Ce Zhang, and Ji Liu. Asynchronous decentralized parallel stochastic gradient descent. In International Conference on Machine Learning, pp. 3043–3052. PMLR, 2018.

[3] Mahmoud Assran, Nicolas Loizou, Nicolas Ballas, and Mike Rabbat. Stochastic gradient push for distributed deep learning. In International Conference on Machine Learning, pp. 344–353. PMLR, 2019.

The reviewer is happy to raise the score if the authors can address the above concerns.

**Questions:**

Refer to weakness.

**Details Of Ethics Concerns:**

N.A.

---

> ### Author Response · Authors · 2023-11-19
>
> ### **Convergence Rate $\rho{(\Phi)}$**
> We appreciate you pointing this out. We have now added an upper bound on $\rho{(\Phi)}$ in Proposition 1, which shows the dependence of the rate on all problem-related parameters, namely step size, number of devices, frequency of computations and communications, and the connectivity of the network graph. Note that $\rho{(\Phi)}$ is determined by the entries of the  matrix. We present these individually and discuss how they are dependent on problem-related parameters in Lemmas 4.1 and 4.2 in Sec. 4.2. The corresponding analysis can be found in step 5 of Appendix J.
>
> ### **Theoretical Comparisons to Baselines**
> We have now inserted a paragraph after Theorem 4.1 in our paper, summarizing the theoretical results obtained in related works, including [1] mentioned by the reviewer. However, there is a key reason why a direct, fair comparison between these important previous works and ours cannot be made: they provide convergence results for the cesaro sum of model parameters, or cesaro sum of expected errors in the case of [1], even for strongly-convex models (please see Theorem 2 in Sec. 6.1 in [1]). We, on the other hand, present convergence analysis for the model parameters themselves, as shown in equations (6)-(9). Hence, our results are stronger in that the convergence of model parameters (which we establish) implies convergence of their Cesaro sums, but not vice-versa.
>
> There are also other important papers showing convergence using the cesaro sums of consensus and optimization errors, i.e., [4] and [5]. For example, see Theorem 4 in [4] and Theorem 5.3 in [5]. We have now summarized these in Sec. 4.2 of the manuscript as well.
>
> ### **Scalability Experiments**
> Thank you for your insightful comment. We have done a new set of experiments by varying the number of nodes in the network, i.e., $m$, and included them in Appendix P.4. The new experiments include numerical evaluations for $m=10,20,30,40,50$. Overall, we see that our DSpodFL algorithm outperforms all other baselines regardless of the number of devices available in the system, using either a convex or a non-convex model, for both IID and non-IID cases.
>
> ### **Other Baselines**
> The AD-PSGD method can be viewed as a special case of sporadic aggregations, which we consider as a baseline in Section 5. This can be verified by looking at the first equation in Sec. 4 of [2], where the update rule per iteration of AD-PSGD is given. We can see that this update rule follows from equation (3) in our paper if we set all $v_i^{(k)} = 1$, which corresponds to only conducting sporadic aggregations.
>
> We have not considered OGSP as a baseline in our paper since it (alongside other gradient push methods) is an algorithm designed for directed graphs. Our scheme is based on the assumption that the links connecting the devices together are bidirectional. When the underlying network graph is undirected, employing approaches designed for directed graphs results in performance degradation. This is because the challenge in directed graphs is that a doubly stochastic transition matrix cannot be designed. Therefore, what OGSP and other push-sum methods do is to try to design just a row-stochastic or a column-stochastic matrix, and then make an adjustment to it (in the push-sum phase). Since in undirected graphs we can easily design a doubly-stochastic matrix, overcomplicating the algorithm is unnecessary.
>
> [4] Lian, Xiangru, Ce Zhang, Huan Zhang, Cho-Jui Hsieh, Wei Zhang, and Ji Liu. "Can decentralized algorithms outperform centralized algorithms? a case study for decentralized parallel stochastic gradient descent." Advances in neural information processing systems 30 (2017).
>
> [5] Sundhar Ram, S., Angelia Nedić, and Venugopal V. Veeravalli. "Distributed stochastic subgradient projection algorithms for convex optimization." Journal of optimization theory and applications 147 (2010): 516-545.

---

> > ### Comment · Reviewer_RbX5 · 2023-11-23
> > **Response to the rebuttal**
> >
> > The reviewer thank the authors' effort in preparing the reply which have addressed most of the reviewer's concerns. The reviewer would maintain the score.

---

### Meta-Review · Area_Chair_F2nm · 2023-12-14

**Metareview:**

The paper studies intermittent computation and communication strategies (called "sporadicity" in the paper) in the context of decentralized optimization in the strongly convex regime. No reviewers suggested acceptance. I have looked at the paper and agree with the reservations. The bounds obtained do not show benefit of the new strategies. Moreover, there exist numerous papers where such "sporadic" strategies lead to provable gains. Some of these papers are even cited (e.g., the ProxSkip work of Mishchenko et al), but for some reasons the authors do not mention their results are worse than those in these works. Other critical works are not cited nor compared to at all (e.g., the GradSkip work of Maranyjan et al which also allows different clients to perform a different number of computations; or the VR-ProxSkip work of Yi et al which accounts for both communication and computation). In fact, the authors do not seem to be aware of these works since they say "However, these techniques are focused on communication efficiency, and do not take processing efficiency into account." Numerical results should have included the SplitSkip method described in Mishchenko et al as those results constitute the current theoretical SOTA for decentralized methods with local steps. I believe the results would be strongly in favor of SplitSkip.

In summary, I find it very hard to appreciate the results. They do not offer any quantitative predictions, and are not compared to their closest competitors. The other reviewers also found the results not convincing.

**Justification For Why Not Higher Score:**

Higher score would mean acceptance, but this paper can't be accepted due to the reasons mentioned above.

**Justification For Why Not Lower Score:**

N/A

---

### Decision · Program_Chairs · 2024-01-16

Reject